taxonomy and systematics/evolution/biogeography

bibliometrics, biodiversity crisis, Journal Impact Factor (JIF), scientometrics, systematics

**Author for correspondence:**
Ângelo Parise Pinto
e-mail: appinto@ufpr.br

# Are publications on zoological taxonomy under attack?

Ângelo Parise Pinto[1], Gabriel Mejdalani[2], Ross Mounce[3], Luís Fábio Silveira[4], Luciane Marinoni[1] and José Albertino Rafael[5]

[1]Departamento de Zoologia, Universidade Federal do Paraná, PO Box 19020, 81531-980, Curitiba, Paraná, Brazil
[2]Departamento de Entomologia, Museu Nacional, Universidade Federal do Rio de Janeiro, Rio de Janeiro, Rio de Janeiro, Brazil
[3]Arcadia Fund, Sixth Floor, 5 Young Street, London W8 6EH, UK
[4]Museu de Zoologia da Universidade de São Paulo, São Paulo, São Paulo, Brazil
[5]Instituto Nacional de Pesquisas da Amazônia, Manaus, Amazonas, Brazil

ÂPP, 0000-0002-1650-5666

Taxonomy is essential to biological sciences and the priority field in face of the biodiversity crisis. The industry of scientific publications has made extensive promotion and display of bibliometric indexes, resulting in side effects such as the Journal Impact Factor™ (JIF) mania. Inadequacies of the widely used indexes to assess taxonomic publications are among the impediments for the progress of this field. Based on an unusually high proportion of self-citations, the mega-journal *Zootaxa*, focused on zoological taxonomy, was suppressed from the Journal Citation Reports (JCR, Clarivate™). A prompt reaction from the scientific community against this decision took place exposing myths and misuses of bibliometrics. Our goal is to shed light on the impact of misuse of bibliometrics to the production in taxonomy. We explored JCR's metrics for 2010–2018 of 123 zoological journals publishing taxonomic studies. *Zootaxa*, with around 15 000 citations, received 311% more citations than the second most cited journal, and shows higher levels of self-citations than similar journals. We consider *Zootaxa*'s scope and the fact that it is a mega-journal are insufficient to explain its high level of self-citation. Instead, this result is related to the '*Zootaxa* phenomenon', a sociological bias that includes visibility and potentially harmful misconceptions that portray the journal as the only one that publishes taxonomic studies. Menaces to taxonomy come from many sources and the low bibliometric indexes, including JIF, are only one factor among a range of threats. Instead of being focused on statistically illiterate journal metrics endorsing the

villainy of policies imposed by profit-motivated companies, taxonomists should be engaged with renewed strength in actions directly connected to the promotion and practice of this science without regard for citation analysis.

## 1. Introduction

Every middle of year large scientific data analytics companies, the American-British Clarivate™ (InCites™) and the Dutch RELX™ Elsevier B.V. (Scopus®), release their metrics for scientific journals indexed in their databases, among them the Journal Impact Factor™ (JIF) and the CiteScore™, respectively. These metrics have been adopted as major means of research assessment by many countries as the sole measure of the quality of the research produced in their universities and institutes. Generally, funding for research in these institutions is derived from the taxes paid by the citizens of their respective countries. This policy produces a sort of quest or JIF mania for publishing in higher-ranked journals [1]. Therefore, depending on the impact factor, a researcher is perceived to have better chances of advancing in her/his career, earning prestige, winning grants, etc. Thus, these metrics have a strong impact on how and what scientific investigation can currently be conducted.

On the last day of June of 2020, an interruption to concerns about the Covid-19 pandemic affected taxonomists around the world. An issue broke through the media due to the involvement of zoologists from many countries: the suppression of the mega-journal *Zootaxa*, a journal focused on zoological taxonomy, from the Journal Citation Reports™ (JCR) Science Edition metrics by Clarivate. Based on a high proportion of self-citations, along with another 32 journals from the 12 000 in the JCR database, *Zootaxa* would not receive a value of Journal Impact Factor (JIF) for 2019; however, it would keep the values for previous years and still be indexed on the Clarivate Analytics platform.

By this time, with the publication of JIF 2019 by JCR, which called the attention of editors and authors who were eager to see how journals were ranked, passionate discussions arose because of *Zootaxa*'s suppression. A prompt reaction, rarely seen before in this community, through many letters of support to *Zootaxa* and petitions from several societies and researchers, forced Clarivate to review its decision. We believe that the suppression of *Zootaxa* entails so many unique elements that it needs a closer inspection. Some supporting letters could actually be considered political manifestos and others were very naive, not to say alarmist or simply inaccurate in interpreting the suppression as a new attack to taxonomy as a science. Among the utterly passionate arguments was the one that *Zootaxa* is the single vehicle to publish taxonomic papers nowadays, a statement obviously far away from the truth. At the end of July, in a short statement on Twitter, Clarivate announced that *Zootaxa* and the *International Journal of Systematic and Evolutionary Microbiology* would be reconsidered in the regular refresh of the JCR, which was published in September. Cases of suppression are common and not unique to the Clarivate platform, most of them being due to accusations of artificial boost or inflation of impact factors (e.g. [2]). A particular case, almost a decade ago, had a wide repercussion among researchers when four journals edited in Brazil were suppressed under accusation of a citation-stacking scheme, a sort of cartel in which self-citations are exchanged among a group of journals [3]. Noticeably, cases of suppression in the past hardly received any sympathy from the scientific community, except from people directly involved as editors, perhaps as a signal that sectors of the academic community agreed with the suppression and considered that the affected journals deserved such 'punishment.' Once discarded from JIF, a journal is excluded from the gold rush of academia targeting high-impact outlets.

In a system full of anachronisms, in which traditional journals supported by museums or scientific societies are struggling to survive and the commercialized scientific publishing industry is led by giant publishers such as John Wiley & Sons, Elsevier and Springer Nature, among others, with profit margins higher than those of major players in drug, bank and auto companies [4], it is at least curious to perceive the commotion around the suppression of *Zootaxa*. We became intrigued and thus decided to provide some reflections aiming to shed light on underlying aspects of this issue. We believe that many of the arguments that were given in the supporting letters are based on misunderstandings about these metrics or are biased by personal interests due to the pressure to publish in high-impact journals. In addition, some points are also potentially misplaced. Bibliometric data are plagued by myths and misunderstandings [5].

Our goal is to shed some light on the impacts of the adoption of bibliometrics to the production in taxonomy by discussing the following questions. Can the suppression of any journal from JIF really

affect the volume and quality of production in the taxonomic field? What are the consequences of *Zootaxa* suppression to taxonomy as a science? What does journal 'self-citation' mean? What is the average JIF for taxonomic journals? Is *Zootaxa* a victim of its success? So, what is actually going on? To properly address these questions, we first need to clarify a few concepts and dig further into the relationship between bibliometrics and taxonomic journals, impact measures of scientific publications, and the role of individuals and mega-corporations in this arena.

## 2. Material and methods

Data on a total of 123 journals were compiled after the filtering described below. We explored citation data including JIF, most-cited journals and self-citation metrics from the Journal Citation Reports (Web of Science Core Collection™) of the last 9 years (2010–2018) of eight of the top 10 zoological journals (TTJ, only eight are included in JCR) when the number of new available names (based on the last 5 years of ION/Zoological Records™—ZR [6]) is considered. We also checked up for journals focused on or regularly publishing taxonomic papers included in the Zoology and Entomology categories. The period of 9 years was adopted because *Zootaxa* was suppressed from the current edition of JCR and only reappeared in September of 2020 after conclusion of this study. Among 168 journals in Zoology and 101 in Entomology, 73 and 48 (both numbers include 'plus' *Zootaxa* and TTJ) were selected. Journals included in both Zoology and Entomology categories were subsequently considered in the Zoology category. For the analysis including all data, after excluding duplicates from the three selected categories, the top 10 journals (TTJ) were considered independently with eight, Zoology with 69 and Entomology with 46 journals. The strategy for journal selection is depicted graphically as a Venn diagram (figure 1).

In order to analyse the selected journals with available data from 2010 to 2018, a descriptive statistics approach including arithmetic mean of the bibliometric variables, their standard deviations, and the ratio between JIF without self-citations (journal) and JIF (including journal self-citations) was used to investigate the influence of self-citations on JIF. This approach was conducted among the top 10 journals (TTJ) in ZR and those in Zoology and Entomology categories when all data were analysed together. The percentage of self-citations and the ratio between JIF without self-citations and JIF per year of the journals between quartiles 2 and 3 (Q2 and Q3) in their categories (Entomology and Zoology) and TTJ totalling 68 periodicals were analysed with one-way ANOVA using permutations for linear models via lmPerm package [7] followed by a Tukey honestly significant difference (Tukey HSD) test in R software [8]. Thus, journals with similar scope and JIF to *Zootaxa* were considered. In practice, all journals publishing taxonomic papers with JIF 2019 ranging from 0.25 to 2.315 in the categories Zoology and Entomology were included. This last criterion can be considered arbitrary because the proportion of taxonomic papers is very dissimilar among the journals. Clearly, *Zootaxa* and *ZooKeys*, for example, have a greater number of taxonomic papers, even when the fact that these journals accept studies on different subjects of biological sciences is considered. However, for the purposes of our discussion, it is reasonable to consider such journals as similar in scope. The comparison is not easy because the original scope of *Zootaxa* is unique, due to its intent of 'rapid publication of high-quality papers on any aspect of systematic zoology' and its focus on long papers. However, *Zootaxa* publishes today virtually any subject associated with zoological taxonomy/ systematics, including biographies and points of view on theoretical subjects. Therefore, it is fair to conclude that all kinds of zoological papers are published in that journal, except those essentially dealing with ecological or experimental issues. The list of the journals, their publishing model and selected metrics are available in table 1 and electronic supplementary material, file 1. Note that the categorization of journals follows Web of Science's criteria, so that many taxonomic journals that publish studies in the areas of zoology or entomology may not have been included simply because they are listed in any one of the other 178 categories in the Science Edition database. The publishing model was determined based on journals guidelines/instructions to authors.

## 3. Results

A small fraction of the 123 taxonomic journals investigated adopt mandatory Article Processing Charges for Gold Open Access (APC-GOA) models (18.7%). Diamond open access (DOA) represents 22.0%, whereas the largest percentage (59.3%) of journals are based on hybrid models with paywall to access their content, usually through readers' payment/subscription (table 1). A few journals, published in

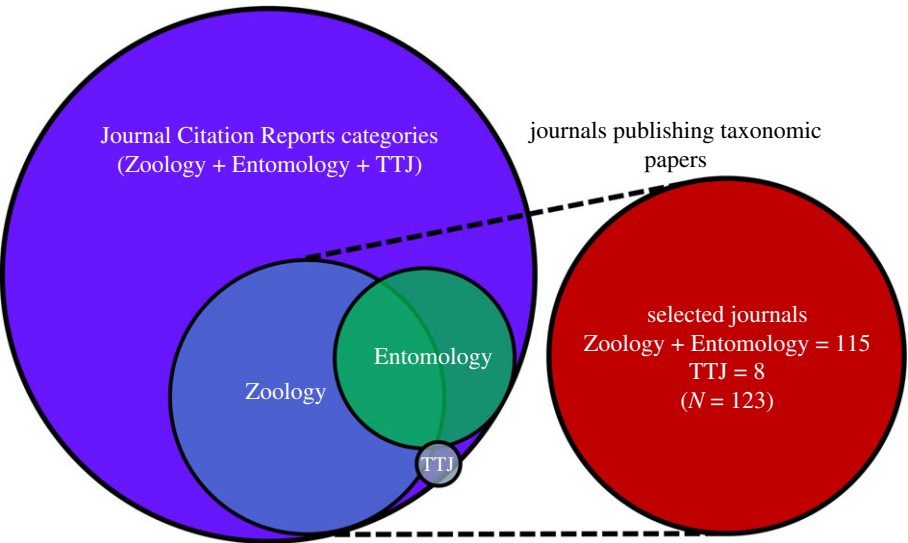

**Figure 1.** Scheme of selection of the 123 journals used in this study from the Web of Science Core Collection, Journal Citation Reports (JCR) Science Edition database of Clarivate. The selection was carried out from a total of 269 journals in Zoology ($n = 69$) and Entomology ($n = 46$) categories, plus top 10 zoological journals (TTJ, just eight are on JCR). Some journals are in more than one category, resulting in overlapping, two TTJ are not indexed in JCR.

distinct platforms maintained by societies, require page charges from authors, irrespective of them being associated or not to the society, they also have their contents paywalled; thus, these journals have both authors' and readers' charges (e.g. *Journal of the Kansas Entomological Society* and *Malacologia*).

The average levels of self-citations in the period of 2010–2018 range from 0.0 to 24.2% in Zoology, 0.0–34.9% in Entomology and 4.6–34.9% in TTJ. For the last 5 years (2014–2018), these levels are 0.0–27.3% in Zoology, 0.0–36.4% in Entomology and 4.5–36.4% in TTJ. The upper bounds of self-citation in the Entomology and TTJ categories are due to *Systematic and Applied Acarology*; excluding this journal, the maximum level of self-citation for Entomology is 21.4% (*Coleopterists Bulletin*) for 2010–2018 and 27.3% (*Odonatologica*) for 2014–2018, while for TTJ they are 26.3% and 27.5% (both correspond to *Cretaceous Research*). In comparison to all other journals, excluding *Systematic and Applied Acarology*, the mean levels of self-citation are higher for *Zootaxa* than for any other journal in Zoology (figure 2), Entomology (figure 3) and TTJ categories (figure 4), being 34.9% for 2010–2018 and 37.6% for 2014–2018 in *Zootaxa*. The levels of self-citation have gradually increased in *Zootaxa* from 27.99% in 2010 to 52.7% in 2018 (electronic supplementary material, file 2). The percentage of self-citations for 2010–2018 is higher in *Zootaxa* and similar only to *Systematic and Applied Acarology* (figure 5). The one-way ANOVA with permutation of the Q2 and Q3 journals, plus TTJ (68 journals), was significant (d.f. $= 67$, $p < 0.001$) for both the ratio of JIF without self-citation and JIF (JIF ratio), as well as to the proportion of self-citation (level of self-citation). The Tukey HSD test revealed that *Zootaxa*'s JIF ratio is significantly different from almost all journals except the TTJ *Journal of Palaeontology* and *Systematic and Applied Acarology* and the following Zoology journals: *Asian Herpetological Research*, *Ichthyological Exploration of Freshwaters*, *International Journal of Odonatology*, *Nematology*, *Neotropical Ichthyology*, *Nota Lepidopterologica* and *Vertebrate Zoology* (*p*-value $< 0.001$). *Zootaxa*'s level of self-citation is also significantly different from most journals except for the TTJ *Cretaceous Research*, *Journal of Palaeontology*, and *Systematic and Applied Acarology*, and the Zoology journal *Nematology* (*p*-value $< 0.001$).

Influence of self-citations on JIF comparing all 123 journals is almost insignificant to boost this metric because most journals from the three categories (Entomology, Zoology and TTJ) have similar means of the ratio between JIF without self-citations and JIF for 2010–2018, except *Shilap-Revista de Lepidopterologia*, a journal devoted to butterflies and moths, *Insects*, published by Multidisciplinary Digital Publishing Institute (MDPI), and *Zootaxa* (figure 6). Some journals have large intervals based on s.d. in the influence of self-citation on JIF, while journals such as *Zootaxa* have a more constant influence of self-citation. For instance, in *Zootaxa* this ratio ranges from 0.55 to 0.60 and JIF reduces 39.6–45.6% when self-citations are excluded (figure 6; electronic supplementary material, file 1). *Zootaxa*, with around 15 000 citations, received 311% more citations than the second most cited

**Table 1.** Journals, and their publishing model, indexed in Journal Citation Reports (Web of Science Core Collection™) and that publish taxonomic studies included in the Zoology and Entomology categories plus the top 10 zoological journals (TTJs) in number of new taxa in the last 5 years based on the Zoological Records. APC-GOA, gold open access through payment of article processing charges; DOA, diamond open access; GOA, gold open access; Hybrid, optional payment of gold open access, access to content via subscription (paywall).

| journal | abbreviation used in JCR | category | ISSN | publisher | publishing model |
|---|---|---|---|---|---|
| Acarologia | ACAROLOGIA | Entomology | 0044-586X | Centre de Biologie pour la Gestion des Populations, France | DOA |
| Acta Amazonica | ACTA AMAZON | Zoology | 0044-5967 | INPA/SciELO | DOA |
| Acta Chiropterologica | ACTA CHIROPTEROL | Zoology | 1508-1109 | Acta Chiropterologica, published by the Museum and Institute of Zoology at the Polish Academy of Sciences, is devoted solely to the study and discussion of bats. | DOA |
| Acta Entomologica Musei Nationalis Pragae | ACTA ENT MUS NAT PRA | Entomology | 1804-6487 | BioOne/ Museum and Institute of Zoology, Polish Academy of Sciences | Hybrid/APC-GOA [?] |
| Acta Zoologica Academiae Scientiarum Hungaricae | ACTA ZOOL ACAD SCI H | Zoology | 1217-8837 | Hungarian Academy of Sciences | DOA |
| African Entomology | AFR ENTOMOL | Entomology | 1021-3589 | BioOne/Entomological Society of Southern Africa | Hybrid/APC-GOA |
| African Invertebrates | AFR INVERTEBR | Zoology | 1681-5556 | Pensoft Publishers | APC-GOA |
| African Zoology | AFR ZOOL | Zoology | 1562-7020 | Taylor & Francis Group | Hybrid/APC-GOA |
| American Malacological Bulletin | AM MALACOL BULL | Zoology | 0740-2783 | The Sheridan Press | Hybrid/APC-GOA |
| American Museum Novitates | AM MUS NOVIT | Zoology | 0003-0082 | BioOne/American Museum of Natural History | APC-GOA |
| Amphibia-Reptilia | AMPHIBIA-REPTILIA | Zoology | 0173-5373 | Brill Academic Publishers | Hybrid/APC-GOA |
| Annals of Carnegie Museum | ANN CARNEGIE MUS | Zoology | 0097-4463 | BioOne/Carnegie Museum | Hybrid/APC-GOA [?] |
| Annals of the Entomological Society of America | ANN ENTOMOL SOC AM | Entomology | 0013-8746 | Oxford University Press | Hybrid/APC-GOA |
| Annales de la Societe Entomologique de France | ANN SOC ENTOMOL FR | Entomology | 0037-9271 | Taylor & Francis Group | Hybrid/APC-GOA |
| Aquatic Insects | AQUAT INSECT | Entomology | 0165-0424 | Taylor & Francis Group | Hybrid/APC-GOA |

(Continued.)

**Table 1.** (*Continued.*)

| journal | abbreviation used in JCR | publisher | ISSN | category | publishing model |
|---|---|---|---|---|---|
| Arthropod Systematics and Phylogeny | ARTHROPOD SYST PHYLO | Senckenberg Naturhistorische, Germany | 1863-7221 | Entomology | DOA |
| Arthropoda Selecta | ARTHROPODA SEL | KMK Scientific Press/Zoological Museum MGU | 0136-006X | Entomology | DOA |
| Asian Herpetological Research | ASIAN HERPETOL RES | Chinese Academy of Sciences/Science Press | 2095-0357 | Zoology | APC-GOA [?] |
| Asian Myrmecology | ASIAN MYRMECOL | International Network for the Study of Asian Ants | 1985-1944 | Entomology | DOA |
| Austral Entomology | AUSTRAL ENTOMOL | John Wiley & Sons | 2052-1758 | Entomology | Hybrid/APC-GOA |
| Bulletin of Insectology | B INSECTOL | Department of Agricultural and Food Sciences, Italy | 1721-8861 | Entomology | DOA |
| Belgian Journal of Zoology | BELG J ZOOL | Royal Belgian Zoological Society and the Royal Belgian Institute of Natural Sciences | 0777-6276 | Zoology | DOA |
| Caldasia | CALDASIA | Universidad Nacional de Colombia/SciELO | 0366-5232 | Zoology | DOA |
| Canadian Entomologist | CAN ENTOMOL | Cambridge University Press | 0008-347X | Entomology | Hybrid/APC-GOA |
| Canadian Journal of Zoology | CAN J ZOOL | Canadian Science Publishing | 0008-4301 | Zoology | Hybrid/APC-GOA |
| Coleopterists Bulletin | COLEOPTS BULL | BioOne/The Coleopterists Society | 0010-065X | Entomology | Hybrid/APC-GOA |
| Contributions to Zoology | CONTRIB ZOOL | Brill Academic Publishers | 1383-4517 | Zoology | APC-GOA |
| Copeia | COPEIA | BioOne/American Society of Ichthyologists and Herpetologists (ASIH) | 0045-8511 | Zoology | APC-GOA |
| Cretaceous Research | CRETACEOUS RES | Elsevier B.V. | 0195-6671 | Palaeontology | Hybrid/APC-GOA |
| Current Herpetology | CURR HERPETOL | BioOne/The Herpetological Society of Japan | 1345-5834 | Zoology | Hybrid/APC-GOA |
| Cybium | CYBIUM | Société Française d'Ichtyologie | 0399-0974 | Zoology | Hybrid/APC-GOA |
| Deutsche Entomologische Zeitschrift | DEUT ENTOMOL Z | Pensoft Publishers | 1435-1951 | Entomology | APC-GOA |
| Entomologica Americana | ENTOMOL AM NY | BioOne/The New York Entomological Society | 1947-5136 | Entomology | Hybrid/APC-GOA |

(*Continued.*)

**Table 1.** (Continued.)

| journal | abbreviation used in JCR | category | ISSN | publisher | publishing model |
|---|---|---|---|---|---|
| Entomologica Fennica | ENTOMOL FENNICA [merged with Annales Zoologici Fennici] | Entomology | 0785-8760 | Finnish Zoological and Botanical Publishing Board | APC-GOA |
| Entomological News | ENTOMOL NEWS | Entomology | 0013-872X | BioOne/The American Entomological Society | Hybrid/APC-GOA |
| Entomological Research | ENTOMOL RES | Entomology | 1738-2297 | John Wiley & Sons | Hybrid/APC-GOA |
| Entomological Science | ENTOMOL SCI | Entomology | 1343-8786 | John Wiley & Sons | Hybrid/APC-GOA |
| European Journal of Entomology | EUR J ENTOMOL | Entomology | 1210-5759 | Institute of Entomology of the Biology Centre, Czech Academy of Sciences | APC-GOA |
| European Journal of Taxonomy | EUR J TAXON | Zoology | 2118-9773 | EJT consortium | DOA |
| Florida Entomologist | FLA ENTOMOL | Entomology | 0015-4040 | BioOne/Florida Entomological Society | APC-GOA |
| Gayana | GAYANA | Zoology | 0717-652X | Universidad de Concepción, Chile/SciELO | APC-GOA |
| Herpetological Journal | HERPETOL J | Zoology | 0268-0130 | British Herpetological Society | Hybrid/APC-GOA |
| Herpetological Monographs | HERPETOL MONOGR | Zoology | 0733-1347 | The Herpetologists' League | Hybrid/APC-GOA |
| Herpetologica | HERPETOLOGICA | Zoology | 0018-0831 | The Herpetologists' League | Hybrid/APC-GOA |
| Hystrix-Italian Journal of Mammalogy | HYSTRIX | Zoology | 0394-1914 | Associazione Teriologica Italiana | DOA |
| Ichthyological Exploration of Freshwaters | ICHTHYOL EXPLOR FRES | Zoology | 0936-9902 | Verlag Dr Friedrich Pfei | Hybrid/APC-GOA [?] |
| Ichthyological Research | ICHTHYOL RES | Zoology | 1341-8998 | Springer Nature | Hybrid/APC-GOA |
| Iheringia Serie Zoologia | IHERINGIA SER ZOOL | Zoology | 0073-4721 | Museu de Ciências Naturais, SEMA, Brazil/SciELO | DOA |
| Insect Systematics & Evolution | INSECT SYST EVOL | Entomology | 1399-560X | Pensoft Publishers | APC-GOA |
| Insects | INSECTS | Entomology | 2075-4450 | Multidisciplinary Digital Publishing Institute (MDPI) | APC-GOA |

**Table 1.** (*Continued.*)

| journal | abbreviation used in JCR | publisher | ISSN | category | publishing model |
|---|---|---|---|---|---|
| International Journal of Acarology | INT J ACAROL | Taylor & Francis Group | 0164-7954 | Entomology | Hybrid/APC-GOA |
| International Journal of Odonatology | INT J ODONATOL | Taylor & Francis Group | 1388-7890 | Entomology | Hybrid/APC-GOA |
| International Journal of Primatology | INT J PRIMATOL | Springer Nature | 0164-0291 | Zoology | Hybrid/APC-GOA |
| Invertebrate Systematics | INVERTEBR SYST | CSIRO Publishing | 1445-5226 | Zoology | Hybrid/APC-GOA |
| Journal of Arachnology | J ARACHNOL | American Arachnological Society | 0161-8202 | Entomology | Hybrid/APC-GOA |
| Journal of Asia-Pacific Entomology | J ASIA PAC ENTOMOL | Elsevier B.V. | 1226-8615 | Entomology | Hybrid/APC-GOA |
| Journal of Conchology | J CONCHOL | The Conchological Society of Great Britain and Ireland | 0022-0019 | Zoology | Unknown |
| Journal of Crustacean Biology | J CRUSTACEAN BIOL | Oxford University Press | 0278-0372 | Zoology | Hybrid/APC-GOA |
| Journal of the Entomological Research Society | J ENTOMOL RES SOC | Gazi Entomological Research Society (GERS) | 1302-0250 | Zoology | DOA [?] |
| Journal of Helminthology | J HELMINTHOL | Cambridge University Press | 0022-149X | Zoology | Hybrid/APC-GOA |
| Journal of Herpetology | J HERPETOL | Society for the Study of Amphibians and Reptiles | 0022-1511 | Zoology | Hybrid/APC-GOA |
| Journal of Hymenoptera Research | J HYMENOPT RES | Pensoft Publishers | 0022-1511 | Entomology | APC-GOA |
| Journal of the Kansas Entomological Society | J KANSAS ENTOMOL SOC | BioOne/Allen Press | 0022-8567 | Entomology | Hybrid/APC-GOA |
| Journal of the Lepidopterists Society | J LEPID SOC | BioOne/The Lepidopterists' Society | 0024-0966 | Entomology | Hybrid/APC-GOA |

**Table 1.** (*Continued.*)

| journal | abbreviation used in JCR | category | ISSN | publisher | publishing model |
|---|---|---|---|---|---|
| Journal of Mammalogy | J MAMMAL | Zoology | 0022-2372 | Oxford University Press | Hybrid/APC-GOA |
| Journal of Molluscan Studies | J MOLLUS STUD | Zoology | 0260-1230 | Oxford University Press | Hybrid/APC-GOA |
| Journal of Natural History | J NAT HIST | Zoology | 0022-2933 | Taylor & Francis Group | Hybrid/APC-GOA |
| Journal of Nematology | J NEMATOL | Zoology | 0022-300X | Exeley | DOA [?] |
| Journal of Systematic Palaeontology | J SYST PALAEONTOL | Palaeontology | 1477-2019 | Taylor & Francis Group | Hybrid/APC-GOA |
| Journal of Zoological Systematics and Evolutionary Research | J ZOOL SYST EVOL RES | Zoology | 0947-5745 | John Wiley & Sons | Hybrid/APC-GOA |
| Malacologia | MALACOLOGIA | Zoology | 0076-2997 | BioOne/Institute of Malacology | Hybrid/APC-GOA |
| Mammalia | MAMMALIA | Zoology | 1864-1547 | Walter de Gruyter GmbH | Hybrid/APC-GOA [?] |
| Molluscan Research | MOLLUSCAN RES | Zoology | 1323-5818 | Taylor & Francis Group | Hybrid/APC-GOA |
| Myrmecological News | MYRMECOL NEWS | Entomology | 1997-3500 | Austrian Society of Entomofaunistics | APC-GOA |
| Nauplius | NAUPLIUS | Zoology | 2358-2936 | BioOne/Brazilian Crustacean Society | DOA |
| Nautilus | NAUTILUS | Zoology | 0028-1344 | Bailey-Matthews National Shell Museum | DOA [?] |
| Nematology | NEMATOLOGY | Zoology | 1388-5545 | Brill Academic Publishers | Hybrid/APC-GOA |
| Neotropical Entomology | NEOTROP ENTOMOL | Entomology | 1519-566X | Springer Nature | Hybrid/APC-GOA |
| Neotropical Ichthyology | NEOTROP ICHTHYOL | Zoology | 1679-6225 | Sociedade Brasileira de Ictiologia/SciELO | APC-GOA |
| New Zealand Journal of Zoology | NEW ZEAL J ZOOL | Zoology | 0301-4223 | Taylor & Francis Group | Hybrid/APC-GOA |
| Nota Lepidopterologica | NOTA LEPIDOPTEROLOGI | Zoology | 0342-7536 | Pensoft Publishers | APC-GOA |
| New Zealand Entomologist | NZ ENTOMOL | Zoology | 0077-9962 | Taylor & Francis Group | Hybrid/APC-GOA |

**Table 1.** (Continued.)

| journal | abbreviation used in JCR | category | ISSN | publisher | publishing model |
|---|---|---|---|---|---|
| Odonatologica | ODONATOLOGICA | Entomology | 0375-0183 | Osmylus Scientific Publishers/International Odonatological Foundation, Societas Internationalis Odonatologica (S.I.O.) | Hybrid/APC-GOA |
| Organisms Diversity & Evolution | ORG DIVERS EVOL | Zoology | 1439-6092 | Springer Nature | Hybrid/APC-GOA |
| Oriental Insects | ORIENT INSECTS | Entomology | 0030-5316 | Taylor & Francis Group | Hybrid/APC-GOA |
| Proceedings of the Entomological Society of Washington | P ENTOMOL SOC WASH | Entomology | 0013-8797 | BioOne/Entomological Society of Washington | Hybrid/APC-GOA |
| Pacific Science | PAC SCI | Zoology | 0030-8870 | BioOne/University of Hawai'i Press | Hybrid/APC-GOA |
| Journal of Palaeontology | PALEONTOL J | Palaeontology | 0022-3360 | Cambridge University Press | Hybrid/APC-GOA |
| Pan-Pacific Entomologist | PAN PAC ENTOMOL | Entomology | 0031-0603 | BioOne/Pacific Coast Entomological Society | Hybrid/APC-GOA |
| Phyllomedusa | PHYLLOMEDUSA | Zoology | 1519-1397 | Esalq/USP | DOA |
| Primates | PRIMATES | Zoology | 0032-8332 | Springer Nature | Hybrid/APC-GOA |
| Records of the Australian Museum | REC AUST MUS | Zoology | 0067-1975 | Australian Museum | DOA [?] |
| Revista Brasileira de Entomologia | REV BRAS ENTOMOL | Entomology | 0085-5626 | Sociedade Brasileira de Entomologia/SciELO | APC-GOA |
| Revista Colombiana de Entomologia | REV COLOMB ENTOMOL | Entomology | 0120-0488 | Colombian Society of Entomology | APC-GOA |
| Revista de la Sociedad Entomológica Argentina | REV SOC ENTOMOL ARGE | Entomology | 0373-5680 | Sociedad Entomológica Argentina/Biotaxa/SciELO | DOA |
| Revue Suisse de Zoologie | REV SUISSE ZOOL | Zoology | 0035-418X | BioOne/Muséum d'histoire naturelle, Genève | DOA |
| Russian Journal of Herpetology | RUSS J HERPETOL | Zoology | 2713-1467 | Folium Publishing Company | Hybrid/APC-GOA |

(Continued.)

**Table 1.** (Continued.)

| journal | abbreviation used in JCR | category | ISSN | publisher | publishing model |
|---|---|---|---|---|---|
| Russian Journal of Nematology | RUSS J NEMATOL | Zoology | 0869-6918 | RUSSIAN ACAD SCI, INST PARASITOLOGY | Hybrid/APC-GOA |
| South American Journal of Herpetology | S AM J HERPETOL | Zoology | 1808-9798 | BioOne/Brazilian Society of Herpetology | DOA [?] |
| Salamandra | SALAMANDRA | Zoology | 0036-3375 | German Society for Herpetology and Herpetoculture | APC-GOA |
| Shilap-Revista de Lepidopterologia | SHILAP REV LEPIDOPT | Entomology | 0300-5267 | Sociedad Hispano-Luso-Americana de Lepidopterología España | DOA |
| Southwestern Entomologist | SOUTHWEST ENTOMOL | Entomology | 0147-1724 | BioOne/Society of Southwestern Entomologists | Hybrid/APC-GOA |
| Spixiana | SPIXIANA | Zoology | 341-8391 | Verlag Dr Friedrich Pfeil | Hybrid/APC-GOA |
| Studies on Neotropical Fauna and Environment | STUD NEOTROP FAUNA E | Zoology | 0165-0521 | Taylor & Francis Group | Hybrid/APC-GOA |
| Systematic and Applied Acarology | SYST APPL ACAROL UK | Entomology | 1326-1971 | BioOne/Systematic and Applied Acarology Society | Hybrid/APC-GOA |
| Systematic Entomology | SYST ENTOMOL | Entomology | 0307-6970 | John Wiley & Sons | Hybrid/APC-GOA |
| Transactions of the American Entomological Society | T AM ENTOMOL SOC | Entomology | 0002-8320 | BioOne/The American Entomological Society | Hybrid/APC-GOA |
| Turkish Journal of Zoology | TURK J ZOOL | Zoology | 1300-0179 | Scientific and Technological Research Council of Turkey | DOA [?] |
| Vertebrate Zoology | VERTEBR ZOOL | Zoology | 1864-5755 | Senckenberg Gesellschaft für Naturforschung | DOA |
| ZooKeys | ZOOKEYS | Zoology | 1313-2989 | Pensoft Publishers | APC-GOA |
| Zoologischer Anzeiger | ZOOL ANZ | Zoology | 0044-5231 | Elsevier B.V. | Hybrid/APC-GOA |
| Zoological Journal of the Linnean Society | ZOOL J LINN SOC LOND | Zoology | 0024-4082 | Oxford University Press | Hybrid/APC-GOA |
| Zoological Letters | ZOOL LETT | Zoology | 2056-306X | Springer Nature | APC-GOA |

(Continued.)

**Table 1.** (*Continued.*)

| journal | abbreviation used in JCR | category | ISSN | publisher | publishing model |
|---|---|---|---|---|---|
| *Zoology in the Middle East* | ZOOL MIDDLE EAST | Zoology | 0939-7140 | Taylor & Francis Group | Hybrid/APC-GOA |
| *Zoological Research* | ZOOL RES | Zoology | 2095-8137 | Chinese Academy of Sciences, and the China Zoological Society | DOA |
| *Zoological Science* | ZOOL SCI | Zoology | 0289-0003 | BioOne/Zoological Society of Japan | Hybrid/APC-GOA |
| *Zoologica Scripta* | ZOOL SCR | Zoology | 0300-3256 | John Wiley & Sons | Hybrid/APC-GOA |
| *Zoological Studies* | ZOOL STUD | Zoology | 1021-5506 | Biodiversity Research Center, Academia Sinica, Taiwan | DOA |
| *Zoologichesky Zhurnal* | ZOOL ZH [merged with Annales Zoologici Fennici] | Zoology | 0044-5134 | MAIK Nauka-Interperiodica PUBL | Unknown |
| *Zoologia* | ZOOLOGIA CURITIBA | Zoology | 1984-4670 | Pensoft Publishers | APC-GOA |
| *Zoology* | ZOOLOGY | Zoology | 0944-2006 | Elsevier B.V. | Hybrid/APC-GOA |
| *Zoosystematics and Evolution* | ZOOSYST EVOL | Zoology | 1435-1935 | Pensoft Publishers | DOA/APC-GOA |
| *Zootaxa* | ZOOTAXA | Zoology | 1175-5326 | Magnolia Press | Hybrid/APC-GOA |

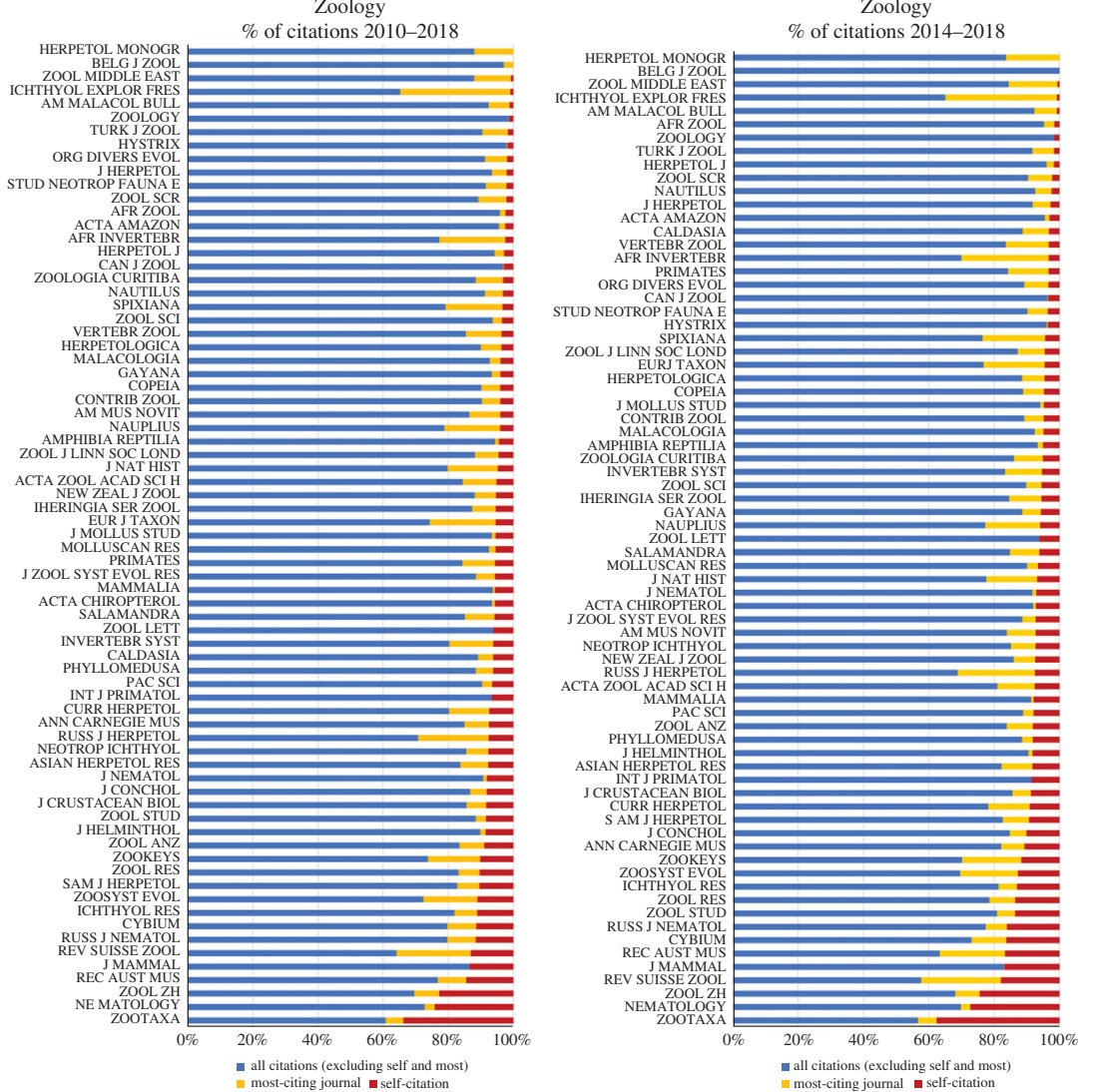

**Figure 2.** Percentage of citations received from all journals (blue), from most-citing journal (yellow), and self-citations (red) in the category Zoology from Journal Citation Reports (JCR) Science Edition database in Clarivate. The data of most-citing journal is from *Zootaxa*; except for *Zootaxa* where it is *ZooKeys*.

journal, *ZooKeys*, during 2010–2018 (electronic supplementary material, files 1, 3). Electronic supplementary material, file 3 shows the number of citations and journals with similar effects in the assessed metrics. Levels of self-citation are unrelated to number of citations.

# 4. Discussion

## 4.1. What are journal-level metrics?

As non-bibliometric researchers, we assume that measures in bibliometric science were created with some genuine purposes, which entail goals other than supposedly assessing the quality of research or researchers. Originally, these indexes aimed to be objective tools for helping librarians in the development of journal collections [9]. According to Keith Collier (Senior Vice President of Product, Science Group Clarivate, https://bit.ly/31gOMg3), the JIF mission is 'to provide a thorough, publisher-neutral, multifaceted view of journal performance, reflecting the world's highest-quality scientific and scholarly literature.' The hope for the consumers of JIF relies on the citation frequency that would reflect a journal value, and on the use made of it, and shows the average citations per published paper in a given journal (see [10]).

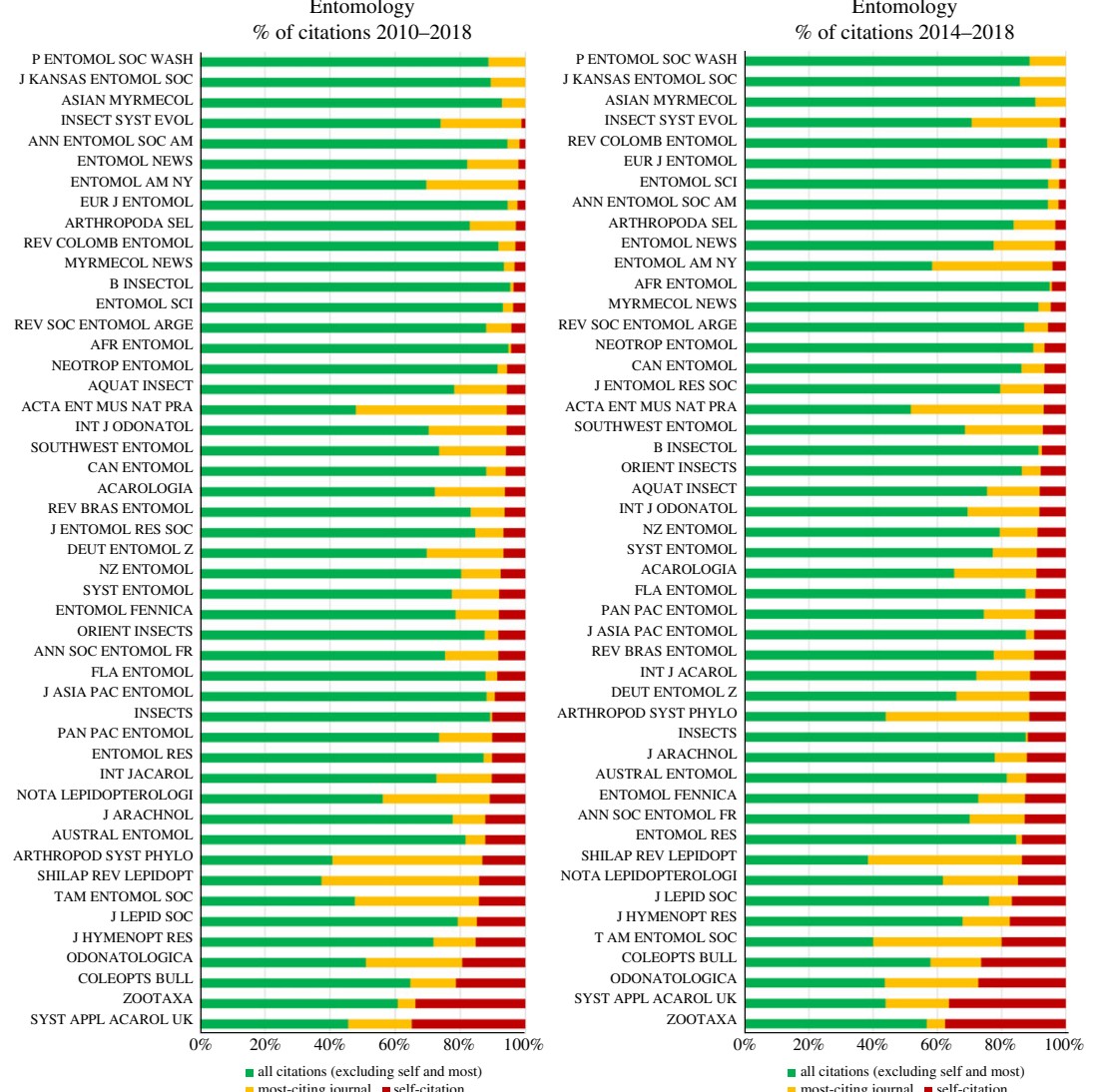

**Figure 3.** Percentage of citations received from all journals (green), from most-citing journal (yellow), and self-citations (red) in the category Entomology from Journal Citation Reports (JCR) Science Edition database of Clarivate. The data of most-citing journal are from *Zootaxa*; except for *Zootaxa* where it is *ZooKeys*.

Apart from the controversies of whether JIF actually assesses a journal 'quality', it aims, together with other bibliometric indexes, to highlight citation patterns and trends in publications. Since its creation in the beginning of the 1970s ([10], however, mentioned that it was designed in 1955; see also [11]), the metric became popular and has been adopted as a major parameter for evaluating the quality of research, a topic certainly controversial [12,13]. The index is a very simple measure calculated from the ratio between the number of citations along a year (numerator) and the number of papers published along the two previous years (denominator)—i.e. JIF 2019 is the number of citations in 2019 from papers published in 2017 and 2018 divided by the number of published papers in 2017 and 2018—[11]. So it shows how trendy papers or subjects published by a journal are, as well as if they are achieving a wide audience. The bad twist occurred when organizations, including governmental funding agencies, reached the conclusion that, since journals are evaluated by their citation impact, bingo, the scientific production in universities, institutes and graduate courses, as well as the researchers themselves, should be evaluated in the same manner. However, there is a flawed logic in extrapolating indexes such as JIF to evaluate work and careers. Hence, the JIF is recognized without doubt as being the most widely misused and abused bibliometric index in academic science [1,9,12].

The adoption of scientific bibliometric indexes such as JIF has grown, especially in the last two decades, as a way to evaluate the strongly competitive field of academic careers with numbers given to three decimal places that give a false impression of 'objectivity'. However, there are many studies

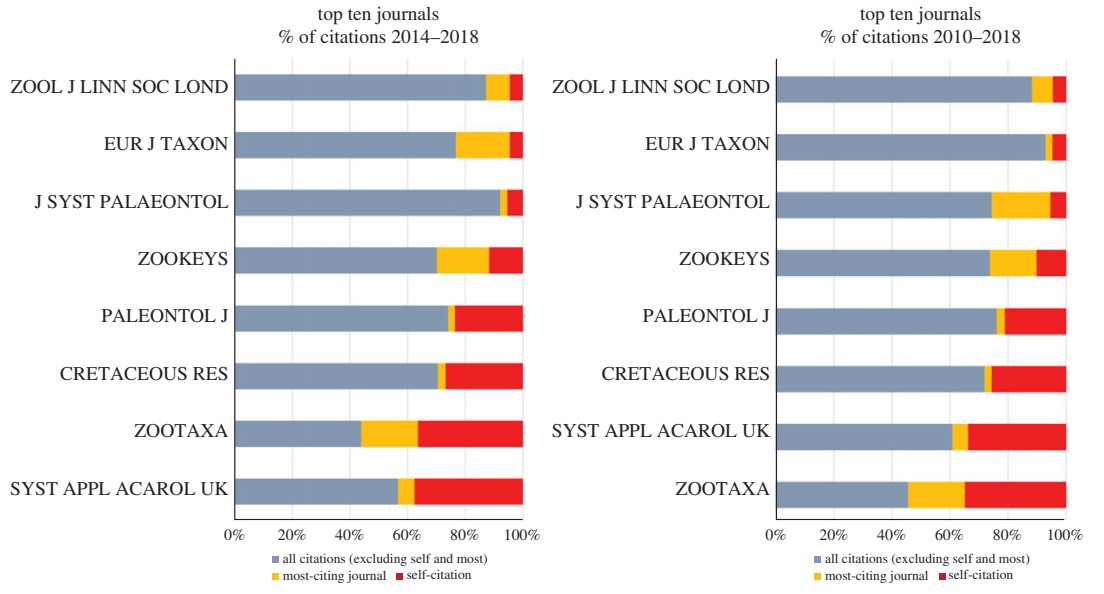

**Figure 4.** Percentage of citations received from all journals (greyish blue), from most-citing journal (yellow), and self-citations (red) for the top 10 zoological journals (TTJ, eight are on JCR) when the number of new available names is considered. Journal Citation Reports (JCR) Science Edition database of Clarivate. The data of most-citing journal are from *Zootaxa*; except for *Zootaxa* where it is *ZooKeys*.

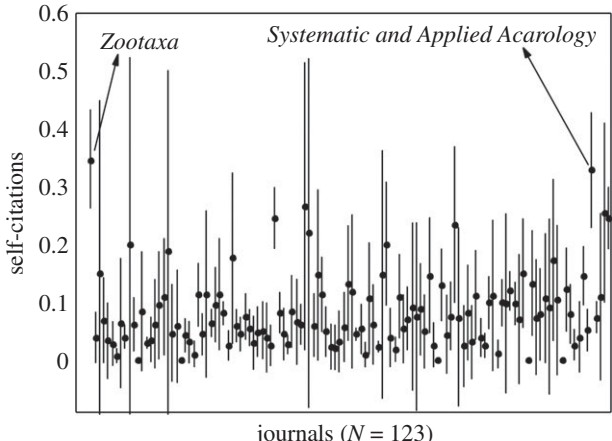

**Figure 5.** Mean percentage (dot) and standard deviation (line) of self-citations from 2010 to 2018 based on JCR/Clarivate 2019.

showing perverse pitfalls, for both researchers and organizations, of this use and interpretation of JIF (e.g. [12–14]). The quest and struggle for publishing in high impact journals produced the JIF mania [1].

Although the use of JIF is not recommended for ranking human beings [13], its impact in the real-life academic career is crystal clear. It is widely perceived that an academic researcher can only evolve in her/ his career by means of publishing in journals with high JIF values. The metric has well-known limitations when used to evaluate both journals or individual papers, because the index is strongly sensitive to what is considered a citable item [15]; also, it is characterized by a misuse of statistics by using the wrong measure of central tendency (mean rather than mode) given the typical skew in the distribution of citations across articles in a journal [16] and may be radically influenced by a single or few papers (e.g. [17]). Its widespread adoption leads to several distortions such as unjustified multi-authored papers and schemes by journals to artificially increase JIF or impact inflation; these schemes are among the most common outcomes of the JIF mania. Because of the metrics inflation, the bibliometric platforms act as judges to prevent these types of distortion, excluding or punishing 'deviant' journals. Indexing platforms such as Clarivate/JCR, for instance, adopt no less than 24 criteria into a putatively unreproducible method of analysis. When a journal disagrees or does not fulfil one of these criteria, it is suppressed [18]. The lack of transparency greatly affects our ability to properly evaluate journal

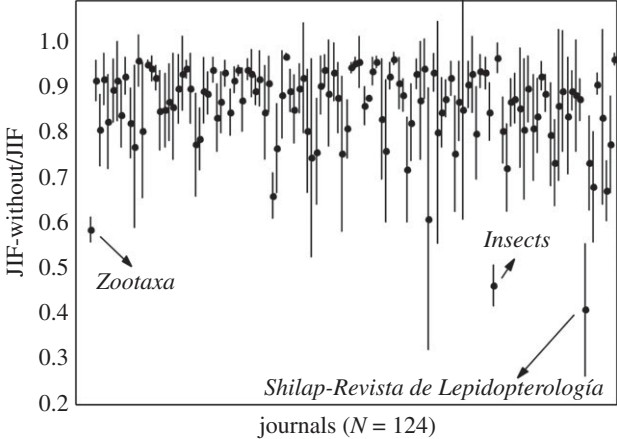

**Figure 6.** Mean of the ratio between JIF without self-citations and JIF (dot) and standard deviation (line) of journal impact factor (JIF) without self-citations from 2010 to 2018 based on JCR/Clarivate 2019.

suppressions. The philosophical dilemma 'Who watches the watchmen?', eternalized in the famous graphic novel *Watchmen*, written by Alan Moore, fits well here.

## 4.2. Metrics and taxonomy

Undeniably, exploring biodiversity is a core issue for the entire biological sciences (and humanity). In this context, taxonomic research is an essential priority in face of the current biodiversity crisis [19]. The concept of taxonomy in biological sciences has a wide range of meanings, varying from the reductionist, atomized and merely descriptive harmful view known as alpha taxonomy (e.g. [20])—largely denoted as a minor science, old fashioned, and intellectually poor—to a wide sense of taxonomy as the biggest among all biological sciences (e.g. [21]), equivalent to the whole field of comparative biology. This wider view, which is adopted here, embraces from primary data acquisition in field expeditions to morphological, genomic and even ecosystem analyses. Thus, it considers taxonomy as a relevant hypothesis-driven science. However, ordinary taxonomic research executed day by day is a generally low-cost activity that employs few technological tools. It is focused on the study of natural history collections with the goal of characterizing and making available basic data on biological entities. This work often involves the study of the morphology of poorly known taxa, an unknown sex of a given species or developmental stages, as well as undescribed taxa. Taxonomists must frequently work with poorly known subjects, looking for the novelty, odd, and thus dealing with unpopular or even neglected topics. Therefore, there are many cases of fine, well-written and beautifully fully illustrated comprehensive taxonomic monographs on animal groups that will probably rarely be cited. The small number of citations might even be related to the fact that such monographs successfully solve most of the basic taxonomic questions affecting one taxon. One colourful example was voiced by one of us (J.A.R.) during the Brazilian Congress of Zoology: 'I am studying one of the smallest orders of insects [Zoraptera, the angel insects] with no more than four dozens of extant species, so what is the chance of citation, within only two years, of a paper that provides a great contribution on this group, including the description of new species?' Problems with the low rate of citations in taxonomy are widely discussed and the inadequacy of JIF for the research assessment of basic sciences is often mentioned (e.g. [22,23]). This is a paradox caused by the fact that taxonomy must, in part, necessarily deal with basic descriptive subjects, the new and unexpected, focusing on small parts of the tree of life.

Taxonomy is currently considered a science in crisis affected by losses of positions in institutions and reduction of funding resources. In addition to this scenario of gradual loss of workforce and grants, the discipline is also damaged by the biases or inadequacies of these so-called indexes of 'quality' (see [24]). This situation is not exclusive to zoological taxonomy, Botany, for instance, has been suffering a drastic devaluing process [25]. Some solutions for low citations suggested the mandatory citation of references in which authorities erected new taxa (original descriptions) whenever a name was mentioned in a study, a rule endorsed by *Zootaxa* (https://bit.ly/34kXmgm) but not strictly enforced by the journal. This rule would partially explain its high level of self-citations. However, this strategy is deeply misleading,

because original descriptions, especially the old ones, are often not adequate for species characterization and recognition. A more straightforward approach would be to make clear which concept of species is being adopted and provide the bibliographic source (see [26]). Another important point is that multidisciplinarity in biological sciences has blurred the limits among traditional disciplines, even the descriptive ones. All these aspects were suggested as reasonable explanations for the high levels of *Zootaxa* self-citations. However, they are not valid because there are many other journals currently accepting taxonomic studies, being either purely descriptive or including broader analytical approaches; these journals are obviously attractive in the context of the JIF mania game (electronic supplementary material, file 1).

## 4.3. *Zootaxa* phenomenon and suppression quarrel

As authors of papers on distinct zoological taxa, editors of special issues, and reviewers of manuscripts submitted to *Zootaxa* during the last 10 years, we feel comfortable to offer an opinion on the journal and its impact in the taxonomic world, an actual phenomenon that transformed it into the leading vehicle for making new zoological names available.

Since its establishment, *Zootaxa* has become a prestigious forum for promotion and discussion of all topics of taxonomic science and thus reached a distinguished position among other similar journals. Unquestionably, the birth of the journal was a milestone to the field of zoological taxonomy. Started in 2001 with a hybrid platform of publication (i.e. the payment of Article Processing Charges, APC, by authors is optional for making the paper Open Access, OA), when 300 pages were published, the journal increased to 32 330 pages in 2010 [27] and ended 2019 with the impressive record of 47 528 pages; the latter comprising 2400 papers in 176 volumes (data compiled from *Zootaxa*'s site). In its first decade, *Zootaxa* has made available about 20–25% of the new nomina per year [28]. In the last 5 years, it has become the main journal, truly the leader in the field of descriptive taxonomy, with 24 722 (26.57% of the total) newly erected taxa made available [6]. Despite its few years of existence, the journal has received remarkable status and visibility among zoologists. Papers published in it have potentially higher chances of being cited by fellow taxonomists, unlike the situation in many other similar journals in the field that clearly have a lower visibility. *Zootaxa* has been the first choice for a legion of young taxonomists for their very first papers. The relatively high JIF of the journal is certainly among the reasons for this choice. Furthermore, for those zoologists who are not primarily taxonomists but who eventually decide to publish a taxonomic paper, the journal is also probably the first choice, if not the single one known. Indeed, *Zootaxa* is so influential nowadays that a somewhat pejorative term, '*Zootaxa* author,' has been coined, meaning those researchers who only publish in the journal or have a massive amount of their papers in it, reaching 80% or more. Why this phenomenon? Why does a journal congregate such a huge parcel of publications in a field? Is this situation actually good for taxonomy?

For almost a decade, *Zootaxa* was the single big (or mega) journal in the field designed to attend taxonomic science, even though several smaller journals also published most of their issues with a high amount of taxonomic papers. Today, *Zootaxa* has competitors with the advantage of having either Gold Open Access (GOA) or DOA policies, such as the *European Journal of Taxonomy* (first issue published in 2011) and *ZooKeys* (first issue in 2008). However, in the case of *ZooKeys*, a minimum APC of €700 is required for mandatory open access; this is a huge obstacle, especially for researchers from developing countries, outside the group of those countries considered of lowest income, who do not automatically qualify for a fee waiver. The *Zootaxa* initiative from Magnolia Press Ltd was so successful that it stimulated the creation of some new journals, including *Phytotaxa*, its sibling version dedicated to plant sciences. Data on Magnolia Press, which is based in New Zealand, is not easy to obtain; for instance, it is not clear to us whether it is a for-profit or not-for-profit organization.

The great significance of *Zootaxa* cannot be denied and it has become the most important vehicle for the publication of taxonomic studies. However, it is obviously not the single journal devoted to taxonomic science, such as depicted by some of the supporting letters. So, why has the suppression caused that enormous commotion? A quick answer is because in some megadiverse countries, such as Brazil in which most of the fauna remains undescribed, the higher education and scientific organizations evaluation systems have entirely embraced bibliometric indexes (e.g. [29–31]). Therefore, these metrics play an important role in the system and, for instance, a Brazil-based author's choice of a scientific journal is largely based on values such as JIF. Consequently, the suppression of *Zootaxa* was received as a serious setback for taxonomists in such countries, especially so of course for those who publish most or even all their papers in the journal. This last aspect has a clear influence on the

high rate of self-citations, as well as on the JIF of *Zootaxa* (figures 2–6), even considering that Clarivate recognized that 20% of papers on zoology were published by the journal.

## 4.4. What is self-citation and its consequences?

An important distinction should be made between two categories of self-citation, individual (author) and collective (journal) self-citation, although both potentially result in a boost of bibliometric indexes. There are many legitimate reasons for a researcher to cite her/his earlier works; in many cases, self-citations are unavoidable, depending on the circumstances or subject [5]. For example, an author could have been the single authority on a taxon during the last 30 years or present a high production in a specialized field. In these situations, self-citation alone is not necessarily fraudulent. Concerns arise when similar citations are not received in the work of other researchers in the field [9] or, more commonly, based on the myth that self-citations help to artificially increase one's own position in the community [5]. Differently, collective (journal) self-citation would be more problematic and is most probably a side-effect of the JIF mania, caused by the competition among journals for higher journal ranking, prestige and higher monetary earnings through higher subscription pricing, which is often connected to journal-level bibliometrics. Dear readers, do not be naive: academic publications are million-dollars businesses, truly having high profit margins [4,32]. It is thus not surprising that journals engage in 'impact factor wars' to manipulate their metrics using strategies such as citation stacking, enlargement of cited references during the review process to include papers from their own journal (sometimes even coercive self-citation) and rejection of studies with low potential of citation [9]. Thus, a high level of self-citations in a journal is not easy to understand and should be evaluated with caution.

Self-citation phenomena, either of author or journal types, have been deeply investigated from various perspectives, including sociological and bibliometric aspects. A review focused on author self-citation and all its technical nuances was presented by Szomszor *et al*. [33]. Generally, high levels of self-citation are condemned, particularly when journal self-citation is interpreted as the result of manipulation for boosting indexes; in these situations, it has of course been determined that the biased metrics should not be considered for analyses of influence or impact [1]. However, self-citation can be legitimate in certain circumstances [34]. Consequently, levels of self-citation are not easy to analyse. Ioannidis & Thombs [1] argued that these levels naturally vary, and high levels may be justifiable in highly specialized journals or in disciplines with few available journals.

*Zootaxa* hardly meets the aforementioned criteria for reasonable justification of high self-citations. Also, self-citation has increased in the journal over the years (electronic supplementary material, file 2). *Zootaxa* is clearly not highly specialized. A quick examination of its issues will confirm this point and taxonomy as a whole is far from having only a few other available journals, at least to most groups. We compiled 123 journals that publish taxonomic papers, solely in the JCR database (table 1). Therefore, there are clearly many options since these journals surely publish a great deal of descriptive taxonomy (figures 2 and 3). If specialization were true for *Zootaxa*, we would expect that more specialized journals devoted to small groups, such as *Odonatologica* (dragonflies) and *Acarologia* (mites), which together represent a small part of extant diversity, would present similar or even higher self-citation levels, which is not the case (figures 3 and 5). On the other hand, we would also expect that journals specialized in megadiverse groups, such as beetles, bees, moths, butterflies, spiders, etc. would likewise have high levels of self-citation, which again is not the case (figures 2, 3 and 5). Even journals dealing with taxa from a specific region of the world, such as *Neotropical Ichthyology* or *South American Journal of Herpetology*, also present significantly lower levels of self-citation. Therefore, the scope of *Zootaxa*, with its focus on taxonomy, does not explain the high level of self-citations. Instead, an explanation should be looked for in the elements of the *Zootaxa* phenomenon depicted above. A relevant aspect to be observed in this discussion is that the great majority of the citations given to the analysed journals came from *Zootaxa* (figures 2–4).

In addition, Chorus & Waltman [34] carefully studied journal self-citation and proposed a measure to evaluate boosts in the JIF, detecting disproportional and potentially unethical behaviour (Impact Factor Biased Self-Citation Practices). They did not consider their measure unfailing and discussed a few cases when self-citation would be legitimated. The latter include distinct situations. For instance, a researcher could be inspired by recent studies published in a journal and thus decides to conduct similar research; accordingly, that journal would naturally be an important source and her/his first choice for publication. Also, there are situations when, after finishing a manuscript, an author realizes that most of the cited references are from a given journal; the latter becomes again a naturally expected option. We believe

that such cases are strongly associated with the *Zootaxa* mega-journal phenomenon and appear to partially explain its high levels of self-citation.

We are confident that a journal can publish high-quality, robust science regardless of its level of self-citation. There is not necessarily any relationship between the rate of journal-level self-citation and the quality of the research published in a given journal, particularly in the case of high-output journals such as *Zootaxa*. Clarivate appears to want to promote a sense of competition among journals, so that it can sell its journal ranking data and analytics—clearly, zoological taxonomists and their publishing and citing behaviours do not fit the model that Clarivate seemingly wants to promote. Who is wrong here? The community of scientists producing taxonomic science for which they were specifically trained, or the profit-driven analytics company that appears to know nothing about taxonomy and yet still wants to rank and supposedly provide sound judgement on the quality of taxonomic journals? We think of course that the scientific community knows best, whereas Clarivate appears to know or indeed care very little about the robustness of science.

## 4.5. Is the suppression a new attack on taxonomy?

Based on the *Zootaxa* suppression and the academic engagement into a bandwagon sympathetic commotion, opinions in social media, and letters from societies and researchers (e.g. [35–37]), mainly from megadiverse countries, which appear to be in favour of the journal and ask Clarivate to review its decision, two main conclusions could be unearthed: (i) JIF appears very important to taxonomists and (ii) taxonomy is 'under attack'. We seriously doubt both conclusions and invite the reader to carefully consider these aspects.

Why do researchers choose to publish in *Zootaxa*? Several reasons influence the preference of a researcher for a specific journal. Certainly, scope, visibility, prestige in the field and JIF are among the most influential criteria. It is realistic to assume that most of the authors of *Zootaxa* are looking for a journal that has fast reviewing and production processes, is free of charge to authors (no APC), has a comparatively high JIF, and has no limit of pages for a manuscript. Authors and readers of the journal seem not to be concerned about the hybrid policy with paywall, with few published articles having open access, achieved through payment by authors (APC-OA). Among the reasons for this complaisance are the article-processing charges for most open access journals with values of hundreds of Euros or US Dollars, generally excessively expensive for researchers from developing countries, the possible economic situation of most contributors (e.g. Brazilian researchers are authors of most papers in the journal, https://bit.ly/2Y0hSQ9, [38]), and the open access is viable through platforms of self-archiving, such as ResearchGate, or websites, such as Sci-Hub; the latter illegally makes paywalled content available for free and is regarded as 'piracy'. High APC costs are clearly impeditive for researchers from most countries and for small research groups lacking big budgets. Also, there are certainly many other priorities for spending limited research money. Nevertheless, open access through platforms such as Sci-Hub is deprived of respect for the intellectual property or copyright laws and certainly raises many moral issues. Therefore, it is at least controversial that authors are opposed to paying fees to APC-GOA journals and are in favour of hybrid platforms because it is possible to break paywalls to access payment-based content.

The holy grail quest for diamond open access (no APC for authors, DOA) versus paywall policies creates a paradox: how can journals cover the considerable costs involved in publishing, copyediting, DOI generation, data insertion into biodiversity databases, file archiving, etc.? These controversies concerning OA were depicted with vibrant colours during the gradual transition of big publishers' journals, such as *Diversity and Distributions*, from readers' payment to authors' payment in an APC-OA model [39]. Gradually, the scientific scholarship publications are changing from paywall to GOA with authors paying the charges (APC-GOA) for publication in biodiversity journals. Certainly, this is the best business model option for the profit-seeking commercial publishers because it avoids losses generated by white (sometimes named black OA) or green platforms such as Sci-Hub, ResearchGate or Academia Inc. (site: academia.edu). Here, it is important to highlight that authors never received messages from *Zootaxa* demanding the removal of files from any such platforms, quite unlike the crusade carried out by big publishers against these kinds of storage and access-granting.

We are aware of the leading role that bibliometric indexes play in the science publishing industry, as well as their considerable influence on how and where science is done nowadays. However, JIF cannot determine the development of a whole scientific field, even when supporting agencies adopt it as a criterion of quality. A high JIF does not necessarily come from a high-quality taxonomic study; it is probably much more connected to the scope and diversity of methods and sources of data that are

usually portrayed by high JIF ranked journals. Another aspect to be considered is that *Zootaxa* was focused initially on long papers on descriptive taxonomy; subsequently, it gradually changed its scope and started accepting short notes and studies on various subjects associated with zoological taxonomy/systematics. Curiously, soon after Clarivate announced the reinstatement of the JIF of *Zootaxa* [40], the journal's website refreshed its JIF, showing perhaps that it is willingly taking part in the JIF games.

# 5. Conclusion

Menaces to taxonomy as a science come from distinct sources and the relatively low bibliometric indexes including citation rates of its journals is only one factor that contributes to establishing the so-called taxonomic impediment. The reversion of the suppression of *Zootaxa* by Clarivate is irrelevant to biological sciences and taxonomy because Journal Impact Factors are statistically illiterate [41] and cause a great deal of harm to science. This reinstatement should certainly not be regarded by taxonomists as a victory for the field. As a community we should not endorse the villainy of bibliometric policies that bring more harm than benefit to our field.

We hope the community of taxonomists gets engaged with renewed strength in actions directly connected to the development and promotion of our science. Instead of being deeply focused on gaming irreproducible journal metrics sold to our institutions and research funders, controlled by a USA/UK-based company, which itself was acquired in 2016 by two private equity funds (Onex Corporation and Baring Private Equity Asia—ONEX/BPEA; see [42], https://prn.to/31nGDYC, [43], https://bit.ly/3hm9yC4), we should perhaps concentrate, for instance, on securing professional positions for young talented taxonomists, who are much needed for the proper development and maintenance of museums, scientific collections and publicly accessible digital databases. We are sure that *Zootaxa* has provided an invaluable service to the field of taxonomy. Suppression from JIF will not change or diminish this remarkable contribution.

We emphasize that menace to taxonomy comes not much from the suppression of any specific journal from a bibliometric platform belonging to a big company. Much more harm is caused by the limited renewal of professional positions and the loss of collections, such as the huge ones that were housed at the Museu Nacional of the Federal University of Rio de Janeiro. These are the real issues that should motivate the engagement and action of taxonomists around the world. In short and loud, taxonomy is produced by taxonomists, not by journals. We recognize the deep impact the JIF mania has on the careers of taxonomists, due to governmental policies that embraced bibliometric evaluations in a highly competitive environment, with researchers struggling for limited grants. However, our current challenges cannot be dealt with through endorsement of the status quo. We need to change the focus. Also, it is contradictory to argue in favour of the reinstatement of *Zootaxa* to the JIF without considering that this journal has this index influenced by the currently high levels of self-citation. Regardless of its real significance, JIF is lamentably considered by many to be one of the attractive qualities of *Zootaxa*. An honourable choice would be to reject bibliometric indexes altogether, including JIF, instead of considering them when convenient. We are witnessing a moral bankruptcy of the system of scientific publications devoted to the knowledge on biodiversity; it would be much better if the system could be somehow reinvented with ways to support DOA as its main goal.

Taxonomic groups that still need massive descriptive studies, with many species waiting to be discovered, such as Coleoptera, Hymenoptera, Lepidoptera, Diptera and Arachnida, have many journals devoted specifically to them. The JIF of these journals is similar to that of *Zootaxa* and, of course, research on those taxa can also be published in more general outlets in Entomology or Zoology categories. Therefore, the high levels of self-citation in *Zootaxa* are hardly justifiable. It appears to us that these high levels are caused by a sociological bias, being a side effect of the *Zootaxa* phenomenon. Myths about *Zootaxa* as the unique journal that publishes taxonomic studies are clearly harmful to the field. In addition, an urgent question must be answered: if *Zootaxa* decides to ignore JIF altogether, would it remain a good vehicle for the publication of taxonomic papers? If your answer is no, there is certainly a big problem with the community of practitioners in the taxonomic world.

Data accessibility. Primary compiled data were arranged in electronic supplementary material, file 1.

Authors' contributions. Â.P.P. designed the study, compiled, organized and analysed the data, wrote the manuscript, revised and approved its final version; G.M., L.F.S. and R.M. wrote the manuscript, revised and approved its final version; L.M. and J.A.R. discussed concepts, revised and approved the final version.

Competing interests. The authors declare no competing interests and their organizations had no role in any steps of the development of this study, from its design to submission for publication.

Funding. This study was partially supported by grants from National Council for Scientific and Technological Development (CNPq) through research productivity fellowships to G.M. (proc. 303229/2018-7), J.A.R. (proc. 300019/2017-3), L.F.S. (proc. 308337/2019-0) and L.M. (proc. 308994/2018-3).

Acknowledgements. André Adrian Padial (DBot—UFPR) helped a great deal with the statistical analysis.

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
