## [Peer Review File · Royal Society Open Science]

Review History

RSOS-201617.R0 (Original submission)

Review form: Reviewer 1

Is the manuscript scientifically sound in its present form?

Yes

Are the interpretations and conclusions justified by the results?

Yes

Is the language acceptable?

No

Do you have any ethical concerns with this paper?

No

Have you any concerns about statistical analyses in this paper?

No

Recommendation?

Major revision is needed (please make suggestions in comments)

Comments to the Author(s)

Donat Agosti and others have published compelling arguments for open content of taxonomic information that might be considered in the conclusions (at authors' discretion). A paper by Valdecasas, Castroviejo and Marcus (Nature 403: 698) should be considered for citation.

The conclusions and observations in this manuscript are important and timely. I believe the paper can have a greater impact if it is tightened up and significantly shortened with details of analysis available as an online source.

One apparent disconnect exists in the concluding arguments. Clearly it is good advice to pay greater attention to employment for those doing taxonomy, but so long as administrators rely on citation impact in their hiring decisions it is not clear how to bridge this gap in thinking by institutional leaders. The answer is not immediately clear, for reasons explained well in the text about the growing reliance on indices.

Review form: Reviewer 2

Is the manuscript scientifically sound in its present form?

Yes

Are the interpretations and conclusions justified by the results?

Yes

Is the language acceptable?

Yes

Do you have any ethical concerns with this paper?

Yes

Have you any concerns about statistical analyses in this paper?

No

Recommendation?

Accept as is

Comments to the Author(s)

This manuscript really needs to be published as it address an important point, namely the under-valuing and targeting of taxonomy and taxonomists by publishers and companies such as Clarivate. Presently many journals that were once bastions of taxonomy now refuse to publish taxonomic papers because of low citation rates that lead to low impact factors. If journals such as Zootaxa go under because of fickle impact matrices then taxonomy as a whole will look unattractive to future taxonomists, particularly in this hyper-competitive world in which citation rates can decide your academic future. Taxonomy is struggling and judging taxonomic papers by citations may kill off the field completely.

This manuscript will create much needed discussion about the role of taxonomy and the journals that publish taxonomic papers. Publishing this manuscript is vital to getting this discussion started. The authors' statement "In short and loud, taxonomy is produced by taxonomists, not by journals" is so very true and should be mantra in the publishing world.

The manuscript is well written, argued and researched and should be published as is.

Review form: Reviewer 3 (Camilla Souto)

Is the manuscript scientifically sound in its present form?

Yes

Are the interpretations and conclusions justified by the results?

Yes

Is the language acceptable?

Yes

Do you have any ethical concerns with this paper?

No

Have you any concerns about statistical analyses in this paper?

No

Recommendation?

Accept with minor revision (please list in comments)

Comments to the Author(s)

The manuscript by Pinto et al. addresses the concept and misuses of journal metrics – specifically to the recent exclusion of Zootaxa, a well-regarded taxonomic journal, from JCR – and their impact on science. This is a valid contribution to the scientific community and it presents metrics data that support/contradict some of the ideas discussed in research forums online. I also applaud the gender-neutral language adopted by the authors. As the discussion addresses a controversial subject, however, it is even more imperative that the intended message is conveyed with clarity. Therefore, I strongly suggest the publication of this manuscript after moderate revision on its presentation. A detailed revision is attached (Appendices A & B).

Decision letter (RSOS-201617.R0)

Dear Dr Pinto

On behalf of the Editors, we are pleased to inform you that your Manuscript RSOS-201617 "Are publications on zoological taxonomy under attack?" has been accepted for publication in Royal Society Open Science subject to minor revision in accordance with the referees' reports. Please find the referees' comments along with any feedback from the Editors below my signature.

We invite you to respond to the comments and revise your manuscript. Below the referees' and Editors' comments (where applicable) we provide additional requirements. Final acceptance of

your manuscript is dependent on these requirements being met. We provide guidance below to help you prepare your revision.

Please submit your revised manuscript and required files (see below) no later than 7 days from today's (ie 11-Dec-2020) date. Note: the ScholarOne system will 'lock' if submission of the revision is attempted 7 or more days after the deadline. If you do not think you will be able to meet this deadline please contact the editorial office immediately.

on behalf of Jeffrey Thompson (Associate Editor) and Kevin Padian (Subject Editor)
openscience@royalsociety.org

Associate Editor Comments to Author (Jeffrey Thompson):

Dear Ângelo et al.

Following the reviews by three expert reviewers, one of which is very detailed, I am recommending publication following some revisions. All of the reviewers thought that this was an important contribution, and should be published, although with some additional modifications. The reviewers thought that some details of the analytical methods, namely clarifying details of journal choice, and analytical procedure, should be explained more clearly and in more detail. Additionally, one reviewer thought that the manuscript could be tightened up by including some of the figures into the supplement, and shortening it. I agree with the reviewers that the manuscript reads more like an opinion piece, and is a bit disconnected from the data. Working to more heavily include reference to the data in the text would help to improve the manuscript, perhaps by merging the results and discussion sections, as suggested by a reviewer. I suggest you to closely look at the review sheets provided by the reviewers, and address all of their concerns prior to resubmission.

All the best,
Jeff Thompson

Reviewer comments to Author:

Reviewer: 1

Comments to the Author(s)

Donat Agosti and others have published compelling arguments for open content of taxonomic information that might be considered in the conclusions (at authors' discretion). A paper by Valdecasas, Castroviejo and Marcus (Nature 403: 698) should be considered for citation.

The conclusions and observations in this manuscript are important and timely. I believe the paper can have a greater impact if it is tightened up and significantly shortened with details of analysis available as an online source.

One apparent disconnect exists in the concluding arguments. Clearly it is good advice to pay greater attention to employment for those doing taxonomy, but so long as administrators rely on citation impact in their hiring decisions it is not clear how to bridge this gap in thinking by institutional leaders. The answer is not immediately clear, for reasons explained well in the text about the growing reliance on indices.

Reviewer: 2

Comments to the Author(s)

This manuscript really needs to be published as it address an important point, namely the undervaluing and targeting of taxonomy and taxonomists by publishers and companies such as Clarivate. Presently many journals that were once bastions of taxonomy now refuse to publish taxonomic papers because of low citation rates that lead to low impact factors. If journals such as Zootaxa go under because of fickle impact matrices then taxonomy as a whole will look unattractive to future taxonomists, particularly in this hyper-competitive world in which citation rates can decide your academic future. Taxonomy is struggling and judging taxonomic papers by citations may kill off the field completely.

This manuscript will create much needed discussion about the role of taxonomy and the journals that publish taxonomic papers. Publishing this manuscript is vital to getting this discussion started. The authors' statement "In short and loud, taxonomy is produced by taxonomists, not by journals" is so very true and should be mantra in the publishing world.

The manuscript is well written, argued and researched and should be published as is.

Reviewer: 3

Comments to the Author(s)

The manuscript by Pinto et al. addresses the concept and misuses of journal metrics – specifically to the recent exclusion of Zootaxa, a well-regarded taxonomic journal, from JCR – and their impact on science. This is a valid contribution to the scientific community and it presents metrics data that support/contradict some of the ideas discussed in research forums online. I also applaud the gender-neutral language adopted by the authors. As the discussion addresses a controversial subject, however, it is even more imperative that the intended message is conveyed with clarity. Therefore, I strongly suggest the publication of this manuscript after moderate revision on its presentation. A detailed revision is attached.

===PREPARING YOUR MANUSCRIPT===

===PREPARING YOUR REVISION IN SCHOLARONE===

-- Ensure that your data access statement meets the requirements at <https://royalsociety.org/journals/authors/author-guidelines/#data>. You should ensure that you cite the dataset in your reference list. If you have deposited data etc in the Dryad repository, please only include the 'For publication' link at this stage. You should remove the 'For review' link.

-- If you have uploaded ESM files, please ensure you follow the guidance at <https://royalsociety.org/journals/authors/author-guidelines/#supplementary-material> to include a suitable title and informative caption. An example of appropriate titling and captioning may be found at https://figshare.com/articles/Table_S2_from_Is_there_a_trade-off_between_peak_performance_and_performance_breadth_across_temperatures_for_aerobic_scops_in_teleost_fishes_/3843624.

Author's Response to Decision Letter for (RSOS-201617.R0)

See Appendix C.

Decision letter (RSOS-201617.R1)

Dear Dr Pinto,

It is a pleasure to accept your manuscript entitled "Are publications on zoological taxonomy under attack?" in its current form for publication in Royal Society Open Science.

on behalf of Dr Jeffrey Thompson (Associate Editor) and Kevin Padian (Subject Editor)
openscience@royalsociety.org

Appendix A

Dear authors,

The manuscript by Pinto et al. addresses the concept and misuses of journal metrics — specifically to the recent exclusion of *Zootaxa*, a well-regarded taxonomic journal, from JCR — and their impact on science. This is a valid contribution to the scientific community and it presents metrics data that support/contradict some of the ideas discussed in research forums online. I also applaud the gender-neutral language adopted by the authors. As the discussion addresses a controversial subject, however, it is even more imperative that the intended message is conveyed with clarity. Therefore, I strongly suggest the publication of this manuscript after moderate revision on its presentation. The following issues should be addressed:

MAJOR POINTS

Introduction (p. 3, lines 44–47): *“Our goal is to discuss the following questions. Can the suppression of any journal from JIF really affect the production in the taxonomic field? What are the consequences of Zootaxa suppression to taxonomy as a science? What means self-citation? What is the average JIF for taxonomic journals? Is Zootaxa a victim of its success? So, what is actually going on?”* These questions are individually clear, but it is hard to gauge the authors’ main goal. As the questions are all interrelated, it would be more interesting if the authors also defined the overarching goal of the manuscript.

Materials and Methods: Methods lack clarity and it’s challenging to understand how the journals were chosen without first looking at the figures. Text should be improved to guarantee research replicability, especially the following:

(p. 2, lines 55–58): *We explored citation data including JIF, most-cited journals, and self-citation metrics from the Journal Citation Reports (Web of Science Core Collection™) of the last nine years (2010–2018) of the top ten zoological journals (TTJ, eight are included in JCR) when the number of new available names (based on the last five years of ION/Zoological Records™ – ZR) is considered.* — Sentence is confusing. A suggestion is to indicate upfront that citation data were collected from 123 journals from JCR and then explain how these journals were chosen. For example, it is not clear that only eight of the top ten journals were included. Another option is to include a simple flowchart.

(p. 2, lines 55–58): *We also checked up for journals focused on or regularly publishing taxonomic...* (p. 4, lines 1–2) *Among 168 journals in Zoology and 101 in Entomology, 73 and 48 (both numbers include “plus” Zootaxa) were selected.* — How did you define “regularly”? What were your criteria to select these journals?

(p. 3, line 14–15): *Thus, journals with similar scope and JIF to Zootaxa were considered. In practice, all journals publishing taxonomic papers with JIF 2019 ranging from 0.25–2.315 in the categories Zoology and Entomology were included.* — Were these the criteria to select journals? If yes, these sentences should be presented sooner. If not, what were these journals “considered” for and where were they “included”?

(p. 3, line 2): *Data of a total of 123 journals were compiled.* — Because the search result in overlapping journals (i.e. some of the top ten journals are also included in the list), it seems like these numbers do not add up. The reader will have to figure it out that: $73+48=121$, but *Zootaxa* is included twice, so total

is 120 + three of the top ten (actually eight of those) journals that are not overlapping because they are in the category Palaeontology = 123. SUGGESTION: mention the criteria to select journals in the Zoology and Entomology categories, and then that you also included the top ten journals (in terms of new named taxa) from other categories (i.e. not Entomology or Zoology) that are included in JCR.

(p. 3, line 2–3): *In order to analyze the selected journals with available data from 2010–2018, ...* (p. 3, lines 5–6) *This approach was conducted among the top ten journals (TTJ) in ZR and those in Zoology and Entomology categories.* — Didn't you compile data from 2010-2018 from all journals? Weren't the described analyses conducted in the whole dataset (i.e. 123 journals)? If not, please clarify.

Finally, data on the “publishing model” of journals were presented in the results and should be included in the methods.

Results/Discussion

The way the manuscript is organized, the results seem secondary and the discussion largely an opinion piece. To improve the robustness of the discussion, I suggest the authors merge Results & Discussion and present the results when they are pertinent to the discussion.

MINOR POINTS

I flagged a few typos in the manuscript, added some suggestions and highlighted the sentences that are mentioned below (annotated pdf file attached).

Summary

The only quantitative results presented in the summary refer to the high levels of self-citation in Zootaxa. I do not think that this favors the message that the authors want to convey with their manuscript. For example, the results that show that Zootaxa is the most-cited journal (i.e. *Zootaxa, with around 15,000 citations, received 311% more citations than the second most cited journal*) should be included.

(p. 1, lines 38–39): *Zootaxa shows higher levels of self-citations than similar journals. We consider Zootaxa's scope and the fact that it is a mega-journal, inadequate explanations. Instead, it is related to the “Zootaxa phenomenon”.* — Sentence is ambiguous; are “inadequate explanations” referring to the suppression of Zootaxa from JCR or to the fact that Zootaxa has high levels of self-citation?

(p. 1, lines 39–40): *Instead, it is related to the “Zootaxa phenomenon”...* Following up on the previous comment, what is “it is” referring to?

Introduction

(p. 2, lines 5–6): *An issue break through the media due to the involvement of zoologists from many countries...* — This sentence sounds odd as it seems that you are connecting the involvement of zoologists to the suppression of Zootaxa. If this is not the case, my suggestion is: “An issue broke through the media due to its effect on zoologists from many countries...”

(p. 2, lines 41–42): *In addition, some points are also potentially misplaced.* — Sentence is unclear for readers not up to date with the forum discussions.

Materials and Methods

(p. 2, lines 59–60): *...journal was suppressed from the current edition and will only reappear in September.* — September of which year? If you refer to 2020, is the data out?

Results/Discussion

(p. 3, lines 49–52): *In comparison to all other journals, excluding Systematic and Applied Acarology, the mean levels of self-citation are higher for Zootaxa than for any other journal in Zoology (Figure 1), Entomology (Figure 2), and TTJ categories (Figure 3), being 34.9% for 2010–2018 and 37.6% for 2014–2018 in Zootaxa.* — I suggest presenting this information before you present “*the upper bounds of self-citation in Entomology and Zoology*”. Highlights in the text indicate contradictory information.

(p. 5, lines 1–3): *Influence of self-citations on JIF comparing all 123 journals is almost insignificant to boost this metric because most journals from the three categories (Entomology, Zoology, and TTJ) have similar means of the ratio between JIF without self-citations and JIF for 2010–2018 (Figure 5)...* — Cause/consequence statement seems off. Do you mean that: most journals from the three categories (Entomology, Zoology, and TTJ) have similar means of the ratio between JIF without self-citations and JIF for 2010–2018 (Figure 5) indicating that influence of self-citations on JIF comparing all 123 journals is almost insignificant to boost this metric...?

(p. 8, lines 22–25): *We compiled 123 journals that publish taxonomic papers, solely in the JCR database (Table 1). Therefore, there are clearly many options since these journals surely publish a great deal of descriptive taxonomy (Figures 1–2).* — The list is indeed robust, but I am still not convinced that there are plenty of options to publish single species descriptions, especially of the many taxa that do not have specialized journals, free of charge. This is just an opinion (mine and of colleagues that work in taxa without specialized journals) and it’s at the discretion of the authors if they want to consider this scenario in their discussion.

Table 1: There is no way of knowing which journals are the TTJ. I suggest highlighting them in bold, or even placing them at the beginning of the list.

Figure 4: It would be more intuitive to read the graph if the x-axis were flipped (i.e. from 2010 on the left to 2018 on the right).

Figures 5 and 6: Shouldn’t it be 123 journals instead of 124?

Warm regards,

Camilla Souto, NMNH, 13.Nov.2020

Appendix B**ROYAL SOCIETY
OPEN SCIENCE****Are publications on zoological taxonomy under attack?**

Journal:	Royal Society Open Science
Manuscript ID	RSOS-201617
Article Type:	Research
Date Submitted by the Author:	09-Sep-2020
Complete List of Authors:	Pinto, Ângelo Parise; Federal University of Parana, Zoology; UFPR Mejdalani, Gabriel; Universidade Federal do Rio de Janeiro Museu Nacional, Entomology Mounce, Ross; Arcadia Fund Silveira, Luis; Universidade de São Paulo Museu de Zoologia Marinoni, Luciane; Federal University of Parana, Zoology Rafael, José ; INPA
Subject:	taxonomy and systematics < BIOLOGY, evolution < BIOLOGY, biogeography < CROSS-DISCIPLINARY SCIENCES
Keywords:	Bibliometrics, Biodiversity Crisis, Journal Impact Factor (JIF), Scientometrics, Systematics
Subject Category:	Organismal and Evolutionary Biology

Author-supplied statements

Relevant information will appear here if provided.

Ethics

Does your article include research that required ethical approval or permits?:

This article does not present research with ethical considerations

Statement (if applicable):

CUST_IF_YES_ETHICS :No data available.

Data

It is a condition of publication that data, code and materials supporting your paper are made publicly available. Does your paper present new data?:

Yes

Statement (if applicable):

Primary compiled data was arranged in Supplementary File 1 and for review purposes, it is embedded into the main file.

Conflict of interest

I/We declare we have no competing interests

Statement (if applicable):

CUST_STATE_CONFLICT :No data available.

Authors' contributions

This paper has multiple authors and our individual contributions were as below

Statement (if applicable):

APP designed the study, compiled, organized, and analyzed the data, wrote the manuscript, revised, and approved its final version; GM, LF, and RM wrote the manuscript, revised, and approved its final version; LM and JR discussed concepts, revised and approved the final version.

Are publications on zoological taxonomy under attack?

Ângelo Parise Pinto^{1*} - <https://orcid.org/0000-0002-1650-5666>, Gabriel Mejdalani² - <https://orcid.org/0000-0003-4513-243X>, Ross Mounce³ - <https://orcid.org/0000-0002-3520-2046>, Luís Fábio Silveira⁴ - <https://orcid.org/0000-0003-2576-7657>, Luciane Marinoni¹ - <https://orcid.org/0000-0001-7034-5395>, José Albertino Rafael⁵ - <https://orcid.org/0000-0002-0170-0514>

¹Departamento de Zoologia, Universidade Federal do Paraná, P. O. Box 19020, 81531-980, Curitiba, PR, Brazil; ²Departamento de Entomologia, Museu Nacional, Universidade Federal do Rio de Janeiro, Rio de Janeiro, RJ, Brazil; ³Arcadia Fund, Sixth Floor, 5 Young Street, London W8 6EH, UK; ⁴Museu de Zoologia da Universidade de São Paulo, São Paulo, SP, Brazil; ⁵Instituto Nacional de Pesquisas da Amazônia, Manaus, AM, Brazil; *corresponding author: appinto@ufpr.br

Keywords: Bibliometrics, Biodiversity Crisis, Journal Impact Factor (JIF), Scientometrics, Systematics.

1. Summary

Taxonomy is essential to biological sciences and the priority field in face of the biodiversity crisis. The industry of scientific publications has made extensive use of bibliometric indexes, resulting in side effects such as the Journal Impact Factor (JIF) mania. Inadequacies of the widely used indexes to assess taxonomic publications are among the impediments for the progress of this field. Based on an unusual high proportion of self-citations the mega-journal *Zootaxa*, focused on zoological taxonomy, was suppressed from the Journal Citation Reports (JCR, Clarivate™). A prompt reaction from the scientific community against this decision took place exposing myths and misuses of bibliometrics. Our goal is to shed light on the impacts of bibliometrics to the production in taxonomy. We explored JCR's metrics for 2010–2018 of 123 zoological journals publishing taxonomic studies. *Zootaxa* shows higher levels of self-citations than similar journals. We consider *Zootaxa's* scope and the fact that it is a mega-journal, **inadequate explanations**. Instead, **it is** related to the “Zootaxa phenomenon”, a sociological bias that includes visibility and potentially harmful misconceptions that portray the journal as the only one that publishes taxonomic studies. Menaces to taxonomy come from many sources and the low bibliometric indexes is only one factor that contributes with its impediments. Taxonomists instead of being focused on journal metrics endorsing the villainy of policies imposed by dominant companies, should be engaged with renewed strength in actions directly connected to the promotion of this science.

2. Introduction

Every middle of year giant companies on scientific data analytics, the American-British Clarivate™ (InCites™) and the Dutch RELX™ Elsevier B.V. (Scopus®), release their metrics for scientific journals indexed in their huge databases, among them the Journal Impact Factor™ (JIF) and the CiteScore™, respectively. These metrics have been adopted as major qualifiers by several countries as a single measure of the quality of the produced research in their universities and institutes. Generally, funding for research in these institutions is derived from the taxes paid by the citizens of a given country. This policy produces a sort of quest or JIF mania for

*Author for correspondence (appinto@ufpr.br).

†Present address: Departamento de Zoologia, Universidade Federal do Paraná, P. O. Box 19020, 81531-980, Curitiba, PR, Brazil

publishing in higher-ranked journals (Ioannidis & Thoms 2019). Therefore, depending on the impact factor, a researcher has better chances of evolving in his/her career, earn prestige, win grants, etc. Thus, these metrics have a strong impact on how and what scientific investigation can currently be conducted.

On the last day of June of this year, an interruption of the colossal concerns about the Covid-19 pandemic affected taxonomists around the world. An issue break through the media due to the involvement of zoologists from many countries: the suppression of the mega-journal *Zootaxa*, a periodical focused on zoological taxonomy, from the Journal Citation Reports™ (JCR) Science Edition metrics by Clarivate. Based on a high proportion of self-citations, along with other 32 journals from the 12,000 in JCR database, *Zootaxa* would not receive a value of Journal Impact Factor (JIF) for 2019; however, it would keep the values for previous years and still be indexed on the Clarivate Analytics platform.

By this time, with the publication of JIF 2019 by JCR, which called the attention of editors and authors who were eager to see how journals were ranked, passionate discussions arose because of *Zootaxa*'s suppression. A prompt reaction, hardly seen before, through many letters of support to *Zootaxa* and petitions from several societies and researchers, forced Clarivate to review its decision. We believe that suppression of *Zootaxa* entails so many unique elements that it needs a closer inspection. Some supporting letters could actually be considered political manifestos and others were very naïve, not to say alarmist or simply inaccurate in interpreting the fact as a new attack to taxonomy as a science. Among the utterly passionate arguments was the one that *Zootaxa* is the single vehicle to publish taxonomic papers nowadays, a statement obviously far away from the truth. At the end of July, in a short statement on Twitter, Clarivate announced that *Zootaxa* and the *International Journal of Systematic and Evolutionary Microbiology* would be reconsidered in the regular refresh of the JCR to be published in September. Cases of suppression are common and not unique to the Clarivate platform, being most of them due to accusations of artificial boost or inflation of impact factors (e.g., Cortegiani et al. 2020). A particular case, almost a decade ago, had a wide repercussion among researchers when four journals edited in Brazil were suppressed under accusation of a citation-stacking scheme, a sort of cartel in which self-citations are exchanged among a group of journals (Van Norden 2013). Noticeably, cases of suppression in the past hardly received any sympathy from the scientific community, except from people directly involved as editors, perhaps as a signal that sectors of the academic community agreed with the suppression and considered that the affected journals deserved such "punishment." Once discarded from JIF, a journal is excluded from the gold rush of academia targeting high-impact outlets.

In a system full of anachronisms, in which traditional journals supported by museums or scientific societies are struggling to survive and the scientific publishing industry is led by giant publishers such as John Wiley & Sons, Elsevier, and Springer Nature, among others with profit margins comparable to those of major players in drug, bank, and auto companies (Larivière et al. 2015), it is at least curious to perceive the commotion around the suppression of *Zootaxa*. We became intrigued and thus decided to provide some reflections aiming to shed light on underlying aspects of this issue. We believe that many of the arguments that were given in the supporting letters are based on misunderstandings about these metrics or are biased by personal interests due to the pressure to publish in high-impact journals. In addition, some points are also potentially misplaced. Bibliometric data are plagued by myths and misunderstandings (Glänzel 2008).

Our goal is to discuss the following questions. Can the suppression of any journal from JIF really affect the production in the taxonomic field? What are the consequences of *Zootaxa* suppression to taxonomy as a science? What means self-citation? What is the average JIF for taxonomic journals? Is *Zootaxa* a victim of its success? So, what is actually going on? To properly address these aspects, we first need to clarify a few concepts and dig further into the current situation of taxonomic journals, impact measures of scientific publications, and the role of individuals and mega-corporations in this arena.

3. Materials and Methods

We explored citation data including JIF, most-cited journals, and self-citation metrics from the Journal Citation Reports (Web of Science Core Collection™) of the last nine years (2010–2018) of the top ten zoological journals (TTJ), eight are included in JCR) when the number of new available names (based on the last five years of ION/Zoological Records™ – ZR) is considered. We also checked up for journals focused on or regularly publishing taxonomic papers included in the Zoology and Entomology categories. Data from *Zootaxa* were retrieved from JCR 2018 because this journal was suppressed from the current edition and will only reappear in September. Journals included in both Zoology and Entomology categories were considered simply as

[revised manuscript text omitted]

Ethical Statement

Not applicable.

Funding Statement

This study was partially supported by grants from National Council for Scientific and Technological Development (CNPq) through research productivity fellowships to GM (proc. 303229/2018-7), JAR (proc. 300019/2017-3), LF (proc. 308337/2019-0), and LM (proc. 308994/2018-3).

Data Accessibility

Primary compiled data was arranged in Supplementary File 1.

Competing Interests

The authors declare no competing interests and their organizations had no role in any steps of the development of this study, from its design to submission for publication

Authors' Contributions

APP designed the study, compiled, organized, and analyzed the data, wrote the manuscript, revised, and approved its final version; GM, LF, and RM wrote the manuscript, revised, and approved its final version; LM and JR revised and approved the final version.

References

- Alberts B (2013) Impact Factor Distortions. *Science* 340 (6134): 787. DOI: <http://dx.doi.org/10.1126/science.1240319>
- Baring Private Equity Asia - BPEA (2019) Baring Private Equity Asia and Onex Partners announce secondary offering of Clarivate Analytics. December 4, 2019. Access August 26, 2020. Available at: <https://www.bpeasia.com/news/191204-joint-press-release-on-clarivate-analytics/>
- Brazilian Society of Herpetology (SBH) (2020) Letter to Clarivate. <http://sbherpetologia.org.br/assets//Documentos/C%C3%B3pia%20de%20SBH%20Zootaxa.docx>
- Chapman CA, Bicca-Marques JG, Calvignac-Spencer S, Fan P, Fashing PJ, Gogarten J, Guo S, Hemingway CA, Leendertz F, Li B, Matsuda I, Hou R, Serio-Silva JC, Stenseth NC (2019) Games academics play and their consequences: how authorship, h-index and journal impact factors are shaping the future of academia. *Proceedings of the Royal Society B, Biological Sciences* 286: 20192047. <http://dx.doi.org/10.1098/rspb.2019.2047>
- Chorus C, Waltman L (2016) A Large-Scale Analysis of Impact Factor Biased Journal Self-Citations. *PLoS ONE* 11(8): e0161021. <https://doi.org/10.1371/journal.pone.0161021>
- Clarivate Analytics (2020a) Web of Science Journal Citation Reports: Suppression Policy. Access August 15, 2020. Available at: <http://help.incites.clarivate.com/incitesLive/ICR/ICRGroup/titleSuppressions.html>
- Clarivate Analytics (2020b) Statement on suppression. Clarivate Web of Science @webofscience on Twitter. July 28, 2020, 10:24 a.m. Available at: <https://twitter.com/webofscience/status/1288103038121648128?s=08>
- Cortegiani A, Ippolito M, Ingoglia G, Manca A, Cugusi L, Severin A, Strinzel M, Panzarella V, Campisi G, Manoj L, Gregoretti C, Einav S, Mohe D, Gregoretti C. (2020) Inflated citations and metrics of journals discontinued from Scopus for publication concerns: the GhoS(t)copus Project. *F1000Research* 9(415), 415. <https://doi.org/10.12688/f1000research.23847.1>
- Cision Ltd./PR Newswire (2016) Acquisition of the Thomson Reuters Intellectual Property and Science Business by Onex and Baring Asia completed. October 3, 2016. Access August 20, 2020. Available at: <https://www.prnewswire.com/news-releases/acquisition-of-the-thomson-reuters-intellectual-property-and-science-business-by-onex-and-baring-asia-completed-300337402.html>
- Curry S (2012) Sick of Impact Factors. *Occam's Typewriter*. August 13, 2012. Access: August 30, 2020. Available at: <http://occamstypewriter.org/scurry/2012/08/13/sick-of-impact-factors/>
- Curry S (2018) Let's move beyond the rhetoric: it's time to change how we judge research. *Nature*: 554: 147. <https://doi.org/10.1038/d41586-018-01642-w>
- Dalgaard P (2008) *Introductory Statistics with R (Second Edition)*. Springer, New York, 363 p. <https://doi.org/10.1007/978-0-387-79054-1>
- Dimitrov J, Kaveri S, Bayry J (2010) Metrics: journal's impact factor skewed by a single paper. *Nature* 466, 179 (2010). <https://doi.org/10.1038/466179b>
- Ebach MC, Valdecasas AG, Wheeler QD (2011) Impediments to taxonomy and users of taxonomy: accessibility and impact evaluation. *Cladistics* 27(5): 550–557. <https://doi.org/10.1111/j.1096-0031.2011.00348.x>
- Garfield E (1972) Analysis as a Tool in Journal Evaluation. *Science* 178(4060): 471–479. <http://www.jstor.org/stable/1735096>
- Garfield E (2006) The History and Meaning of the Journal Impact Factor. *JAMA* 295(1): 90. doi: <https://doi.org/10.1001/jama.295.1.90>.
- Glänzel W (2008) Seven Myths in Bibliometrics. About facts and fiction in quantitative science studies. *COLLNET Journal of Scientometrics and Information Management* 2(1): 9–17. <https://doi.org/10.1080/09737766.2008.10700836>
- Haustein S, Larivière V (2015) The Use of Bibliometrics for Assessing Research: Possibilities, Limitations and Adverse Effects. In: Welpel I, Wollersheim J, Ringelhan S, Osterloh M (eds) *Incentives and Performance*. Springer, Cham. https://doi.org/10.1007/978-3-319-09785-5_8
- Hecht F, Hecht BK, Sandberg AA (1998) The journal "impact factor": a misnamed, misleading, misused measure. *Cancer Genetics and Cytogenetics* 104(2): 77–81. [https://doi.org/10.1016/S0165-4608\(97\)00459-7](https://doi.org/10.1016/S0165-4608(97)00459-7)

- Ioannidis JPA, Thombbs BD (2019) A user's guide to inflated and manipulated impact factors. *European Journal of Clinical Investigation* 49(9): e13151 [1–6]. <https://doi.org/10.1111/eci.13151>
- Index to Organism Names (ION)/Zoological Records (2020) Top systematics journals. Access August 26, 2020. Available at: <http://www.organismnames.com/metrics.htm?page=tsj>
- Krell FT (2000) Impact factors aren't relevant to taxonomy. *Nature* 405: 507–508. <https://doi.org/10.1038/35014664>
- Krüger AK (2020) Quantification 2.0? Bibliometric Infrastructures in Academic Evaluation. *Politics and Governance* 8(2): 58–67. <https://www.doi.org/10.17645/pag.v8i2.2575>
- Larivière V, Haustein S, Mongeon P (2015) The Oligopoly of Academic Publishers in the Digital Era. *PLoS ONE* 10(6): e0127502 [1–15]. <https://doi.org/10.1371/journal.pone.0127502>
- Mayr E (1969) *Principles of systematic zoology*. New York, McGraw-Hill.
- Meier R (2017) Citation of taxonomic publications: the why, when, what and what not. *Systematic Entomology* 42(2): 301–304. <https://doi.org/10.1111/syen.12215>
- Monbiot G (2011) Academic publishers make Murdoch look like a socialist. *The Guardian*, Monday 29 August 2011. [A fully referenced version available at: <https://www.monbiot.com/2011/08/29/the-lairds-of-learning/>]
- Peterson AT, Anderson RP, Beger M, Bolliger J, Brotons L, Burrige CP, Cobos ME, Cuervo-Robayo AP, Di Minin E, Diez J, Elith J, Embling CB, Escobar LE, Essl F, Feeley KJ, Hawkes L, Jiménez-García D, Jimenez L, Green DM, Knop E, Kühn I, Lahoz-Monfort JJ, Lira-Noriega A, Lobo JM, Loyola R, Nally RM, Machado-Strede F, Martínez-Meyer E, McCarthy M, Merow C, Nori J, Nuñez-Penichet V, Osorio-Olvera L, Pyšek P, Rejmánek M, Ricciardi A, Robertson M, Soto OR, Romero-Alvarez D, Roura-Pascual N, Santini L, Schoeman DS, Schröder B, Soberon J, Strubbe D, Thuiller W, Traveset A, Trembl EA, Václavík T, Varela S, Watson JEM, Wiersma Y, Wintle B, Yañez-Arenas C, Zure D (2019) Open access solutions for biodiversity journals: Do not replace one problem with another. *Diversity and Distributions* 25(1): 5–8. <https://doi.org/10.1111/ddi.12888>
- Rafael JA, Aguiar AP, Amorim DS (2009) Knowledge of insect diversity in Brazil: challenges and advances. *Neotropical Entomology* 38(5): 565–570.
- R Core Team (2018) [version 3.5.0] R: A Language and Environment for Statistical Computing. R Foundation for Statistical Computing. <https://www.R-project.org>
- Reategui E, Pires A, Carniato M, Franco SRK (2020) Evaluation of Brazilian research output in education: confronting international and national contexts. *Scientometrics* <https://doi.org/10.1007/s11192-020-03617-z>
- Sociedad de Odonatología Latinoamericana - SOL (2020) Letter to Marian Hollingsworth Clarivate Analytic. July 3, 2020. Access August 20, 2020. Available at: <http://www.odonatasol.org/wp-content/uploads/2020/07/SOL-support-for-ZOOTAXA-1.pdf>
- Szomszor M, Pendlebury DA, Adams J (2020) How much is too much? The difference between research influence and self-citation excess. *Scientometrics* 123: 1119–1147. <https://doi.org/10.1007/s11192-020-03417-5>
- The PLoS Medicine Editors (2006) The Impact Factor Game. *PLoS Med* 3(6): e291. <https://doi.org/10.1371/journal.pmed.0030291>
- Van Damme K (2020) Zootaxa Suppressed from 2019 JCR Data (2020 release). July 16, 2020. Access August 20, 2020. Available at: https://www.gopetition.com/petitions/zootaxa-suppressed-from-2019-jcr-data-2020-release.html?fbclid=IwAR3dL0dntvkIw_xBdx00rB7Os9e6y-NvRCC_i8WMvo2I0KJJEa1VXuMD0P8
- Van Norden R (2013) Brazilian citation scheme ousted. Thomson Reuters suspends journals from its rankings for 'citation stacking'. *Nature* 500(7464), 510–511. <https://doi.org/10.1038/500510a>
- Vanclay JK (2012) Impact factor: outdated artefact or stepping-stone to journal certification? *Scientometrics* 92: 211–238. <https://doi.org/10.1007/s11192-011-0561-0>
- Wheeler B, Torchiano M (2016) *lmPerm: Permutation Tests for Linear Models*. Available at: <https://github.com/mtorchiano/lmPerm>
- Wheeler QD (2008) Introductory: Toward the New Taxonomy. In Q D Wheeler (ed) *The New Taxonomy*, Systematics Association Special Volume Series, Boca Raton, CRC Press.
- Wheeler QD, Knapp S, Stevenson DW, Stevenson J, Blum SD, Boom BM, Borisy GG, Buizer JL, De Carvalho MR, Cibrian A, Donoghue MJ, Doyle V, Gerson EM, Graham CH, Graves P, Graves SJ, Guralnick RP, Hamilton AL, Hanken J, Law W, Lipscomb DL, Lovejoy TE, Miller H, Miller JS, Naeem S, Novacek MJ, Page LM, Platnick NI, Porter-Morgan H, Raven PH, Solis MA, Valdecasas AG, Van Der Leeuw S, Vasco A, Vermeulen N, Vogel J, Walls RL, Wilson EO, Woolley JB (2012) Mapping the biosphere: exploring species

to understand the origin, organization and sustainability of biodiversity. *Systematics and Biodiversity*

10(1): 1–20. <https://doi.org/10.1080/14772000.2012.665095>

Zhang Z-Q (2011) Accelerating biodiversity descriptions and transforming taxonomic publishing: the first
decade of *Zootaxa*. *Zootaxa* 2896: 1–7.

Zhang Z-Q (2014) Sustaining the development of world's foremost journal in biodiversity discovery and
inventory: *Zootaxa* editors and their contributions. *Zootaxa* 3753 (6): 597–600.

<http://dx.doi.org/10.11646/zootaxa.3753.6.6>

Figure legends

Figure 1. Amount of citations including all journals (blue), from most-citing journal (yellow), and self-citations (red) in the category Zoology from Journal Citation Reports (JCR) Science Edition database in Clarivate. * The data of most-citing journal is from *Zootaxa*; except for *Zootaxa* where it is *ZooKeys*.

Figure 2. Amount of citations including all journals (green), from most-citing journal (yellow), and self-citations (red) in the category Entomology from Journal Citation Reports (JCR) Science Edition database of Clarivate. * The data of most-citing journal is from *Zootaxa*; except for *Zootaxa* where it is *ZooKeys*.

Figure 3. Amount of citations including all journals (blue), from most-citing journal (yellow), and self-citations (red) for the top ten zoological journals (TTJ, eight are on JCR) when the number of new available names is considered. Journal Citation Reports (JCR) Science Edition database of Clarivate. *The data of most-citing journal is from *Zootaxa*; except for *Zootaxa* where it is *ZooKeys*.

Figure 4. Evolution of percentage of journal-level self-citation in *Zootaxa* based on JCR/Clarivate 2019.

Figure 5. Mean percentage (dot) and standard deviation (line) of self-citations from 2010 to 2018 based on JCR/Clarivate 2019.

Figure 6. Mean of the ratio between JIF without self-citations and JIF (dot) and standard deviation (line) of journal impact factor (JIF) without self-citations from 2010 to 2018 based on JCR/Clarivate 2019.

Figure 7. Number of citations based on JCR/Clarivate 2019 and notations of observed effects on bibliometric measures compared to *Zootaxa*.

Tables, Figure captions and graphs

Table 1. Journals, and their publishing model, indexed in Journal Citation Reports (Web of Science Core Collection™) and that publish taxonomic studies included in the Zoology and Entomology categories plus the top ten zoological journals (TTJs) in number of new taxa in the last five years based on the Zoological Records. APC-GOA = gold open access through payment of article processing charges; DOA = diamond open access; GOA = gold open access; Hybrid = optional payment of gold open access, access to the content via subscription (paywall).

Journal	Abbreviation used in JCR	Category	ISSN	Publisher	Publishing model
Acarologia	ACAROLOGIA	Entomology	0044-586X	Centre de Biologie pour la Gestion des Populations, France	DOA
Acta Amazonica	ACTA AMAZON	Zoology	0044-5967	INPA/SciELO	DOA
Acta Chiropterologica	ACTA CHIROPTEROL	Zoology	1508-1109	Acta Chiropterologica, published by the Museum and Institute of Zoology at the Polish Academy of Sciences, is devoted solely to the study and discussion of bats.	DOA
Acta Entomologica Musei Nationalis Pragae	ACTA ENT MUS NAT PRA	Entomology	1804-6487	BioOne/ Museum and Institute of Zoology, Polish Academy of Sciences	Hybrid/APC-GOA [?]
Acta Zoologica Academiae Scientiarum Hungaricae	ACTA ZOOL ACAD SCI H	Zoology	1217-8837	Hungarian Academy of Sciences	DOA
African Entomology	AFR ENTOMOL	Entomology	1021-3589	BioOne/Entomological Society of Southern Africa	Hybrid/APC-GOA
African Invertebrates	AFR INVERTEBR	Zoology	1681-5556	Pensoft Publishers	APC-GOA
African Zoology	AFR ZOOL	Zoology	1562-7020	Taylor & Francis Group	Hybrid/APC-GOA

Table 1. Continued.

Journal	Abbreviation used in JCR	Category	ISSN	Publisher	Publishing model
American Malacological Bulletin	AM MALACOL BULL	Zoology	0740-2783	The Sheridan Press	Hybrid/AP C-GOA
American Museum Novitates	AM MUS NOVIT	Zoology	0003-0082	BioOne/American Museum of Natural History	APC-GOA
Amphibia-Reptilia	AMPHIBIA REPTILIA	Zoology	0173-5373	Brill Academic Publishers	Hybrid/AP C-GOA
Annals of Carnegie Museum	ANN CARNEGIE MUS	Zoology	0097-4463	BioOne/Carnegie Museum	Hybrid/AP C-GOA [?]
Annals of the Entomological Society of America	ANN ENTOMOL SOC AM	Entomology	0013-8746	Oxford University Press	Hybrid/AP C-GOA
Annales de la Societe Entomologique de France	ANN SOC ENTOMOL FR	Entomology	0037-9271	Taylor & Francis Group	Hybrid/AP C-GOA
Aquatic Insects	AQUAT INSECT	Entomology	0165-0424	Taylor & Francis Group	Hybrid/AP C-GOA
Arthropod Systematics & Phylogeny	ARTHROPOD SYST PHYLO	Entomology	1863-7221	Senckenberg Naturhistorische, Germany	DOA
Arthropoda Selecta	ARTHROPODA SEL	Entomology	0136-006X	KMK Scientific Press/Zoological Museum MGU	DOA
Asian Herpetological Research	ASIAN HERPETOL RES	Zoology	2095-0357	Chinese Academy of Sciences/Science Press	APC-GOA [?]
Asian Myrmecology	ASIAN MYRMECOL	Entomology	1985-1944	International Network for the Study of Asian Ants	DOA
Austral Entomology	AUSTRAL ENTOMOL	Entomology	2052-1758	John Wiley & Sons	Hybrid/AP C-GOA
Bulletin of Insectology	B INSECTOL	Entomology	1721-8861	Department of Agricultural and Food Sciences, Italy	DOA
Belgian Journal of Zoology	BELG J ZOOL	Zoology	0777-6276	Royal Belgian Zoological Society and the Royal Belgian Institute of Natural Sciences	DOA
Caldasia	CALDASIA	Zoology	0366-5232	Universidad Nacional de Colombia/SciELO	DOA
Canadian Entomologist	CAN ENTOMOL	Entomology	0008-347X	Cambridge University Press	Hybrid/AP C-GOA
Canadian Journal of Zoology	CAN J ZOOL	Zoology	0008-4301	Canadian Science Publishing	Hybrid/AP C-GOA
Coleopterists Bulletin	COLEOPTS BULL	Entomology	0010-065X	BioOne/The Coleopterists Society	Hybrid/AP C-GOA
Contributions to Zoology	CONTRIB ZOOL	Zoology	1383-4517	Brill Academic Publishers	APC-GOA
Copeia	COPEIA	Zoology	0045-8511	BioOne/American Society of Ichthyologists and Herpetologists (ASIH)	APC-GOA
Cretaceous Research	CRETACEOUS RES	Paleontology	0195-6671	Elsevier B.V.	Hybrid/AP C-GOA
Current Herpetology	CURR HERPETOL	Zoology	1345-5834	BioOne/The Herpetological Society of Japan	Hybrid/AP C-GOA
Cybium	CYBIUM	Zoology	0399-0974	Société Française d'Ichtyologie	Hybrid/AP C-GOA
Deutsche Entomologische Zeitschrift	DEUT ENTOMOL Z	Entomology	1435-1951	Pensoft Publishers	APC-GOA
Entomologica Americana	ENTOMOL AM NY	Entomology	1947-5136	BioOne/The New York Entomological Society	Hybrid/AP C-GOA

Table 1. Continued.

Journal	Abbreviation used in JCR	Category	ISSN	Publisher	Publishing model
Entomologica Fennica	ENTOMOL FENNICA [merged with Annales Zoologici Fennici]	Entomolog y	0785-8760	Finnish Zoological and Botanical Publishing Board	APC-GOA
Entomological News	ENTOMOL NEWS	Entomolog y	0013-872X	BioOne/The American Entomological Society	Hybrid/AP C-GOA
Entomological Research	ENTOMOL RES	Entomolog y	1738-2297	John Wiley & Sons	Hybrid/AP C-GOA
Entomological Science	ENTOMOL SCI	Entomolog y	1343-8786	John Wiley & Sons	Hybrid/AP C-GOA
European Journal of Entomology	EUR J ENTOMOL	Entomolog y	1210-5759	Institute of Entomology of the Biology Centre, Czech Academy of Sciences	APC-GOA
European Journal of Taxonomy	EUR J TAXON	Zoology	2118-9773	EJT consortium	DOA
Florida Entomologist	FLA ENTOMOL	Entomolog y	0015-4040	BioOne/Florida Entomological Society	APC-GOA
Gayana	GAYANA	Zoology	0717-652X	Universidad de Concepción, Chile/SciELO	APC-GOA
Herpetological Journal	HERPETOL J	Zoology	0268-0130	British Herpetological Society	Hybrid/AP C-GOA
Herpetological Monographs	HERPETOL MONOGR	Zoology	0733-1347	The Herpetologists' League	Hybrid/AP C-GOA
Herpetologica	HERPETOLOGIC A	Zoology	0018-0831	The Herpetologists' League	Hybrid/AP C-GOA
Hystrix-Italian Journal of Mammalogy	HYSTRIX	Zoology	0394-1914	Associazione Teriologica Italiana	DOA
Ichthyological Exploration of Freshwaters	ICHTHYOL EXPLOR FRES	Zoology	0936-9902	Verlag Dr. Friedrich Pfei	Hybrid/AP C-GOA [?]
Ichthyological Research	ICHTHYOL RES	Zoology	1341-8998	Springer Nature	Hybrid/AP C-GOA
Iheringia Serie Zoologia	IHERINGIA SER ZOO	Zoology	0073-4721	Museu de Ciências Naturais, SEMA, Brazil/SciELO	DOA
Insect Systematics & Evolution	INSECT SYST EVOL	Entomolog y	1399-560X	Pensoft Publishers	APC-GOA
Insects	INSECTS	Entomolog y	2075-4450	Multidisciplinary Digital Publishing Institute (MDPI)	APC-GOA
International Journal of Acarology	INT J ACAROL	Entomolog y	0164-7954	Taylor & Francis Group	Hybrid/AP C-GOA
International Journal of Odonatology	INT J ODONATOL	Entomolog y	1388-7890	Taylor & Francis Group	Hybrid/AP C-GOA
International Journal of Primatology	INT J PRIMATOL	Zoology	0164-0291	Springer Nature	Hybrid/AP C-GOA
Invertebrate Systematics	INVERTEBR SYST	Zoology	1445-5226	CSIRO Publishing	Hybrid/AP C-GOA
Journal of Arachnology	J ARACHNOL	Entomolog y	0161-8202	American Arachnological Society	Hybrid/AP C-GOA
Journal of Asia-Pacific Entomology	J ASIA PAC ENTOMOL	Entomolog y	1226-8615	Elsevier B.V.	Hybrid/AP C-GOA
Journal of Conchology	J CONCHOL	Zoology	0022-0019	The Conchological Society of Great Britain and Ireland	Unknown
Journal of Crustacean Biology	J CRUSTACEAN BIOL	Zoology	0278-0372	Oxford University Press	Hybrid/AP C-GOA

Table 1. Continued.

Journal	Abbreviation used in JCR	Category	ISSN	Publisher	Publishing model
Journal of the Entomological Research Society	J ENTOMOL RES SOC	Zoology	1302-0250	Gazi Entomological Research Society (GERS)	DOA [?]
Journal of Helminthology	J HELMINTHOL	Zoology	0022-149X	Cambridge University Press	Hybrid/AP C-GOA

[revised manuscript text omitted]

Figure. 7. Number of citations based on JCR/Clarivate 2019 and notations of observed effects on bibliometric measures compared to *Zootaxa*.

Supplementary File 1

Bibliometric data from 2010–2018 of the 123 selected journals among the 168 journals in Zoology and 101 in Entomology, plus top ten zoological journals (TTJ, eight are on JCR) available in the Web of Science Core Collection, Journal Citation Reports (JCR) Science Edition database of Clarivate.

Journal;Category JCR;Status;Quartile 2019 (*Zootaxa 2018);Metrics;"

All Yrs

";2019;2018;2017;2016;2015;2014;2013;2012;2011;2010;Rest;2010-2018;2014-2018;

ZOOTAXA;ZOOLOGY;Top Ten;3;ALL

citations;19280;0;640;1892;2097;2003;1983;1783;1592;1571;1354;1069;14915;8615;

ZOOTAXA;ZOOLOGY;Top Ten;3;ALL citations (excluding self and

most);12076;0;251;966;1216;1166;1284;1146;1023;1115;902;792;9069;4883;

ZOOTAXA;ZOOLOGY;Top Ten;3;ZOOKEYS;1030;0;52;108;118;118;96;93;83;66;73;0;807;492;

ZOOTAXA;ZOOLOGY;Top Ten;3;Self-citation;6174;0;337;818;763;719;603;544;486;390;379;277;5039;3240;

ZOOTAXA;ZOOLOGY;Top Ten;3;Ratio of self-

citations;0,320228216;#DIV/0!;0,5265625;0,432346723;0,363853124;0,358961558;0,30408472;0,305103758;0,3052763

82;0,248249523;0,279911374;0,259120674;3,12434966;1,985808624;

ZOOTAXA;ZOOLOGY;Top Ten;3;JIF;0;0;0,99;0,931;0,972;0,994;0,906;1,06;0,974;0,927;0,853;0;8,607;4,793;

ZOOTAXA;ZOOLOGY;Top Ten;3;JIF (without self-

citations);0;0;0,598;0,532;0,577;0,608;0,521;0,577;0,58;0,545;0,47;0,5,008;2,836;

ACTA AMAZON;ZOOLOGY;ZOOLOGY;3;ALL citations;1583;10;35;38;87;69;60;42;63;69;97;1013;560;289;

ACTA AMAZON;ZOOLOGY;ZOOLOGY;3;ALL citations (excluding self and

most);1500;8;29;36;86;67;58;41;60;66;92;957;535;276;

ACTA AMAZON;ZOOLOGY;ZOOLOGY;3;ZOOTAXA;45;1;1;0;1;1;0;1;2;4;33;11;4;

ACTA AMAZON;ZOOLOGY;ZOOLOGY;3;Self-citation;38;1;5;1;1;1;1;2;1;1;23;14;9;

ACTA AMAZON;ZOOLOGY;ZOOLOGY;3;Ratio of self-

citations;0,024005054;0,1;0,142857143;0,026315789;0,011494253;0,014492754;0,016666667;0,023809524;0,03174603

2;0,014492754;0,010309278;0,022704837;0,292184193;0,211826605;

ACTA AMAZON;ZOOLOGY;ZOOLOGY;3;JIF;0;0,768;1,042;0,837;0,775;0,408;0;0;0;0;0;3,062;3,062;

ACTA AMAZON;ZOOLOGY;ZOOLOGY;3;JIF (without self-

citations);0;0,705;1,01;0,772;0,716;0,342;0;0;0;0;0;2,84;2,84;

ACTA CHIROPTEROL;ZOOLOGY;ZOOLOGY;3;ALL citations;824;2;26;53;54;41;61;39;67;37;61;383;439;235;

ACTA CHIROPTEROL;ZOOLOGY;ZOOLOGY;3;ALL citations (excluding self and

most);761;0;24;47;49;39;57;38;62;35;59;351;410;216;

ACTA CHIROPTEROL;ZOOLOGY;ZOOLOGY;3;ZOOTAXA;4;0;0;0;1;1;0;0;1;0;1;0;4;2;

ACTA CHIROPTEROL;ZOOLOGY;ZOOLOGY;3;Self-citation;59;2;2;6;4;1;4;1;4;2;1;32;25;17;

ACTA CHIROPTEROL;ZOOLOGY;ZOOLOGY;3;Ratio of self-

citations;0,071601942;1;0,076923077;0,113207547;0,074074074;0,024390244;0,06557377;0,025641026;0,059701493;0

,054054054;0,016393443;0,083550914;0,509958727;0,354168713;

ACTA

CHIROPTEROL;ZOOLOGY;ZOOLOGY;3;JIF;0;1;1,569;1,097;1,04;1,105;1,133;0,831;0,894;1,116;1,012;0,9797;5,94

4;

ACTA CHIROPTEROL;ZOOLOGY;ZOOLOGY;3;JIF (without self-

citations);0;0,899;1,361;0,917;0,907;0,829;0,916;0,674;0,635;0,837;0,671;0;7,747;4,93;

ACTA ZOOL ACAD SCI H;ZOOLOGY;ZOOLOGY;4;ALL citations;598;5;11;15;7;23;23;9;12;13;22;458;135;79;

ACTA ZOOL ACAD SCI H;ZOOLOGY;ZOOLOGY;4;ALL citations (excluding self and

most);432;1;9;13;5;19;18;6;12;13;19;317;114;64;

ACTA ZOOL ACAD SCI H;ZOOLOGY;ZOOLOGY;4;ZOOTAXA;150;3;0;1;2;3;3;2;0;0;3;133;14;9;

ACTA ZOOL ACAD SCI H;ZOOLOGY;ZOOLOGY;4;Self-citation;16;1;2;1;0;1;2;1;0;0;0;8;7;6;

ACTA ZOOL ACAD SCI H;ZOOLOGY;ZOOLOGY;4;Ratio of self-

citations;0,026755853;0,2;0,181818182;0,066666667;0,0,043478261;0,086956522;0,111111111;0;0;0;0,017467249;0,49

0030742;0,378919631;

ACTA ZOOL ACAD SCI

H;ZOOLOGY;ZOOLOGY;4;JIF;0;0,591;0,421;0,846;0,52;0,353;0,5;0,263;0,472;0,564;0,474;0,4,413;2,64;

ACTA ZOOL ACAD SCI H;ZOOLOGY;ZOOLOGY;4;JIF (without self-

citations);0;0,523;0,395;0,769;0,48;0,275;0,464;0,246;0,415;0,545;0,447;0;4,036;2,383;

AFR INVERTEBR;ZOOLOGY;ZOOLOGY;3;ALL citations;199;1;5;12;7;22;14;22;13;14;10;79;119;60;
AFR INVERTEBR;ZOOLOGY;ZOOLOGY;3;ALL citations (excluding self and
most);155;0;3;9;6;12;12;19;11;12;8;63;92;42;
AFR INVERTEBR;ZOOLOGY;ZOOLOGY;3;ZOOTAXA;39;1;1;3;1;9;2;3;2;2;1;14;24;16;
AFR INVERTEBR;ZOOLOGY;ZOOLOGY;3;Self-citation;5;0;1;0;0;1;0;0;0;1;2;3;2;
AFR INVERTEBR;ZOOLOGY;ZOOLOGY;3;Ratio of self-
citations;0,025125628;0,0;2;0;0;0,045454545;0;0;0;0,1;0,025316456;0,345454545;0,245454545;
AFR
INVERTEBR;ZOOLOGY;ZOOLOGY;3;JIF;0;0,81;0,667;0,516;0,622;0,708;0,464;0,5;0,739;0,432;0,8;0,5;448;2,977;
AFR INVERTEBR;ZOOLOGY;ZOOLOGY;3;JIF (without self-
citations);0;0,762;0,571;0,438;0,581;0,5;0,42;0,379;0,587;0,27;0,6;0,4;346;2,51;
AFR ZOOL;ZOOLOGY;ZOOLOGY;3;ALL citations;488;1;8;29;16;51;22;30;48;61;24;198;289;126;
AFR ZOOL;ZOOLOGY;ZOOLOGY;3;ALL citations (excluding self and
most);466;1;7;27;16;50;20;30;46;58;23;188;277;120;
AFR ZOOL;ZOOLOGY;ZOOLOGY;3;ZOOTAXA;12;0;0;2;0;1;1;0;0;1;0;7;5;4;
AFR ZOOL;ZOOLOGY;ZOOLOGY;3;Self-citation;10;0;1;0;0;0;1;0;2;2;1;3;7;2;
AFR ZOOL;ZOOLOGY;ZOOLOGY;3;Ratio of self-
citations;0,020491803;0,0,125;0;0;0,045454545;0;0,041666667;0,032786885;0,041666667;0,015151515;0,286574764;
0,170454545;
AFR ZOOL;ZOOLOGY;ZOOLOGY;3;JIF;0;0,86;0,962;0,761;0,6;0,739;0,612;0,848;0,746;0,9;1,018;0,7;186;3,674;
AFR ZOOL;ZOOLOGY;ZOOLOGY;3;JIF (without self-
citations);0;0,837;0,846;0,687;0,586;0,609;0,5;0,785;0,676;0,783;0,789;0,6;261;3,228;
AM MALACOL BULL;ZOOLOGY;ZOOLOGY;4;ALL citations;558;0;13;11;28;20;45;62;28;30;13;308;250;117;
AM MALACOL BULL;ZOOLOGY;ZOOLOGY;4;ALL citations (excluding self and
most);532;0;12;10;27;20;39;59;23;28;13;301;231;108;
AM MALACOL BULL;ZOOLOGY;ZOOLOGY;4;ZOOTAXA;21;0;1;1;1;0;5;2;4;2;0;5;16;8;
AM MALACOL BULL;ZOOLOGY;ZOOLOGY;4;Self-citation;5;0;0;0;0;1;1;1;0;2;3;1;
AM MALACOL BULL;ZOOLOGY;ZOOLOGY;4;Ratio of self-
citations;0,008960573;#DIV/0!;0;0;0;0,022222222;0,016129032;0,035714286;0;0,006493506;0,07406554;0,02222222;
22;
AM MALACOL
BULL;ZOOLOGY;ZOOLOGY;4;JIF;0;0,558;0,741;0,519;0,911;1,049;0,939;0,843;1,1;219;0,948;0,8;169;4,159;
AM MALACOL BULL;ZOOLOGY;ZOOLOGY;4;JIF (without self-
citations);0;0,558;0,722;0,481;0,844;0,852;0,864;0,843;0,889;0,875;0,914;0,7;284;3,763;
AM MUS NOVIT;ZOOLOGY;ZOOLOGY;1;ALL citations;2312;12;58;53;24;23;47;40;58;33;34;1930;370;205;
AM MUS NOVIT;ZOOLOGY;ZOOLOGY;1;ALL citations (excluding self and
most);2033;8;42;43;22;21;44;36;49;32;31;1705;320;172;
AM MUS NOVIT;ZOOLOGY;ZOOLOGY;1;ZOOTAXA;236;0;5;8;0;2;3;4;9;1;3;201;35;18;
AM MUS NOVIT;ZOOLOGY;ZOOLOGY;1;Self-citation;43;4;11;2;2;0;0;0;0;24;15;15;
AM MUS NOVIT;ZOOLOGY;ZOOLOGY;1;Ratio of self-
citations;0,018598616;0,333333333;0,189655172;0,037735849;0,083333333;0;0;0;0,012435233;0,310724355;0,31
0724355;
AM MUS
NOVIT;ZOOLOGY;ZOOLOGY;1;JIF;0;2,313;1,605;0,979;0,873;1,123;2,186;1,636;1,685;1,882;1,151;0;13,12;6,766;
AM MUS NOVIT;ZOOLOGY;ZOOLOGY;1;JIF (without self-
citations);0;2,042;1,395;0,833;0,836;0,895;1,78;1,309;1,148;1,544;0,932;0;10,672;5,739;
AMPHIBIA REPTILIA;ZOOLOGY;ZOOLOGY;2;ALL citations;1514;16;49;89;46;38;93;79;65;62;64;913;585;315;
AMPHIBIA REPTILIA;ZOOLOGY;ZOOLOGY;2;ALL citations (excluding self and
most);1420;15;45;83;44;36;86;74;64;61;59;853;552;294;
AMPHIBIA REPTILIA;ZOOLOGY;ZOOLOGY;2;ZOOTAXA;36;1;1;1;0;3;0;0;0;2;28;7;5;
AMPHIBIA REPTILIA;ZOOLOGY;ZOOLOGY;2;Self-citation;58;0;3;5;2;2;4;5;1;1;3;32;26;16;
AMPHIBIA REPTILIA;ZOOLOGY;ZOOLOGY;2;Ratio of self-
citations;0,038309115;0,06122449;0,056179775;0,043478261;0,052631579;0,043010753;0,063291139;0,015384615;0,
016129032;0,046875;0,035049288;0,398204644;0,256524858;

AMPHIBIA

REPTILIA;ZOOLOGY;ZOOLOGY;2;JIF;0;1,408;0,943;1,105;1,287;1,396;0,887;1,138;0,68;1,056;0,976;0,9,468;5,618;
 AMPHIBIA REPTILIA;ZOOLOGY;ZOOLOGY;2;JIF (without self-
 citations);0;1,327;0,915;0,947;1,184;1,313;0,84;1,043;0,617;0,895;0,897;0,8,651;5,199;
 ANN CARNEGIE MUS;ZOOLOGY;ZOOLOGY;4;ALL citations;476;1;4;7;6;9;2;10;15;15;12;395;80;28;
 ANN CARNEGIE MUS;ZOOLOGY;ZOOLOGY;4;ALL citations (excluding self and
 most);408;0;3;5;5;9;1;9;13;12;11;340;68;23;
 ANN CARNEGIE MUS;ZOOLOGY;ZOOLOGY;4;ZOOTAXA;46;0;1;0;1;0;1;2;1;0;40;6;2;
 ANN CARNEGIE MUS;ZOOLOGY;ZOOLOGY;4;Self-citation;22;1;0;2;0;0;1;0;0;2;1;15;6;3;
 ANN CARNEGIE MUS;ZOOLOGY;ZOOLOGY;4;Ratio of self-
 citations;0,046218487;1;0;0,285714286;0;0;0,5;0;0,133333333;0,083333333;0,037974684;1,002380952;0,785714286;
 ANN CARNEGIE
 MUS;ZOOLOGY;ZOOLOGY;4;JIF;0;0,688;0,458;0,81;0,75;0,583;0,724;0,56;0,652;0,955;1;0;6,492;3,325;
 ANN CARNEGIE MUS;ZOOLOGY;ZOOLOGY;4;JIF (without self-
 citations);0;0,563;0,375;0,762;0,5;0,458;0,552;0,44;0,565;0,818;0,897;0,5,367;2,647;
 ASIAN HERPETOL RES;ZOOLOGY;ZOOLOGY;3;ALL citations;269;5;26;35;26;47;35;26;30;30;3;6;258;169;
 ASIAN HERPETOL RES;ZOOLOGY;ZOOLOGY;3;ALL citations (excluding self and
 most);223;4;21;26;25;40;27;21;27;26;3;3;216;139;
 ASIAN HERPETOL RES;ZOOLOGY;ZOOLOGY;3;ZOOTAXA;25;1;3;5;0;5;3;3;2;1;0;2;22;16;
 ASIAN HERPETOL RES;ZOOLOGY;ZOOLOGY;3;Self-citation;21;0;2;4;1;2;5;2;1;3;0;1;20;14;
 ASIAN HERPETOL RES;ZOOLOGY;ZOOLOGY;3;Ratio of self-
 citations;0,078066914;0;0,076923077;0,114285714;0,038461538;0,042553191;0,142857143;0,076923077;0,033333333;
 0,1;0;0,166666667;0,625337074;0,415080664;
 ASIAN HERPETOL
 RES;ZOOLOGY;ZOOLOGY;3;JIF;0;1,052;0,721;0,594;0,385;0,5;0,513;0,671;0,681;0,294;0;0;4,359;2,713;
 ASIAN HERPETOL RES;ZOOLOGY;ZOOLOGY;3;JIF (without self-
 citations);0;0,948;0,574;0,464;0,354;0,471;0,41;0,557;0,362;0,118;0;0;3,31;2,273;
 BELG J ZOOL;ZOOLOGY;ZOOLOGY;3;ALL citations;390;2;12;12;9;8;19;7;11;5;19;286;102;60;
 BELG J ZOOL;ZOOLOGY;ZOOLOGY;3;ALL citations (excluding self and
 most);377;2;12;12;9;8;19;7;11;5;16;276;99;60;
 BELG J ZOOL;ZOOLOGY;ZOOLOGY;3;ZOOTAXA;13;0;0;0;0;0;0;0;0;3;10;3;0;
 BELG J ZOOL;ZOOLOGY;ZOOLOGY;3;Self-citation;0;0;0;0;0;0;0;0;0;0;0;0;0;
 BELG J ZOOL;ZOOLOGY;ZOOLOGY;3;Ratio of self-citations;0;0;0;0;0;0;0;0;0;0;0;0;0;
 BELG J ZOOL;ZOOLOGY;ZOOLOGY;3;JIF;0;1,607;0,778;0,826;0,583;0,871;0,333;0,702;0,531;0,36;0,5,591;3,665;
 BELG J ZOOL;ZOOLOGY;ZOOLOGY;3;JIF (without self-
 citations);0;1;0,607;0,741;0,739;0,583;0,806;0,333;0,681;0,531;0,3;0,5,321;3,476;
 CALDASIA;ZOOLOGY;ZOOLOGY;4;ALL citations;368;9;7;13;12;12;18;16;15;23;13;230;129;62;
 CALDASIA;ZOOLOGY;ZOOLOGY;4;ALL citations (excluding self and
 most);314;5;5;13;10;11;16;15;15;19;11;194;115;55;
 CALDASIA;ZOOLOGY;ZOOLOGY;4;ZOOTAXA;26;1;2;0;1;0;2;0;0;1;0;19;6;5;
 CALDASIA;ZOOLOGY;ZOOLOGY;4;Self-citation;28;3;0;0;1;1;0;1;0;3;2;17;8;2;
 CALDASIA;ZOOLOGY;ZOOLOGY;4;Ratio of self-
 citations;0,076086957;0,333333333;0;0;0,083333333;0,083333333;0,0,0625;0;0,130434783;0,153846154;0,073913043;
 0,513447603;0,166666667;
 CALDASIA;ZOOLOGY;ZOOLOGY;4;JIF;0;0,317;0,458;0,292;0,241;0,25;0,266;0,203;0,31;0;0;0;2,02;1,507;
 CALDASIA;ZOOLOGY;ZOOLOGY;4;JIF (without self-
 citations);0;0,317;0,333;0,271;0,19;0,217;0,203;0,174;0,155;0;0;0;1,543;1,214;
 CAN J ZOOL;ZOOLOGY;ZOOLOGY;2;ALL
 citations;11382;47;192;141;159;157;325;193;209;211;302;9446;1889;974;
 CAN J ZOOL;ZOOLOGY;ZOOLOGY;2;ALL citations (excluding self and
 most);10996;46;180;136;153;153;316;190;201;205;295;9121;1829;938;
 CAN J ZOOL;ZOOLOGY;ZOOLOGY;2;ZOOTAXA;125;0;1;1;0;0;0;0;1;0;0;122;3;2;
 CAN J ZOOL;ZOOLOGY;ZOOLOGY;2;Self-citation;261;1;11;4;6;4;9;3;7;6;7;203;57;34;
 CAN J ZOOL;ZOOLOGY;ZOOLOGY;2;Ratio of self-
 citations;0,022930944;0,021276596;0,057291667;0,028368794;0,037735849;0,025477707;0,027692308;0,015544041;0,
 033492823;0,028436019;0,023178808;0,021490578;0,277218016;0,176566325;

CAN J
ZOOLOGY;ZOOLOGY;ZOOLOGY;2;JIF;0;1,243;1,311;1,184;1,347;1,52;1,303;1,346;1,498;1,205;1,196;0;11,91;6,665;
CAN J ZOOLOGY;ZOOLOGY;ZOOLOGY;2;JIF (without self-
citations);0;1,187;1,23;1,14;1,307;1,436;1,244;1,267;1,431;1,129;1,134;0;11,318;6,357;
CONTRIB ZOOLOGY;ZOOLOGY;ZOOLOGY;2;ALL citations;513;4;11;30;65;39;40;18;48;41;29;188;321;185;
CONTRIB ZOOLOGY;ZOOLOGY;ZOOLOGY;2;ALL citations (excluding self and
most);465;3;11;26;55;37;36;16;44;39;26;172;290;165;
CONTRIB ZOOLOGY;ZOOLOGY;ZOOLOGY;2;ZOOTAXA;33;1;0;0;8;1;2;1;2;1;3;14;18;11;
CONTRIB ZOOLOGY;ZOOLOGY;ZOOLOGY;2;Self-citation;15;0;0;4;2;1;2;1;2;1;0;2;13;9;
CONTRIB ZOOLOGY;ZOOLOGY;ZOOLOGY;2;Ratio of self-
citations;0,029239766;0;0;0,133333333;0,030769231;0,025641026;0,05;0,055555556;0,041666667;0,024390244;0;0,01
0638298;0,361356056;0,23974359;
CONTRIB
ZOOLOGY;ZOOLOGY;ZOOLOGY;2;JIF;0;1,242;2,139;1,641;1,972;1,844;1,656;2,029;2,452;1,231;1,118;0;16,082;9,252;
CONTRIB ZOOLOGY;ZOOLOGY;ZOOLOGY;2;JIF (without self-
citations);0;1,121;2,111;1,564;1,889;1,75;1,531;1,829;2,161;1,115;1,059;0;15,009;8,845;
COPEIA;ZOOLOGY;ZOOLOGY;3;ALL citations;5730;18;61;91;143;125;83;103;104;57;98;4847;865;503;
COPEIA;ZOOLOGY;ZOOLOGY;3;ALL citations (excluding self and
most);5347;9;49;82;124;113;79;94;95;53;91;4558;780;447;
COPEIA;ZOOLOGY;ZOOLOGY;3;ZOOTAXA;247;4;7;6;10;6;3;3;7;3;5;193;50;32;
COPEIA;ZOOLOGY;ZOOLOGY;3;Self-citation;136;5;5;3;9;6;1;6;2;1;2;96;35;24;
COPEIA;ZOOLOGY;ZOOLOGY;3;Ratio of self-
citations;0,023734729;0,277777778;0,081967213;0,032967033;0,062937063;0,048;0,012048193;0,058252427;0,019230
769;0,01754386;0,020408163;0,019806066;0,353354721;0,237919502;
COPEIA;ZOOLOGY;ZOOLOGY;3;JIF;0;1,16;1,018;1,22;0,98;1,144;1,034;0,901;0,644;1,101;0,78;0,8,822;5,396;
COPEIA;ZOOLOGY;ZOOLOGY;3;JIF (without self-
citations);0;1,099;0,97;1,145;0,855;1;0,959;0,81;0,593;1,024;0,722;0;8,078;4,929;
CURR HERPETOL;ZOOLOGY;ZOOLOGY;4;ALL citations;150;4;14;7;5;16;13;7;7;5;65;81;55;
CURR HERPETOL;ZOOLOGY;ZOOLOGY;4;ALL citations (excluding self and
most);115;2;11;5;4;13;10;7;3;7;5;48;65;43;
CURR HERPETOL;ZOOLOGY;ZOOLOGY;4;ZOOTAXA;21;1;2;1;0;3;1;0;3;0;0;10;10;7;
CURR HERPETOL;ZOOLOGY;ZOOLOGY;4;Self-citation;14;1;1;1;0;2;0;1;0;0;7;6;5;
CURR HERPETOL;ZOOLOGY;ZOOLOGY;4;Ratio of self-
citations;0,093333333;0,25;0,071428571;0,142857143;0,2;0;0,153846154;0;0,142857143;0;0;0,107692308;0,710989011
;0,568131868;
CURR HERPETOL;ZOOLOGY;ZOOLOGY;4;JIF;0;0,525;0,45;0,524;0;0;0;0;0;0;0,974;0,974;
CURR HERPETOL;ZOOLOGY;ZOOLOGY;4;JIF (without self-
citations);0;0,475;0,35;0,429;0;0;0;0;0;0;0,779;0,779;
CYBIUM;ZOOLOGY;ZOOLOGY;4;ALL citations;859;3;23;24;36;25;33;20;63;25;22;585;271;141;
CYBIUM;ZOOLOGY;ZOOLOGY;4;ALL citations (excluding self and
most);737;3;11;18;25;21;28;17;53;24;19;518;216;103;
CYBIUM;ZOOLOGY;ZOOLOGY;4;ZOOTAXA;68;0;5;1;3;3;3;1;6;1;1;44;24;15;
CYBIUM;ZOOLOGY;ZOOLOGY;4;Self-citation;54;0;7;5;8;1;2;2;4;0;2;23;31;23;
CYBIUM;ZOOLOGY;ZOOLOGY;4;Ratio of self-
citations;0,062863795;0,304347826;0,208333333;0,222222222;0,04;0,060606061;0,1;0,063492063;0,0,090909091;0,0
39316239;1,089910597;0,835509442;
CYBIUM;ZOOLOGY;ZOOLOGY;4;JIF;0;0,534;0,812;0,346;0,575;0,487;0,578;0,726;0,379;0,404;0,411;0;4,718;2,798;
CYBIUM;ZOOLOGY;ZOOLOGY;4;JIF (without self-
citations);0;0,398;0,635;0,308;0,479;0,449;0,422;0,705;0,322;0,33;0,389;0;4,039;2,293;
EUR J TAXON;ZOOLOGY;Top Ten;2;ALL citations;600;30;130;189;86;46;31;56;30;1;0;1;569;482;
EUR J TAXON;ZOOLOGY;Top Ten;2;ALL citations (excluding self and
most);441;17;92;158;61;31;28;35;18;0;0;1;423;370;
EUR J TAXON;ZOOLOGY;Top Ten;2;ZOOTAXA;121;6;28;26;23;12;1;18;7;0;0;0;115;90;
EUR J TAXON;ZOOLOGY;Top Ten;2;Self-citation;38;7;10;5;2;3;2;3;5;1;0;0;31;22;

EUR J TAXON;ZOOLOGY;Top Ten;2;Ratio of self-citations;0,063333333;0,233333333;0,076923077;0,026455026;0,023255814;0,065217391;0,064516129;0,053571429;0,166666667;1;#DIV/0!;0;#DIV/0!;0,256367438;

EUR J TAXON;ZOOLOGY;Top Ten;2;JIF;0;1,393;1,188;0,872;0,649;0,873;1,312;0;0;0;0;4,894;4,894;

EUR J TAXON;ZOOLOGY;Top Ten;2;JIF (without self-citations);0;1,328;1,106;0,73;0,485;0,69;1,208;0;0;0;0;4,219;4,219;

GAYANA;ZOOLOGY;ZOOLOGY;4;ALL citations;246;0;3;2;13;10;7;6;22;4;8;171;75;35;

GAYANA;ZOOLOGY;ZOOLOGY;4;ALL citations (excluding self and most);226;0;3;2;10;9;7;6;22;3;8;156;70;31;

GAYANA;ZOOLOGY;ZOOLOGY;4;ZOOTAXA;12;0;0;0;2;0;0;0;0;10;2;2;

GAYANA;ZOOLOGY;ZOOLOGY;4;Self-citation;8;0;0;0;1;1;0;0;0;1;0;5;3;2;

GAYANA;ZOOLOGY;ZOOLOGY;4;Ratio of self-citations;0,032520325;#DIV/0!;0;0;0,076923077;0,1;0;0;0,25;0;0,029239766;0,426923077;0,176923077;

GAYANA;ZOOLOGY;ZOOLOGY;4;JIF;0;0,179;0,455;0,419;0,553;0,286;0,222;0,267;0,294;0,388;0,397;0,3,281;1,935;

GAYANA;ZOOLOGY;ZOOLOGY;4;JIF (without self-citations);0;0,179;0,364;0,419;0,395;0,257;0,2;0,222;0,206;0,265;0,345;0,2,673;1,635;

HERPETOL J;ZOOLOGY;ZOOLOGY;4;ALL citations;775;3;9;28;43;53;43;34;34;36;33;459;313;176;

HERPETOL J;ZOOLOGY;ZOOLOGY;4;ALL citations (excluding self and most);744;2;9;26;42;52;40;32;30;35;29;447;295;169;

HERPETOL J;ZOOLOGY;ZOOLOGY;4;ZOOTAXA;16;0;0;1;1;1;0;3;0;2;7;9;4;

HERPETOL J;ZOOLOGY;ZOOLOGY;4;Self-citation;15;1;0;1;0;0;2;2;1;1;2;5;9;3;

HERPETOL J;ZOOLOGY;ZOOLOGY;4;Ratio of self-citations;0,019354839;0,333333333;0,035714286;0;0;0,046511628;0,058823529;0,029411765;0,027777778;0,060606061;0,010893246;0,258845046;0,082225914;

HERPETOL J;ZOOLOGY;ZOOLOGY;4;JIF;0;0,561;0,875;1,268;0,896;0,808;0,9;1,338;1,081;0,812;0,661;0,8,639;4,747;

HERPETOL J;ZOOLOGY;ZOOLOGY;4;JIF (without self-citations);0;0,545;0,85;1,225;0,836;0,671;0,688;1,213;0,946;0,719;0,525;0;7,673;4,27;

HERPETOL MONOGR;ZOOLOGY;ZOOLOGY;2;ALL citations;322;1;6;14;9;18;8;10;10;2;6;238;83;55;

HERPETOL MONOGR;ZOOLOGY;ZOOLOGY;2;ALL citations (excluding self and most);299;1;5;10;8;17;6;9;10;2;6;225;73;46;

HERPETOL MONOGR;ZOOLOGY;ZOOLOGY;2;ZOOTAXA;23;0;1;4;1;2;1;0;0;13;10;9;

HERPETOL MONOGR;ZOOLOGY;ZOOLOGY;2;Self-citation;0;0;0;0;0;0;0;0;0;0;0;0;

HERPETOL MONOGR;ZOOLOGY;ZOOLOGY;2;Ratio of self-citations;0;0;0;0;0;0;0;0;0;0;0;0;

HERPETOL MONOGR;ZOOLOGY;ZOOLOGY;2;JIF;0;1,667;1,643;2,2,5;1,9;1,727;1,333;1,818;2,818;1,556;0;17,295;9,77;

HERPETOL MONOGR;ZOOLOGY;ZOOLOGY;2;JIF (without self-citations);0;1,667;1,643;2,2,5;1,9;1,545;1,25;1,545;2,364;1,222;0;15,969;9,588;

HERPETOLOGICA;ZOOLOGY;ZOOLOGY;2;ALL citations;2524;9;35;60;71;49;47;48;69;43;37;2056;459;262;

HERPETOLOGICA;ZOOLOGY;ZOOLOGY;2;ALL citations (excluding self and most);2318;8;30;54;63;44;41;46;62;40;33;1897;413;232;

HERPETOLOGICA;ZOOLOGY;ZOOLOGY;2;ZOOTAXA;130;0;3;3;4;3;5;1;5;1;4;101;29;18;

HERPETOLOGICA;ZOOLOGY;ZOOLOGY;2;Self-citation;76;1;2;3;4;2;1;1;2;2;0;58;17;12;

HERPETOLOGICA;ZOOLOGY;ZOOLOGY;2;Ratio of self-citations;0,030110935;0,111111111;0,057142857;0,05;0,056338028;0,040816327;0,021276596;0,020833333;0,028985507;0,046511628;0;0,028210117;0,321904276;0,225573808;

HERPETOLOGICA;ZOOLOGY;ZOOLOGY;2;JIF;0;1,284;1,38;1,013;1,333;1,312;1,14;1,067;1,08;1,605;1,667;0;11,597;6,178;

HERPETOLOGICA;ZOOLOGY;ZOOLOGY;2;JIF (without self-citations);0;1,216;1,304;0,913;1,205;1,247;1,07;1,056;1,011;1,474;1,536;0;10,816;5,739;

HYSTRIX;ZOOLOGY;ZOOLOGY;2;ALL citations;836;2;44;69;47;108;40;308;36;20;28;134;700;308;

HYSTRIX;ZOOLOGY;ZOOLOGY;2;ALL citations (excluding self and most);816;2;39;68;44;108;37;307;35;19;28;129;685;296;

HYSTRIX;ZOOLOGY;ZOOLOGY;2;ZOOTAXA;2;0;0;0;0;1;1;0;0;0;2;1;

HYSTRIX;ZOOLOGY;ZOOLOGY;2;Self-citation;18;0;5;1;3;0;2;0;1;1;0;5;13;11;

HYSTRIX;ZOOLOGY;ZOOLOGY;2;Ratio of self-citations;0,0215311;0,0,113636364;0,014492754;0,063829787;0,0,05;0,0,027777778;0,05;0,0,037313433;0,319736682;0,241958904;

HYSTRIX;ZOOLOGY;ZOOLOGY;2;JIF;0;1,449;1,195;1,862;1,479;4,333;2,86;0,593;0,352;0,333;0,308;0;13,315;11,729;

HYSTRIX;ZOOLOGY;ZOOLOGY;2;JIF (without self-citations);0;1,372;1,078;1,741;1,354;4,02;2,64;0,5;0,352;0,238;0,256;0;12,179;10,833;

ICHTHYOL EXPLOR FRES;ZOOLOGY;ZOOLOGY;2;ALL citations;528;2;15;10;28;28;42;29;17;24;20;313;213;123;

ICHTHYOL EXPLOR FRES;ZOOLOGY;ZOOLOGY;2;ALL citations (excluding self and most);399;2;8;1;21;21;29;15;13;17;14;258;139;80;

ICHTHYOL EXPLOR FRES;ZOOLOGY;ZOOLOGY;2;ZOOTAXA;124;0;6;9;7;7;13;14;4;6;6;52;72;42;

ICHTHYOL EXPLOR FRES;ZOOLOGY;ZOOLOGY;2;Self-citation;5;0;1;0;0;0;0;0;1;0;3;2;1;

ICHTHYOL EXPLOR FRES;ZOOLOGY;ZOOLOGY;2;Ratio of self-citations;0,009469697;0,0,066666667;0;0;0;0;0,041666667;0,009584665;0,108333333;0,066666667;

ICHTHYOL EXPLOR FRES;ZOOLOGY;ZOOLOGY;2;JIF;0;1,786;0,872;0,783;0,953;0,594;1,4;2,275;1,648;0,9;0,828;0;10,253;4,602;

ICHTHYOL EXPLOR FRES;ZOOLOGY;ZOOLOGY;2;JIF (without self-citations);0;1,786;0,766;0,681;0,719;0,438;1;1,706;1,241;0,617;0,484;0;7,652;3,604;

ICHTHYOL RES;ZOOLOGY;ZOOLOGY;4;ALL citations;853;14;32;33;59;69;37;41;36;42;38;452;387;230;

ICHTHYOL RES;ZOOLOGY;ZOOLOGY;4;ALL citations (excluding self and most);714;9;25;27;46;59;30;36;30;30;34;388;317;187;

ICHTHYOL RES;ZOOLOGY;ZOOLOGY;4;ZOOTAXA;62;3;1;4;1;5;2;4;2;7;1;32;27;13;

ICHTHYOL RES;ZOOLOGY;ZOOLOGY;4;Self-citation;77;2;6;2;12;5;5;1;4;5;3;32;43;30;

ICHTHYOL RES;ZOOLOGY;ZOOLOGY;4;Ratio of self-citations;0,090269637;0,142857143;0,1875;0,060606061;0,203389831;0,072463768;0,135135135;0,024390244;0,11111111;0,119047619;0,078947368;0,07079646;0,992591137;0,659094794;

ICHTHYOL RES;ZOOLOGY;ZOOLOGY;4;JIF;0;0,657;0,98;0,765;1,258;1,023;0,81;0,962;0,895;0,865;0,63;0;8,188;4,836;

ICHTHYOL RES;ZOOLOGY;ZOOLOGY;4;JIF (without self-citations);0;0,576;0,822;0,627;0,899;0,909;0,63;0,657;0,667;0,635;0,49;0;6,336;3,887;

IHERINGIA SER ZOOL;ZOOLOGY;ZOOLOGY;4;ALL citations;710;3;13;57;20;31;41;33;46;40;50;376;331;162;

IHERINGIA SER ZOOL;ZOOLOGY;ZOOLOGY;4;ALL citations (excluding self and most);620;3;12;45;20;26;34;30;39;38;45;328;289;137;

IHERINGIA SER ZOOL;ZOOLOGY;ZOOLOGY;4;ZOOTAXA;69;0;0;8;0;5;3;1;3;1;3;45;24;16;

IHERINGIA SER ZOOL;ZOOLOGY;ZOOLOGY;4;Self-citation;21;0;1;4;0;0;4;2;4;1;2;3;18;9;

IHERINGIA SER ZOOL;ZOOLOGY;ZOOLOGY;4;Ratio of self-citations;0,029577465;0,0,076923077;0,070175439;0;0,097560976;0,060606061;0,086956522;0,025;0,04;0,007978723;0,457222073;0,244659491;

IHERINGIA SER ZOOL;ZOOLOGY;ZOOLOGY;4;JIF;0;0,526;0,42;0,294;0,403;0,216;0,573;0,505;0,423;0,23;0,292;0;3,356;1,906;

IHERINGIA SER ZOOL;ZOOLOGY;ZOOLOGY;4;JIF (without self-citations);0;0,489;0,389;0,271;0,387;0,207;0,445;0,398;0,337;0,214;0,27;0;2,918;1,699;

INT J PRIMATOL;ZOOLOGY;ZOOLOGY;1;ALL citations;3112;31;78;132;72;102;107;121;192;145;122;2010;1071;491;

INT J PRIMATOL;ZOOLOGY;ZOOLOGY;1;ALL citations (excluding self and most);2910;15;68;118;69;97;97;115;187;138;111;1895;1000;449;

INT J PRIMATOL;ZOOLOGY;ZOOLOGY;1;ZOOTAXA;2;0;0;0;0;0;0;0;0;0;2;0;0;

INT J PRIMATOL;ZOOLOGY;ZOOLOGY;1;Self-citation;200;16;10;14;3;5;10;6;5;7;11;113;71;42;

INT J PRIMATOL;ZOOLOGY;ZOOLOGY;1;Ratio of self-citations;0,064267352;0,516129032;0,128205128;0,106060606;0,041666667;0,049019608;0,093457944;0,049586777;0,026041667;0,048275862;0,090163934;0,056218905;0,632478193;0,418409953;

INT J

PRIMATOL;ZOOLOGY;ZOOLOGY;1;JIF;0;1,858;1,922;1,278;1,285;1,649;1,993;1,994;1,786;1,538;1,793;0;15,238;8,
127;

INT J PRIMATOL;ZOOLOGY;ZOOLOGY;1;JIF (without self-

citations);0;1,646;1,5;1,157;1,123;1,455;1,827;1,768;1,558;1,328;1,687;0;13,403;7,062;

INVERTEBR SYST;ZOOLOGY;ZOOLOGY;1;ALL citations;1125;19;120;100;68;49;107;50;80;62;76;394;712;444;

INVERTEBR SYST;ZOOLOGY;ZOOLOGY;1;ALL citations (excluding self and

most);894;15;100;84;54;40;92;34;67;39;62;307;572;370;

INVERTEBR SYST;ZOOLOGY;ZOOLOGY;1;ZOOTAXA;171;3;16;9;11;5;9;12;10;16;8;72;96;50;

INVERTEBR SYST;ZOOLOGY;ZOOLOGY;1;Self-citation;60;1;4;7;3;4;6;4;3;7;6;15;44;24;

INVERTEBR SYST;ZOOLOGY;ZOOLOGY;1;Ratio of self-

citations;0,053333333;0,052631579;0,033333333;0,07;0,044117647;0,081632653;0,056074766;0,08;0,0375;0,11290322

6;0,078947368;0,038071066;0,594508994;0,2851584;

INVERTEBR

SYST;ZOOLOGY;ZOOLOGY;1;JIF;0;2,2;2,306;1,651;2,172;2,155;2,264;1,594;1,983;1,351;2,492;0;17,968;10,548;

INVERTEBR SYST;ZOOLOGY;ZOOLOGY;1;JIF (without self-

citations);0;2,09;2,167;1,603;2,031;2,07;2,069;1,469;1,817;1,298;2,206;0;16,73;9,94;

J CONCHOL;ZOOLOGY;ZOOLOGY;3;ALL citations;263;0;6;23;10;11;9;11;5;9;14;165;98;59;

J CONCHOL;ZOOLOGY;ZOOLOGY;3;ALL citations (excluding self and

most);227;0;5;21;9;8;7;9;4;8;14;142;85;50;

J CONCHOL;ZOOLOGY;ZOOLOGY;3;ZOOTAXA;18;0;0;0;2;1;2;0;0;0;13;5;3;

J CONCHOL;ZOOLOGY;ZOOLOGY;3;Self-citation;18;0;1;2;1;1;1;0;1;1;0;10;8;6;

J CONCHOL;ZOOLOGY;ZOOLOGY;3;Ratio of self-

citations;0,068441065;#DIV/0!;0,166666667;0,086956522;0,1;0,090909091;0,111111111;0,0;2,0;1,11111111;0,0;0,060606

061;0,866754502;0,55564339;

J

CONCHOL;ZOOLOGY;ZOOLOGY;3;JIF;0;0,784;0,311;0,512;0,341;0,38;0,523;0,85;0,447;0,54;0,135;0;4,039;2,067;

J CONCHOL;ZOOLOGY;ZOOLOGY;3;JIF (without self-

citations);0;0,703;0,311;0,419;0,293;0,28;0,432;0,6;0,383;0,42;0,108;0;3,246;1,735;

J CRUSTACEAN BIOL;ZOOLOGY;ZOOLOGY;2;ALL

citations;2576;34;118;109;68;118;101;96;113;73;80;1666;876;514;

J CRUSTACEAN BIOL;ZOOLOGY;ZOOLOGY;2;ALL citations (excluding self and

most);2193;19;102;90;59;97;92;88;92;61;69;1424;750;440;

J CRUSTACEAN BIOL;ZOOLOGY;ZOOLOGY;2;ZOOTAXA;169;2;6;8;0;11;4;4;12;4;4;114;53;29;

J CRUSTACEAN BIOL;ZOOLOGY;ZOOLOGY;2;Self-citation;214;13;10;11;9;10;5;4;9;8;7;128;73;45;

J CRUSTACEAN BIOL;ZOOLOGY;ZOOLOGY;2;Ratio of self-

citations;0,083074534;0,382352941;0,084745763;0,100917431;0,132352941;0,084745763;0,04950495;0,041666667;0,0

79646018;0,109589041;0,0875;0,076830732;0,770668574;0,452266848;

J CRUSTACEAN

BIOL;ZOOLOGY;ZOOLOGY;2;JIF;0;1,254;1,069;1,119;1,064;0,922;1,081;1,187;1,019;1,116;1,115;0;9,692;5,255;

J CRUSTACEAN BIOL;ZOOLOGY;ZOOLOGY;2;JIF (without self-

citations);0;1,138;0,909;1,017;0,919;0,705;0,936;1,035;0,832;0,939;0,919;0;8,211;4,486;

J HELMINTHOL;ZOOLOGY;ZOOLOGY;2;ALL citations;2019;93;132;139;133;148;101;79;90;65;58;981;945;653;

J HELMINTHOL;ZOOLOGY;ZOOLOGY;2;ALL citations (excluding self and

most);1874;84;116;123;123;135;94;69;77;59;53;941;849;591;

J HELMINTHOL;ZOOLOGY;ZOOLOGY;2;ZOOTAXA;24;4;3;2;2;0;1;3;2;1;1;5;15;8;

J HELMINTHOL;ZOOLOGY;ZOOLOGY;2;Self-citation;121;5;13;14;8;13;6;7;11;5;4;35;81;54;

J HELMINTHOL;ZOOLOGY;ZOOLOGY;2;Ratio of self-

citations;0,059930659;0,053763441;0,098484848;0,100719424;0,060150376;0,087837838;0,059405941;0,088607595;0,

122222222;0,076923077;0,068965517;0,03567788;0,763316839;0,406598427;

J

HELMINTHOL;ZOOLOGY;ZOOLOGY;2;JIF;0;1,54;1,436;1,344;1,42;1,63;1,421;1,303;1,157;1,38;1,544;0;12,635;7,2

51;

J HELMINTHOL;ZOOLOGY;ZOOLOGY;2;JIF (without self-

citations);0;1,386;1,272;1,22;1,266;1,519;1,35;1,204;1,104;1,314;1,447;0;11,696;6,627;

J HERPETOL;ZOOLOGY;ZOOLOGY;3;ALL citations;3349;13;52;82;87;104;76;85;123;116;93;2518;818;401;

J HERPETOL;ZOOLOGY;ZOOLOGY;3;ALL citations (excluding self and
most);3137;9;42;73;84;96;73;82;115;112;87;2364;764;368;
J HERPETOL;ZOOLOGY;ZOOLOGY;3;ZOOTAXA;109;3;8;7;3;1;3;2;5;4;69;37;22;
J HERPETOL;ZOOLOGY;ZOOLOGY;3;Self-citation;103;1;2;2;0;7;0;1;3;0;2;85;17;11;
J HERPETOL;ZOOLOGY;ZOOLOGY;3;Ratio of self-
citations;0,030755449;0,076923077;0,038461538;0,024390244;0;0,067307692;0;0,011764706;0,024390244;0;0,021505
376;0,03375695;0,187819801;0,130159475;
J
HERPETOL;ZOOLOGY;ZOOLOGY;3;JIF;0;1,078;1,077;0,893;0,838;0,832;1,03;0,911;0,865;1.030;0.971;0;6,446;4,67
;
J HERPETOL;ZOOLOGY;ZOOLOGY;3;JIF (without self-
citations);0;1;1;0,821;0,791;0,791;0,994;0,876;0,798;0,963;0,942;0;7,976;4,397;
J MAMMAL;ZOOLOGY;ZOOLOGY;1;ALL
citations;8779;102;220;334;523;268;219;242;405;288;321;5857;2820;1564;
J MAMMAL;ZOOLOGY;ZOOLOGY;1;ALL citations (excluding self and
most);7813;45;157;283;417;247;194;213;377;257;294;5329;2439;1298;
J MAMMAL;ZOOLOGY;ZOOLOGY;1;ZOOTAXA;12;0;0;0;0;1;0;0;0;11;1;1;
J MAMMAL;ZOOLOGY;ZOOLOGY;1;Self-citation;954;57;63;51;106;21;24;29;28;31;27;517;380;265;
J MAMMAL;ZOOLOGY;ZOOLOGY;1;Ratio of self-
citations;0,108668413;0,558823529;0,286363636;0,152694611;0,202676864;0,078358209;0,109589041;0,119834711;0,
069135802;0,107638889;0,08411215;0,088270446;1,210403913;0,829682361;
J
MAMMAL;ZOOLOGY;ZOOLOGY;1;JIF;0;1,891;2,13;2,139;1,63;1,558;1,84;2,225;2,308;1,614;1,541;0;16,985;9,297;
J MAMMAL;ZOOLOGY;ZOOLOGY;1;JIF (without self-
citations);0;1,502;1,767;1,735;1,443;1,383;1,688;1,749;1,766;1,42;1,361;0;14,312;8,016;
J MOLLUS STUD;ZOOLOGY;ZOOLOGY;2;ALL citations;1887;11;63;86;121;99;88;67;42;84;60;1166;710;457;
J MOLLUS STUD;ZOOLOGY;ZOOLOGY;2;ALL citations (excluding self and
most);1753;10;58;84;119;90;79;62;39;74;58;1080;663;430;
J MOLLUS STUD;ZOOLOGY;ZOOLOGY;2;ZOOTAXA;40;0;1;1;0;1;2;0;0;3;0;32;8;5;
J MOLLUS STUD;ZOOLOGY;ZOOLOGY;2;Self-citation;94;1;4;1;2;8;7;5;3;7;2;54;39;22;
J MOLLUS STUD;ZOOLOGY;ZOOLOGY;2;Ratio of self-
citations;0,04981452;0,090909091;0,063492063;0,011627907;0,016528926;0,080808081;0,079545455;0,074626866;0,0
71428571;0,083333333;0,033333333;0,046312178;0,514724535;0,252002431;
J MOLLUS
STUD;ZOOLOGY;ZOOLOGY;2;JIF;0;1,461;1,345;1,483;1,25;1,185;1,362;1,495;1,358;1,227;0,969;0;11,674;6,625;
J MOLLUS STUD;ZOOLOGY;ZOOLOGY;2;JIF (without self-
citations);0;1,412;1,227;1,392;1,109;1,059;1,17;1,361;1,274;1,124;0,897;0;10,613;5,957;
J NAT HIST;ZOOLOGY;ZOOLOGY;3;ALL citations;3334;30;152;142;120;118;155;145;121;135;102;2114;1190;687;
J NAT HIST;ZOOLOGY;ZOOLOGY;3;ALL citations (excluding self and
most);2732;22;101;105;92;103;132;127;94;119;77;1760;950;533;
J NAT HIST;ZOOLOGY;ZOOLOGY;3;ZOOTAXA;506;7;38;24;17;11;17;14;25;14;23;316;183;107;
J NAT HIST;ZOOLOGY;ZOOLOGY;3;Self-citation;96;1;13;13;11;4;6;4;2;2;38;57;47;
J NAT HIST;ZOOLOGY;ZOOLOGY;3;Ratio of self-
citations;0,028794241;0,033333333;0,085526316;0,091549296;0,091666667;0,033898305;0,038709677;0,027586207;0,
016528926;0,014814815;0,019607843;0,017975402;0,419888051;0,341350261;
J NAT
HIST;ZOOLOGY;ZOOLOGY;3;JIF;0;1,032;0,837;0,875;0,834;1,01;0,881;0,927;0,778;0,953;0,782;0;7,877;4,437;
J NAT HIST;ZOOLOGY;ZOOLOGY;3;JIF (without self-
citations);0;0,94;0,773;0,842;0,779;0,94;0,835;0,868;0,716;0,862;0,728;0;7,343;4,169;
J NEMATOL;ZOOLOGY;ZOOLOGY;2;ALL citations;2719;9;43;104;67;86;63;52;108;43;49;2095;615;363;
J NEMATOL;ZOOLOGY;ZOOLOGY;2;ALL citations (excluding self and
most);2513;8;36;93;61;82;61;46;97;40;42;1947;558;333;
J NEMATOL;ZOOLOGY;ZOOLOGY;2;ZOOTAXA;63;1;1;2;1;0;0;1;0;2;55;7;4;
J NEMATOL;ZOOLOGY;ZOOLOGY;2;Self-citation;143;0;6;9;5;4;2;6;10;3;5;93;50;26;

J NEMATOL;ZOOLOGY;ZOOLOGY;2;Ratio of self-citations;0,052592865;0,0,139534884;0,086538462;0,074626866;0,046511628;0,031746032;0,115384615;0,092592593;0,069767442;0,102040816;0,044391408;0,758743337;0,378957871;

J NEMATOL;ZOOLOGY;ZOOLOGY;2;JIF;0;1,47;1,386;1,2;1,087;1,333;1,081;0,689;0,333;0,522;0,506;0,8;137;6,087;

J NEMATOL;ZOOLOGY;ZOOLOGY;2;JIF (without self-citations);0;1,32;1,057;1,067;0,957;1,217;0,851;0,541;0,333;0,463;0,456;0,6,942;5,149;

J ZOOL SYST EVOL RES;ZOOLOGY;ZOOLOGY;1;ALL citations;1173;66;94;55;72;84;64;52;49;91;56;490;617;369;

J ZOOL SYST EVOL RES;ZOOLOGY;ZOOLOGY;1;ALL citations (excluding self and most);1047;54;81;48;65;76;57;44;45;77;53;447;546;327;

J ZOOL SYST EVOL RES;ZOOLOGY;ZOOLOGY;1;ZOOTAXA;79;9;4;1;5;4;1;6;1;13;1;34;36;15;

J ZOOL SYST EVOL RES;ZOOLOGY;ZOOLOGY;1;Self-citation;47;3;9;6;2;4;6;2;3;1;2;9;35;27;

J ZOOL SYST EVOL RES;ZOOLOGY;ZOOLOGY;1;Ratio of self-citations;0,040068201;0,045454545;0,095744681;0,109090909;0,027777778;0,047619048;0,09375;0,038461538;0,06122449;0,010989011;0,035714286;0,018367347;0,52037174;0,373982415;

J ZOOL SYST EVOL RES;ZOOLOGY;ZOOLOGY;1;JIF;0;2,159;2,268;3,286;2,444;1,821;1,677;1,91;1,796;1,95;1,384;0;18,536;11,496;

J ZOOL SYST EVOL RES;ZOOLOGY;ZOOLOGY;1;JIF (without self-citations);0;1,942;2,089;3,089;2,302;1,761;1,6;1,854;1,656;1,788;1,302;0;17,441;10,841;

MALACOLOGIA;ZOOLOGY;ZOOLOGY;1;ALL citations;967;2;7;101;17;44;15;23;21;25;51;661;304;184;

MALACOLOGIA;ZOOLOGY;ZOOLOGY;1;ALL citations (excluding self and most);897;2;7;90;16;42;15;22;18;22;50;613;282;170;

MALACOLOGIA;ZOOLOGY;ZOOLOGY;1;ZOOTAXA;32;0;0;5;0;0;1;0;3;1;22;10;5;

MALACOLOGIA;ZOOLOGY;ZOOLOGY;1;Self-citation;38;0;0;6;1;2;0;0;3;0;0;26;12;9;

MALACOLOGIA;ZOOLOGY;ZOOLOGY;1;Ratio of self-citations;0,039296794;0;0;0,059405941;0,058823529;0,045454545;0;0;0,142857143;0;0;0,039334342;0,306541158;0,163684015;

MALACOLOGIA;ZOOLOGY;ZOOLOGY;1;JIF;0;13,5;3,25;1,657;0,943;1,211;0,532;0,977;1,592;1,429;1,024;0;12,615;7,593;

MALACOLOGIA;ZOOLOGY;ZOOLOGY;1;JIF (without self-citations);0;13,375;3,25;1,629;0,886;1,053;0,489;0,814;1,449;1,321;0,951;0;11,842;7,307;

MAMMALIA;ZOOLOGY;ZOOLOGY;2;ALL citations;1318;19;51;57;66;47;56;25;51;51;71;824;475;277;

MAMMALIA;ZOOLOGY;ZOOLOGY;2;ALL citations (excluding self and most);1250;19;46;50;58;44;55;24;51;47;70;786;445;253;

MAMMALIA;ZOOLOGY;ZOOLOGY;2;ZOOTAXA;5;0;0;1;0;1;0;0;1;0;2;3;2;

MAMMALIA;ZOOLOGY;ZOOLOGY;2;Self-citation;63;0;5;6;8;2;1;1;0;3;1;36;27;22;

MAMMALIA;ZOOLOGY;ZOOLOGY;2;Ratio of self-citations;0,047799697;0;0,098039216;0,105263158;0,121212121;0,042553191;0,017857143;0,04;0;0,058823529;0,014084507;0,04368932;0,497832866;0,384924829;

MAMMALIA;ZOOLOGY;ZOOLOGY;2;JIF;0;0,679;0,732;0,714;0,805;0,538;0,681;0,824;0,809;0,808;0,616;0,6,527;3,47;

MAMMALIA;ZOOLOGY;ZOOLOGY;2;JIF (without self-citations);0;0,61;0,65;0,639;0,754;0,479;0,628;0,686;0,755;0,702;0,525;0;5,818;3,15;

MOLLUSCAN RES;ZOOLOGY;ZOOLOGY;3;ALL citations;317;10;23;37;22;19;20;17;25;19;18;107;200;121;

MOLLUSCAN RES;ZOOLOGY;ZOOLOGY;3;ALL citations (excluding self and most);288;9;22;31;19;18;19;16;25;18;17;94;185;109;

MOLLUSCAN RES;ZOOLOGY;ZOOLOGY;3;ZOOTAXA;14;0;0;0;2;1;1;0;0;0;10;4;4;

MOLLUSCAN RES;ZOOLOGY;ZOOLOGY;3;Self-citation;15;1;1;6;1;0;0;1;0;1;1;3;11;8;

MOLLUSCAN RES;ZOOLOGY;ZOOLOGY;3;Ratio of self-citations;0,047318612;0,1,0,043478261;0,162162162;0,045454545;0;0,058823529;0,052631579;0,055555556;0,028037383;0,418105632;0,251094968;

MOLLUSCAN RES;ZOOLOGY;ZOOLOGY;3;JIF;0;0,938;0,845;0,574;0,491;0,708;0,512;0,69;0,617;0,906;0,625;0;5,968;3,13;

MOLLUSCAN RES;ZOOLOGY;ZOOLOGY;3;JIF (without self-citations);0;0,828;0,81;0,519;0,434;0,583;0,512;0,667;0,617;0,781;0,542;0;5,465;2,858;

NAUPLIUS;ZOOLOGY;ZOOLOGY;4;ALL citations;275;3;19;30;18;19;15;30;18;13;8;102;170;101;

NAUPLIUS;ZOOLOGY;ZOOLOGY;4;ALL citations (excluding self and
most);217;2;9;26;14;17;12;25;14;11;6;81;134;78;
NAUPLIUS;ZOOLOGY;ZOOLOGY;4;ZOOTAXA;46;1;7;3;4;2;1;5;4;1;2;16;29;17;
NAUPLIUS;ZOOLOGY;ZOOLOGY;4;Self-citation;12;0;3;1;0;0;2;0;0;1;0;5;7;6;
NAUPLIUS;ZOOLOGY;ZOOLOGY;4;Ratio of self-
citations;0,043636364;0,0,157894737;0,033333333;0,0,0,133333333;0,0,0,076923077;0,0,049019608;0,40148448;0,324
561404;
NAUPLIUS;ZOOLOGY;ZOOLOGY;4;JIF;0,0,69;1,381;0;0;0;0;0;0;0;0;1,381;1,381;
NAUPLIUS;ZOOLOGY;ZOOLOGY;4;JIF (without self-citations);0,0,634;1,063;0;0;0;0;0;0;0;0;1,063;1,063;
NAUTILUS;ZOOLOGY;ZOOLOGY;3;ALL citations;475;1;8;25;11;20;17;9;11;10;14;349;125;81;
NAUTILUS;ZOOLOGY;ZOOLOGY;3;ALL citations (excluding self and
most);415;0;6;24;10;19;16;8;9;9;13;301;114;75;
NAUTILUS;ZOOLOGY;ZOOLOGY;3;ZOOTAXA;50;1;2;0;1;1;0;1;1;0;42;7;4;
NAUTILUS;ZOOLOGY;ZOOLOGY;3;Self-citation;10;0;0;1;0;0;1;0;1;0;1;6;4;2;
NAUTILUS;ZOOLOGY;ZOOLOGY;3;Ratio of self-
citations;0,021052632;0,0,0,04;0,0,0,058823529;0,0,090909091;0,0,071428571;0,017191977;0,261161192;0,098823529
;
NAUTILUS;ZOOLOGY;ZOOLOGY;3;JIF;0;1,189;0,789;0,457;0,343;0,472;0,279;0,458;0,574;0,481;0,5,042;3,25;
NAUTILUS;ZOOLOGY;ZOOLOGY;3;JIF (without self-
citations);0,0,97;1,108;0,737;0,4;0,343;0,417;0,256;0,354;0,5;0,444;0,4,559;3,005;
NEMATOLOGY;ZOOLOGY;ZOOLOGY;3;ALL citations;1950;42;81;109;110;101;122;77;116;107;66;1019;889;523;
NEMATOLOGY;ZOOLOGY;ZOOLOGY;3;ALL citations (excluding self and
most);1525;26;51;76;72;78;88;54;92;90;46;852;647;365;
NEMATOLOGY;ZOOLOGY;ZOOLOGY;3;ZOOTAXA;50;5;4;4;5;0;2;5;2;3;2;18;27;15;
NEMATOLOGY;ZOOLOGY;ZOOLOGY;3;Self-citation;375;11;26;29;33;23;32;18;22;14;18;149;215;143;
NEMATOLOGY;ZOOLOGY;ZOOLOGY;3;Ratio of self-
citations;0,192307692;0,261904762;0,320987654;0,266055046;0,3;0,227722772;0,262295082;0,233766234;0,18965517
2;0,130841121;0,272727273;0,146221786;2,204050355;1,377060554;
NEMATOLOGY;ZOOLOGY;ZOOLOGY;3;JIF;0;1,188;1,216;1,12;1,162;1,061;1,239;1,247;0,914;0,911;0,962;0,9,832
;5,798;
NEMATOLOGY;ZOOLOGY;ZOOLOGY;3;JIF (without self-
citations);0,0,844;0,869;0,749;0,686;0,68;0,711;0,861;0,638;0,631;0,589;0,6,414;3,695;
NEOTROP ICHTHYOL;ZOOLOGY;ZOOLOGY;2;ALL
citations;1635;8;95;140;85;111;100;146;135;136;121;558;1069;531;
NEOTROP ICHTHYOL;ZOOLOGY;ZOOLOGY;2;ALL citations (excluding self and
most);1389;7;77;122;73;99;81;124;119;123;96;468;914;452;
NEOTROP ICHTHYOL;ZOOLOGY;ZOOLOGY;2;ZOOTAXA;117;0;3;10;7;7;13;10;5;6;12;44;73;40;
NEOTROP ICHTHYOL;ZOOLOGY;ZOOLOGY;2;Self-citation;129;1;15;8;5;5;6;12;11;7;13;46;82;39;
NEOTROP ICHTHYOL;ZOOLOGY;ZOOLOGY;2;Ratio of self-
citations;0,078899083;0,125;0,157894737;0,057142857;0,058823529;0,045045045;0,06;0,082191781;0,081481481;0,05
1470588;0,107438017;0,082437276;0,701488036;0,378906168;
NEOTROP
ICHTHYOL;ZOOLOGY;ZOOLOGY;2;JIF;0;1,741;1,543;1,216;1,203;0,917;0,802;0,766;1,048;1,064;0,774;0,9,333;5,6
81;
NEOTROP ICHTHYOL;ZOOLOGY;ZOOLOGY;2;JIF (without self-
citations);0;1,57;1,243;0,928;0,973;0,72;0,564;0,52;0,784;0,669;0,497;0,6,898;4,428;
NEW ZEAL J ZOOL;ZOOLOGY;ZOOLOGY;3;ALL citations;928;13;19;30;17;16;25;19;17;11;14;747;168;107;
NEW ZEAL J ZOOL;ZOOLOGY;ZOOLOGY;3;ALL citations (excluding self and
most);806;12;16;25;15;12;24;19;15;9;13;646;148;92;
NEW ZEAL J ZOOL;ZOOLOGY;ZOOLOGY;3;ZOOTAXA;89;1;1;3;1;2;0;0;1;2;1;77;11;7;
NEW ZEAL J ZOOL;ZOOLOGY;ZOOLOGY;3;Self-citation;33;0;2;2;1;2;1;0;1;0;0;24;9;8;
NEW ZEAL J ZOOL;ZOOLOGY;ZOOLOGY;3;Ratio of self-
citations;0,035560345;0,0,105263158;0,066666667;0,058823529;0,125;0,04;0,0,058823529;0,0,0,032128514;0,4545768
83;0,395753354;

NEW ZEAL J

ZOOLOG;ZOOLOGY;ZOOLOGY;3;JIF;0;0,98;0,673;0,415;0,811;0,758;0,964;0,745;0,889;0,919;0,629;0,6,803;3,621;
NEW ZEAL J ZOOLOG;ZOOLOGY;ZOOLOGY;3;JIF (without self-
citations);0;0,9;0,654;0,302;0,66;0,742;0,875;0,745;0,741;0,774;0,486;0,5,979;3,233;
ORG DIVERS EVOL;ZOOLOGY;ZOOLOGY;1;ALL
citations;1512;20;60;123;133;115;69;71;363;89;65;404;1088;500;
ORG DIVERS EVOL;ZOOLOGY;ZOOLOGY;1;ALL citations (excluding self and
most);1385;15;54;104;119;104;65;65;338;86;58;377;993;446;
ORG DIVERS EVOL;ZOOLOGY;ZOOLOGY;1;ZOOTAXA;101;4;3;15;9;8;2;5;24;3;5;23;74;37;
ORG DIVERS EVOL;ZOOLOGY;ZOOLOGY;1;Self-citation;26;1;3;4;5;3;2;1;1;0;2;4;21;17;
ORG DIVERS EVOL;ZOOLOGY;ZOOLOGY;1;Ratio of self-
citations;0,017195767;0,05;0,05;0,032520325;0,037593985;0,026086957;0,028985507;0,014084507;0,002754821;0;0,0
30769231;0,00990099;0,222795333;0,175186774;
ORG DIVERS
EVOL;ZOOLOGY;ZOOLOGY;1;JIF;0;2,153;2,143;2,369;2,313;1,734;2,888;3,365;2,259;1,648;1,581;0;20,3;11,447;
ORG DIVERS EVOL;ZOOLOGY;ZOOLOGY;1;JIF (without self-
citations);0;2,071;2,086;2,32;2,205;1,658;2,825;3,27;2,19;1,593;1,516;0;19,663;11,094;
PAC SCI;ZOOLOGY;ZOOLOGY;3;ALL citations;1475;5;35;26;38;40;49;66;25;33;41;1117;353;188;
PAC SCI;ZOOLOGY;ZOOLOGY;3;ALL citations (excluding self and
most);1354;4;31;20;36;32;48;61;23;29;39;1031;319;167;
PAC SCI;ZOOLOGY;ZOOLOGY;3;ZOOTAXA;72;0;1;2;0;2;1;2;0;2;1;61;11;6;
PAC SCI;ZOOLOGY;ZOOLOGY;3;Self-citation;49;1;3;4;2;6;0;3;2;2;1;25;23;15;
PAC SCI;ZOOLOGY;ZOOLOGY;3;Ratio of self-
citations;0,033220339;0,2;0,085714286;0,153846154;0,052631579;0,15;0;0,045454545;0,08;0,060606061;0,024390244;
0,022381379;0,652642868;0,442192019;
PAC SCI;ZOOLOGY;ZOOLOGY;3;JIF;0;0,824;0,861;0,822;0,9;1,163;0,924;0,824;0,541;0,862;0,584;0,7,481;4,67;
PAC SCI;ZOOLOGY;ZOOLOGY;3;JIF (without self-
citations);0;0,73;0,778;0,644;0,813;1,035;0,848;0,716;0,518;0,747;0,517;0;6,616;4,118;
PHYLLMEDUSA;ZOOLOGY;ZOOLOGY;4;ALL citations;227;2;7;16;12;14;12;8;16;15;12;113;112;61;
PHYLLMEDUSA;ZOOLOGY;ZOOLOGY;4;ALL citations (excluding self and
most);200;2;6;16;9;13;10;6;15;12;12;99;99;54;
PHYLLMEDUSA;ZOOLOGY;ZOOLOGY;4;ZOOTAXA;17;0;0;0;1;1;0;1;1;2;0;11;6;2;
PHYLLMEDUSA;ZOOLOGY;ZOOLOGY;4;Self-citation;10;0;1;0;2;0;2;1;0;1;0;3;7;5;
PHYLLMEDUSA;ZOOLOGY;ZOOLOGY;4;Ratio of self-
citations;0,044052863;0,0,142857143;0,0,166666667;0,0,166666667;0,125;0,0,066666667;0,0,026548673;0,667857143;
0,476190476;
PHYLLMEDUSA;ZOOLOGY;ZOOLOGY;4;JIF;0;0,383;0,364;0,333;0,548;0,185;0;0;0;0;0;1,43;1,43;
PHYLLMEDUSA;ZOOLOGY;ZOOLOGY;4;JIF (without self-
citations);0;0,367;0,255;0,286;0,484;0,148;0;0;0;0;0;1,173;1,173;
PRIMATES;ZOOLOGY;ZOOLOGY;2;ALL citations;1954;27;76;91;81;55;84;69;53;56;54;1308;619;387;
PRIMATES;ZOOLOGY;ZOOLOGY;2;ALL citations (excluding self and
most);1696;20;58;77;65;51;75;58;43;47;48;1154;522;326;
PRIMATES;ZOOLOGY;ZOOLOGY;2;Self-citation;156;5;14;10;15;2;7;5;3;4;2;89;62;48;
PRIMATES;ZOOLOGY;ZOOLOGY;2;Self-citation;102;2;4;4;1;2;2;6;7;5;4;65;35;13;
PRIMATES;ZOOLOGY;ZOOLOGY;2;Ratio of self-
citations;0,052200614;0,074074074;0,052631579;0,043956044;0,012345679;0,036363636;0,023809524;0,086956522;0,
132075472;0,089285714;0,074074074;0,04969419;0,551498244;0,169106462;
PRIMATES;ZOOLOGY;ZOOLOGY;2;JIF;0;1,59;1,363;1,202;1,196;1,142;1,337;1,395;1,292;1,405;1,256;0;11,588;6,2
4;
PRIMATES;ZOOLOGY;ZOOLOGY;2;JIF (without self-
citations);0;1,362;1,124;1,02;0,98;1,066;1,198;1,267;1,202;1,333;1,163;0;10,353;5,388;
REC AUST MUS;ZOOLOGY;ZOOLOGY;4;ALL citations;652;6;5;7;7;8;3;2;4;18;15;577;69;30;
REC AUST MUS;ZOOLOGY;ZOOLOGY;4;ALL citations (excluding self and
most);479;1;4;5;4;6;0;1;2;16;15;425;53;19;
REC AUST MUS;ZOOLOGY;ZOOLOGY;4;ZOOTAXA;134;1;1;2;0;1;2;0;0;0;127;6;6;
REC AUST MUS;ZOOLOGY;ZOOLOGY;4;Self-citation;39;4;0;0;3;1;1;1;2;2;0;25;10;5;

REC AUST MUS;ZOOLOGY;ZOOLOGY;4;Ratio of self-
citations;0,059815951;0,666666667;0;0;0,428571429;0,125;0,333333333;0,5;0,5;0,111111111;0;0,043327556;1,998015
873;0,886904762;
REC AUST MUS;ZOOLOGY;ZOOLOGY;4;JIF;0;0,6;0,8;0,458;0,636;0,714;0,3;0,65;0,969;1,345;0,889;0,6,761;2,908;
REC AUST MUS;ZOOLOGY;ZOOLOGY;4;JIF (without self-
citations);0;0,6;0,8;0,333;0,636;0,643;0,2;0,65;0,969;1,241;0,833;0;6,305;2,612;
REV SUISSE ZOOL;ZOOLOGY;ZOOLOGY;4;ALL citations;923;3;12;18;17;22;9;10;7;25;25;775;145;78;
REV SUISSE ZOOL;ZOOLOGY;ZOOLOGY;4;ALL citations (excluding self and
most);585;0;7;6;12;14;6;8;2;16;22;492;93;45;
REV SUISSE ZOOL;ZOOLOGY;ZOOLOGY;4;ZOOTAXA;306;0;3;5;4;3;2;3;6;3;273;33;19;
REV SUISSE ZOOL;ZOOLOGY;ZOOLOGY;4;Self-citation;32;3;2;7;1;4;0;2;3;0;10;19;14;
REV SUISSE ZOOL;ZOOLOGY;ZOOLOGY;4;Ratio of self-
citations;0,034669556;1;0,166666667;0,388888889;0,058823529;0,181818182;0;0;0,285714286;0,12;0,012903226;1,2
01911552;0,796197267;
REV SUISSE
ZOOL;ZOOLOGY;ZOOLOGY;4;JIF;0;0,732;0,63;0,759;0,38;0,431;0,431;0,479;0,382;0,351;0,426;0,4,269;2,631;
REV SUISSE ZOOL;ZOOLOGY;ZOOLOGY;4;JIF (without self-
citations);0;0,512;0,63;0,611;0,3;0,235;0,397;0,394;0,329;0,228;0,377;0;3,501;2,173;
RUSS J HERPETOL;ZOOLOGY;ZOOLOGY;4;ALL citations;271;3;12;16;27;30;21;18;16;18;13;97;171;106;
RUSS J HERPETOL;ZOOLOGY;ZOOLOGY;4;ALL citations (excluding self and
most);200;3;9;13;22;19;10;14;13;13;8;76;121;73;
RUSS J HERPETOL;ZOOLOGY;ZOOLOGY;4;ZOOTAXA;53;0;3;2;2;7;11;2;3;3;4;16;37;25;
RUSS J HERPETOL;ZOOLOGY;ZOOLOGY;4;Self-citation;18;0;0;1;3;4;0;2;0;2;1;5;13;8;
RUSS J HERPETOL;ZOOLOGY;ZOOLOGY;4;Ratio of self-
citations;0,066420664;0;0,0625;0,111111111;0,133333333;0,0,111111111;0,0,111111111;0,076923077;0,051546392;
0,606089744;0,306944444;
RUSS J HERPETOL;ZOOLOGY;ZOOLOGY;4;JIF;0;0,346;0,325;0,407;0,384;0,347;0;0;0;0;0;1,463;1,463;
RUSS J HERPETOL;ZOOLOGY;ZOOLOGY;4;JIF (without self-
citations);0;0,333;0,289;0,346;0,205;0,167;0;0;0;0;0;1,007;1,007;
RUSS J NEMATOL;ZOOLOGY;ZOOLOGY;4;ALL citations;179;0;4;7;5;6;9;9;6;10;13;110;69;31;
RUSS J NEMATOL;ZOOLOGY;ZOOLOGY;4;ALL citations (excluding self and
most);155;0;1;5;4;6;8;8;5;7;11;100;55;24;
RUSS J NEMATOL;ZOOLOGY;ZOOLOGY;4;ZOOTAXA;13;0;1;1;0;0;0;0;2;2;7;6;2;
RUSS J NEMATOL;ZOOLOGY;ZOOLOGY;4;Self-citation;11;0;2;1;1;0;1;1;1;0;3;8;5;
RUSS J NEMATOL;ZOOLOGY;ZOOLOGY;4;Ratio of self-
citations;0,061452514;#DIV/0!;0,5;0,142857143;0,2;0,0,111111111;0,111111111;0,166666667;0,1;0,0,027272727;1,331
746032;0,953968254;
RUSS J
NEMATOL;ZOOLOGY;ZOOLOGY;4;JIF;0;0,393;0,577;0,519;0,533;0,481;0,793;0,294;0,472;0,472;0,5;0,4,641;2,903;
RUSS J NEMATOL;ZOOLOGY;ZOOLOGY;4;JIF (without self-
citations);0;0,286;0,346;0,519;0,4;0,407;0,483;0,176;0,361;0,361;0,412;0,3,465;2,155;
S AM J HERPETOL;ZOOLOGY;ZOOLOGY;2;ALL citations;407;4;30;38;34;15;44;23;45;41;26;107;296;161;
S AM J HERPETOL;ZOOLOGY;ZOOLOGY;2;ALL citations (excluding self and
most);335;3;24;31;27;14;37;19;34;36;23;87;245;133;
S AM J HERPETOL;ZOOLOGY;ZOOLOGY;2;ZOOTAXA;26;0;4;4;1;0;4;3;3;1;0;6;20;13;
S AM J HERPETOL;ZOOLOGY;ZOOLOGY;2;Self-citation;46;1;2;3;6;1;3;1;8;4;3;14;31;15;
S AM J HERPETOL;ZOOLOGY;ZOOLOGY;2;Ratio of self-
citations;0,113022113;0,25;0,066666667;0,078947368;0,176470588;0,066666667;0,068181818;0,043478261;0,1777777
78;0,097560976;0,115384615;0,130841121;0,891134738;0,456933108;
S AM J HERPETOL;ZOOLOGY;ZOOLOGY;2;JIF;0,1,388;1,122;0,596;1,143;0,837;0;0;0;0;0;3,698;3,698;
S AM J HERPETOL;ZOOLOGY;ZOOLOGY;2;JIF (without self-
citations);0;1,286;0,939;0,574;1,102;0,694;0;0;0;0;0;3,309;3,309;
SALAMANDRA;ZOOLOGY;ZOOLOGY;2;ALL citations;406;6;41;54;30;28;25;17;13;15;19;158;242;178;

SALAMANDRA;ZOOLOGY;ZOOLOGY;2;ALL citations (excluding self and
 1 most);343;4;33;48;24;22;24;14;12;12;17;133;206;151;
 SALAMANDRA;ZOOLOGY;ZOOLOGY;2;ZOOTAXA;41;2;2;6;4;4;0;2;1;2;1;17;22;16;
 SALAMANDRA;ZOOLOGY;ZOOLOGY;2;Self-citation;22;0;6;0;2;2;1;1;0;1;1;8;14;11;
 SALAMANDRA;ZOOLOGY;ZOOLOGY;2;Ratio of self-
 citations;0,054187192;0,0,146341463;0,0,066666667;0,071428571;0,04;0,058823529;0,0,066666667;0,052631579;0,05
 0632911;0,502558477;0,324436702;
 SALAMANDRA;ZOOLOGY;ZOOLOGY;2;JIF;0;1,532;1,313;1,46;1,25;1,25;1,1;1,229;1;0;0;0;8,602;6,373;
 SALAMANDRA;ZOOLOGY;ZOOLOGY;2;JIF (without self-
 citations);0;1,435;1,219;1,34;1,136;1,125;1,067;1,086;0,973;0;0;0;7,946;5,887;
 SPIXIANA;ZOOLOGY;ZOOLOGY;4;ALL citations;323;2;16;11;19;10;12;15;9;20;8;201;120;68;
 SPIXIANA;ZOOLOGY;ZOOLOGY;4;ALL citations (excluding self and
 most);239;0;11;8;14;9;10;12;9;16;6;144;95;52;
 SPIXIANA;ZOOLOGY;ZOOLOGY;4;ZOOTAXA;78;2;5;3;3;1;1;3;0;3;2;55;21;13;
 SPIXIANA;ZOOLOGY;ZOOLOGY;4;Self-citation;6;0;0;0;2;0;1;0;0;1;0;2;4;3;
 SPIXIANA;ZOOLOGY;ZOOLOGY;4;Ratio of self-
 citations;0,018575851;0;0;0;0,105263158;0,0,083333333;0;0;0,05;0,009950249;0,238596491;0,188596491;
 SPIXIANA;ZOOLOGY;ZOOLOGY;4;JIF;0;0,659;0,442;0,375;0,784;0,673;0,537;0,553;0,605;0,447;0,205;0,4,621;2,81
 1;
 SPIXIANA;ZOOLOGY;ZOOLOGY;4;JIF (without self-
 citations);0;0,659;0,423;0,357;0,784;0,653;0,463;0,447;0,474;0,368;0,128;0,4,097;2,68;
 STUD NEOTROP FAUNA E;ZOOLOGY;ZOOLOGY;3;ALL
 citations;626;6;19;31;29;10;23;18;19;28;12;431;189;112;
 STUD NEOTROP FAUNA E;ZOOLOGY;ZOOLOGY;3;ALL citations (excluding self and
 most);561;5;16;31;27;8;19;18;19;26;9;383;173;101;
 STUD NEOTROP FAUNA E;ZOOLOGY;ZOOLOGY;3;ZOOTAXA;59;1;3;0;1;1;2;0;0;2;3;46;12;7;
 STUD NEOTROP FAUNA E;ZOOLOGY;ZOOLOGY;3;Self-citation;6;0;0;0;1;1;2;0;0;0;0;2;4;4;
 STUD NEOTROP FAUNA E;ZOOLOGY;ZOOLOGY;3;Ratio of self-
 citations;0,009584665;0;0;0,034482759;0,1;0,086956522;0;0;0;0,004640371;0,22143928;0,22143928;
 STUD NEOTROP FAUNA
 E;ZOOLOGY;ZOOLOGY;3;JIF;0;0,943;0,982;0,722;0,315;0,385;0,796;0,569;0,652;0,357;0,596;0,5,374;3,2;
 STUD NEOTROP FAUNA E;ZOOLOGY;ZOOLOGY;3;JIF (without self-
 citations);0;0,943;0,93;0,685;0,259;0,365;0,735;0,549;0,652;0,286;0,574;0;5,035;2,974;
 TURK J ZOOL;ZOOLOGY;ZOOLOGY;4;ALL citations;1133;11;50;85;111;121;115;63;64;69;26;418;704;482;
 TURK J ZOOL;ZOOLOGY;ZOOLOGY;4;ALL citations (excluding self and
 most);1026;8;43;79;107;111;102;56;53;63;23;381;637;442;
 TURK J ZOOL;ZOOLOGY;ZOOLOGY;4;ZOOTAXA;82;1;6;6;1;10;9;6;9;5;3;26;55;32;
 TURK J ZOOL;ZOOLOGY;ZOOLOGY;4;Self-citation;25;2;1;0;3;0;4;1;2;1;0;11;12;8;
 TURK J ZOOL;ZOOLOGY;ZOOLOGY;4;Ratio of self-
 citations;0,022065313;0,181818182;0,02;0,027027027;0,0,034782609;0,015873016;0,03125;0,014492754;0,0,026315
 789;0,143425405;0,081809636;
 TURK J
 ZOOL;ZOOLOGY;ZOOLOGY;4;JIF;0;0,628;0,607;0,558;0,785;0,88;0,63;0,585;0,414;0,591;0,647;0;5,697;3,46;
 TURK J ZOOL;ZOOLOGY;ZOOLOGY;4;JIF (without self-
 citations);0;0,623;0,579;0,522;0,742;0,73;0,464;0,467;0,361;0,508;0,541;0;4,914;3,037;
 VERTEBR ZOOL;ZOOLOGY;ZOOLOGY;3;ALL citations;208;10;21;21;26;29;25;10;19;8;5;34;164;122;
 VERTEBR ZOOL;ZOOLOGY;ZOOLOGY;3;ALL citations (excluding self and
 most);173;5;16;15;24;25;22;8;18;8;4;28;140;102;
 VERTEBR ZOOL;ZOOLOGY;ZOOLOGY;3;ZOOTAXA;26;4;4;5;1;4;2;1;1;0;0;4;18;16;
 VERTEBR ZOOL;ZOOLOGY;ZOOLOGY;3;Self-citation;9;1;1;1;0;1;1;0;0;1;2;6;4;
 VERTEBR ZOOL;ZOOLOGY;ZOOLOGY;3;Ratio of self-
 citations;0,043269231;0,1;0,047619048;0,047619048;0,038461538;0,0,04;0,1;0;0;2;0,058823529;0,473699634;0,1736
 99634;
 VERTEBR ZOOL;ZOOLOGY;ZOOLOGY;3;JIF;0;1,167;1,282;0,961;1,059;0,722;0,593;1,109;0,86;0;0;0;6,586;4,617;
 VERTEBR ZOOL;ZOOLOGY;ZOOLOGY;3;JIF (without self-
 citations);0;1,111;1,231;0,882;0,784;0,5;0,389;0,652;0,442;0;0;0;4,88;3,786;

ZOOKEYS;ZOOLOGY;Top Ten;3;ALL citations;5138;227;642;547;693;603;548;517;435;665;146;115;4796;3033;
ZOOKEYS;ZOOLOGY;Top Ten;3;ALL citations (excluding self and
most);3727;104;434;377;458;471;388;378;358;578;95;86;3537;2128;
ZOOKEYS;ZOOLOGY;Top Ten;3;ZOOTAXA;816;31;114;102;145;77;112;81;52;55;32;15;770;550;
ZOOKEYS;ZOOLOGY;Top Ten;3;Self-citation;595;92;94;68;90;55;48;58;25;32;19;14;489;355;
ZOOKEYS;ZOOLOGY;Top Ten;3;Ratio of self-
citations;0,115803815;0,405286344;0,146417445;0,124314442;0,12987013;0,091210614;0,087591241;0,112185687;0,0
57471264;0,048120301;0,130136986;0,12173913;0,92731811;0,579403872;
ZOOKEYS;ZOOLOGY;Top Ten;3;JIF;0;1,137;1,143;1,079;1,031;0,938;0,933;0,917;0,864;0,879;0,514;0;8,298;5,124;
ZOOKEYS;ZOOLOGY;Top Ten;3;JIF (without self-
citations);0;0,982;0,963;0,926;0,838;0,759;0,758;0,744;0,669;0,582;0,385;0;6,624;4,244;
ZOO ANZ;ZOOLOGY;ZOOLOGY;2;ALL citations;2084;53;99;136;127;85;55;92;41;45;45;1306;725;502;
ZOO ANZ;ZOOLOGY;ZOOLOGY;2;ALL citations (excluding self and
most);1639;27;90;120;87;74;50;77;29;39;39;1007;605;421;
ZOO ANZ;ZOOLOGY;ZOOLOGY;2;ZOOTAXA;323;4;8;8;10;11;3;4;3;5;3;264;55;40;
ZOO ANZ;ZOOLOGY;ZOOLOGY;2;Self-citation;122;22;1;8;30;0;2;11;9;1;3;35;65;41;
ZOO ANZ;ZOOLOGY;ZOOLOGY;2;Ratio of self-
citations;0,058541267;0,41509434;0,01010101;0,058823529;0,236220472;0,0,036363636;0,119565217;0,219512195;0,
022222222;0,066666667;0,026799387;0,76947495;0,341508648;
ZOO ANZ;ZOOLOGY;ZOOLOGY;2;JIF;0;1,366;1,601;1,345;1,2;1,512;1,483;1,821;1,4;1,415;1,846;0;13,623;7,141;
ZOO ANZ;ZOOLOGY;ZOOLOGY;2;JIF (without self-
citations);0;1,314;1,497;1,269;1,137;1,419;1,414;1,731;1,3;1,341;1,692;0;12,8;6,736;
ZOO J LINN SOC LOND;ZOOLOGY;Top Ten;1;ALL
citations;5613;143;314;296;347;301;281;212;245;307;249;2918;2552;1539;
ZOO J LINN SOC LOND;ZOOLOGY;Top Ten;1;ALL citations (excluding self and
most);5024;110;268;263;302;270;239;193;217;268;231;2663;2251;1342;
ZOO J LINN SOC LOND;ZOOLOGY;Top Ten;1;ZOOTAXA;381;25;36;21;26;19;25;11;19;18;9;172;184;127;
ZOO J LINN SOC LOND;ZOOLOGY;Top Ten;1;Self-citation;208;8;10;12;19;12;17;8;9;21;9;83;117;70;
ZOO J LINN SOC LOND;ZOOLOGY;Top Ten;1;Ratio of self-
citations;0,037056832;0,055944056;0,031847134;0,040540541;0,054755043;0,03986711;0,060498221;0,037735849;0,0
36734694;0,068403909;0,036144578;0,02844414;0,406527078;0,227508048;
ZOO J LINN SOC LOND;ZOOLOGY;Top
Ten;1;JIF;0;2,824;2,909;2,685;2,711;2,316;2,717;2,658;2,583;2,433;2,319;0;23,331;13,338;
ZOO J LINN SOC LOND;ZOOLOGY;Top Ten;1;JIF (without self-
citations);0;2,722;2,791;2,606;2,588;2,164;2,53;2,547;2,471;2,294;2,23;0;22,221;12,679;
ZOO LETT;ZOOLOGY;ZOOLOGY;1;ALL citations;293;17;61;49;66;100;0;0;0;0;0;276;276;
ZOO LETT;ZOOLOGY;ZOOLOGY;1;ALL citations (excluding self and
most);274;15;57;43;63;96;0;0;0;0;0;259;259;
ZOO LETT;ZOOLOGY;ZOOLOGY;1;ZOOTAXA;2;2;0;0;0;0;0;0;0;0;0;0;0;0;
ZOO LETT;ZOOLOGY;ZOOLOGY;1;Self-citation;17;0;4;6;3;4;0;0;0;0;0;0;17;17;
ZOO LETT;ZOOLOGY;ZOOLOGY;1;Ratio of self-
citations;0,058020478;0,0,06557377;0,12244898;0,045454545;0,04;#DIV/0!;#DIV/0!;#DIV/0!;#DIV/0!;#DIV/0!;#DIV/
0!;#DIV/0!;#DIV/0!;
ZOO LETT;ZOOLOGY;ZOOLOGY;1;JIF;0;2,075;2,064;2,9;0;0;0;0;0;0;0;0;4,964;4,964;
ZOO LETT;ZOOLOGY;ZOOLOGY;1;JIF (without self-citations);0;1,887;1,83;2,85;0;0;0;0;0;0;0;4,68;4,68;
ZOO MIDDLE EAST;ZOOLOGY;ZOOLOGY;4;ALL citations;490;11;15;26;41;25;54;20;28;30;27;213;266;161;
ZOO MIDDLE EAST;ZOOLOGY;ZOOLOGY;4;ALL citations (excluding self and
most);440;7;13;19;36;20;48;20;25;27;26;199;234;136;
ZOO MIDDLE EAST;ZOOLOGY;ZOOLOGY;4;ZOOTAXA;48;4;1;7;5;5;6;0;2;3;1;14;30;24;
ZOO MIDDLE EAST;ZOOLOGY;ZOOLOGY;4;Self-citation;2;0;1;0;0;0;0;1;0;0;0;2;1;
ZOO MIDDLE EAST;ZOOLOGY;ZOOLOGY;4;Ratio of self-
citations;0,004081633;0,0,066666667;0;0;0;0;0,035714286;0;0;0,102380952;0,066666667;
ZOO MIDDLE
EAST;ZOOLOGY;ZOOLOGY;4;JIF;0;0,456;0,701;0,528;0,525;0,628;0,411;0,524;0,434;0,49;0,412;0;4,653;2,793;

ZOOL MIDDLE EAST;ZOOLOGY;ZOOLOGY;4;JIF (without self-citations);0;0;4;0,629;0,472;0,455;0,543;0,355;0,46;0,381;0,392;0,313;0;4;2,454;

ZOOL RES;ZOOLOGY;ZOOLOGY;1;ALL citations;836;54;79;103;78;52;56;44;31;29;39;271;511;368;

ZOOL RES;ZOOLOGY;ZOOLOGY;1;ALL citations (excluding self and most);688;31;60;80;55;44;50;41;30;27;38;232;425;289;

ZOOL RES;ZOOLOGY;ZOOLOGY;1;ZOOTAXA;67;6;7;13;7;1;1;2;1;1;0;28;33;29;

ZOOL RES;ZOOLOGY;ZOOLOGY;1;Self-citation;81;17;12;10;16;7;5;1;0;1;1;11;53;50;

ZOOL RES;ZOOLOGY;ZOOLOGY;1;Ratio of self-citations;0,096889952;0,314814815;0,151898734;0,097087379;0,205128205;0,134615385;0,089285714;0,022727273;0,034482759;0,025641026;0,040590406;0,760866474;0,678015417;

ZOOL RES;ZOOLOGY;ZOOLOGY;1;JIF;0;2,638;1,556;0;0;0;0;0;0;0;1,556;1,556;

ZOOL RES;ZOOLOGY;ZOOLOGY;1;JIF (without self-citations);0;2,319;1,361;0;0;0;0;0;0;0;1,361;1,361;

ZOOL SCI;ZOOLOGY;ZOOLOGY;3;ALL citations;2525;12;44;69;81;81;108;136;122;120;119;1633;880;383;

ZOOL SCI;ZOOLOGY;ZOOLOGY;3;ALL citations (excluding self and most);2383;8;35;62;74;70;103;132;120;116;112;1551;824;344;

ZOOL SCI;ZOOLOGY;ZOOLOGY;3;ZOOTAXA;80;1;6;4;2;4;2;2;1;1;3;54;25;18;

ZOOL SCI;ZOOLOGY;ZOOLOGY;3;Self-citation;62;3;3;3;5;7;3;2;1;3;4;28;31;21;

ZOOL SCI;ZOOLOGY;ZOOLOGY;3;Ratio of self-citations;0,024554455;0,25;0,068181818;0,043478261;0,061728395;0,086419753;0,027777778;0,014705882;0,008196721;0,025;0,033613445;0,017146356;0,369102054;0,287586005;

ZOOL SCI;ZOOLOGY;ZOOLOGY;3;JIF;0;0,843;0,846;0,906;0,755;0,814;0,857;0,876;1,076;0,952;1,087;0;8,169;4,178;

ZOOL SCI;ZOOLOGY;ZOOLOGY;3;JIF (without self-citations);0;0,799;0,832;0,881;0,703;0,775;0,812;0,784;0,948;0,888;0,996;0;7,619;4,003;

ZOOL SCR;ZOOLOGY;ZOOLOGY;1;ALL citations;2401;53;116;186;167;128;150;125;101;127;99;1149;1199;747;

ZOOL SCR;ZOOLOGY;ZOOLOGY;1;ALL citations (excluding self and most);2127;43;104;170;155;113;133;109;84;113;89;1014;1070;675;

ZOOL SCR;ZOOLOGY;ZOOLOGY;1;ZOOTAXA;236;8;10;10;9;11;15;12;14;14;8;125;103;55;

ZOOL SCR;ZOOLOGY;ZOOLOGY;1;Self-citation;38;2;2;6;3;4;2;4;3;0;2;10;26;17;

ZOOL SCR;ZOOLOGY;ZOOLOGY;1;Ratio of self-citations;0,015826739;0,037735849;0,017241379;0,032258065;0,017964072;0,03125;0,013333333;0,032;0,02970297;0,02020202;0,00870322;0,19395184;0,112046849;

ZOOL SCR;ZOOLOGY;ZOOLOGY;1;JIF;0;2,603;2,609;3,057;2,837;2,733;3,224;2,922;2,793;2,913;3,091;0;26,179;14,46;

ZOOL SCR;ZOOLOGY;ZOOLOGY;1;JIF (without self-citations);0;2,534;2,478;2,848;2,765;2,644;3,035;2,844;2,663;2,75;2,943;0;24,97;13,77;

ZOOL STUD;ZOOLOGY;ZOOLOGY;2;ALL citations;1330;15;78;49;49;89;80;67;121;77;90;615;700;345;

ZOOL STUD;ZOOLOGY;ZOOLOGY;2;ALL citations (excluding self and most);1196;5;43;38;44;79;75;66;112;74;88;572;619;279;

ZOOL STUD;ZOOLOGY;ZOOLOGY;2;ZOOTAXA;60;0;5;2;1;6;5;0;2;1;0;38;22;19;

ZOOL STUD;ZOOLOGY;ZOOLOGY;2;Self-citation;74;10;30;9;4;4;0;1;7;2;2;5;59;47;

ZOOL STUD;ZOOLOGY;ZOOLOGY;2;Ratio of self-citations;0,055639098;0,666666667;0,384615385;0,183673469;0,081632653;0,04494382;0,014925373;0,05785124;0,025974026;0,022222222;0,008130081;0,815838188;0,694865327;

ZOOL STUD;ZOOLOGY;ZOOLOGY;2;JIF;0;1,257;0,726;1,054;1,008;0,885;0,776;1,014;1,261;0,975;1,046;0;8,745;4,449;

ZOOL STUD;ZOOLOGY;ZOOLOGY;2;JIF (without self-citations);0;0,871;0,674;0,911;1,008;0,862;0,699;0,716;0,815;0,755;0,719;0;7,159;4,154;

ZOOL ZH;ZOOLOGY;ZOOLOGY;4;ALL citations;1479;9;28;49;42;40;29;31;29;37;31;1154;316;188;

ZOOL ZH;ZOOLOGY;ZOOLOGY;4;ALL citations (excluding self and most);1176;9;15;29;30;30;24;23;23;28;18;947;220;128;

ZOOL ZH;ZOOLOGY;ZOOLOGY;4;ZOOTAXA;170;0;3;3;5;2;1;4;1;1;4;146;24;14;

ZOOL ZH;ZOOLOGY;ZOOLOGY;4;Self-citation;133;0;10;17;7;8;4;4;5;8;9;61;72;46;

ZOOL ZH;ZOOLOGY;ZOOLOGY;4;Ratio of self-citations;0,089925625;0,0357142857;0,346938776;0,166666667;0,2;0,137931034;0,129032258;0,172413793;0,216216216;0,290322581;0,052859619;2,016664182;1,208679334;

ZOOL ZH;ZOOLOGY;ZOOLOGY;4;JIF;0;0,297;0,291;0,226;0,091;0,142;0,14;0,194;0,253;0,305;0,265;0,1,907;0,89;
ZOOLOGIA CURITIBA;ZOOLOGY;ZOOLOGY;4;JIF (without self-
citations);0;0,193;0,189;0,14;0,091;0,142;0,133;0,128;0,154;0,208;0,183;0,1,368;0,695;
ZOOLOGIA CURITIBA;ZOOLOGY;ZOOLOGY;3;ALL citations;755;4;33;42;64;61;73;89;84;98;121;86;665;273;
ZOOLOGIA CURITIBA;ZOOLOGY;ZOOLOGY;3;ALL citations (excluding self and
most);672;3;19;36;55;59;66;80;66;92;115;81;588;235;
ZOOLOGIA CURITIBA;ZOOLOGY;ZOOLOGY;3;ZOOTAXA;62;1;8;4;6;1;5;7;16;5;4;5;56;24;
ZOOLOGIA CURITIBA;ZOOLOGY;ZOOLOGY;3;Self-citation;21;0;6;2;3;1;2;2;2;1;2;0;21;14;
ZOOLOGIA CURITIBA;ZOOLOGY;ZOOLOGY;3;Ratio of self-
citations;0,02781457;0,0,181818182;0,047619048;0,046875;0,016393443;0,02739726;0,02247191;0,023809524;0,0102
04082;0,016528926;0,0,393117374;0,320102932;
ZOOLOGIA
CURITIBA;ZOOLOGY;ZOOLOGY;3;JIF;0;0,852;0,743;0,723;0,642;0,584;0,538;0,652;0,658;0,587;0,373;0,5,5;3,23;
ZOOLOGIA CURITIBA;ZOOLOGY;ZOOLOGY;3;JIF (without self-
citations);0;0,761;0,703;0,692;0,62;0,547;0,5;0,601;0,61;0,525;0,324;0,5,122;3,062;
ZOOLOGY;ZOOLOGY;ZOOLOGY;2;ALL citations;1735;13;100;127;169;66;86;67;106;98;87;816;906;548;
ZOOLOGY;ZOOLOGY;ZOOLOGY;2;ALL citations (excluding self and
most);1615;12;97;127;166;64;85;67;105;97;85;710;893;539;
ZOOLOGY;ZOOLOGY;ZOOLOGY;2;ZOOTAXA;101;0;0;0;0;0;0;0;1;100;1;0;
ZOOLOGY;ZOOLOGY;ZOOLOGY;2;Self-citation;19;1;3;0;3;2;1;0;1;1;1;6;12;9;
ZOOLOGY;ZOOLOGY;ZOOLOGY;2;Ratio of self-
citations;0,010951009;0,076923077;0,03;0,0,017751479;0,03030303;0,011627907;0,0,009433962;0,010204082;0,01149
4253;0,007352941;0,120814713;0,089682417;
ZOOLOGY;ZOOLOGY;ZOOLOGY;2;JIF;0;1,681;1,779;1,938;1,51;1,691;1,67;1,596;1,471;1,5;1,651;0;14,806;8,588;
ZOOLOGY;ZOOLOGY;ZOOLOGY;2;JIF (without self-
citations);0;1,659;1,713;1,884;1,439;1,628;1,591;1,562;1,414;1,452;1,53;0;14,213;8,255;
ZOOSYST EVOL;ZOOLOGY;ZOOLOGY;3;ALL citations;215;8;33;23;14;13;19;19;14;9;19;44;163;102;
ZOOSYST EVOL;ZOOLOGY;ZOOLOGY;3;ALL citations (excluding self and
most);154;4;19;18;9;12;13;13;13;7;14;32;118;71;
ZOOSYST EVOL;ZOOLOGY;ZOOLOGY;3;ZOOTAXA;38;2;8;3;4;0;3;5;0;2;2;9;27;18;
ZOOSYST EVOL;ZOOLOGY;ZOOLOGY;3;Self-citation;23;2;6;2;1;1;3;1;1;0;3;3;18;13;
ZOOSYST EVOL;ZOOLOGY;ZOOLOGY;3;Ratio of self-
citations;0,106976744;0,25;0,181818182;0,086956522;0,071428571;0,076923077;0,157894737;0,052631579;0,0714285
71;0,0,157894737;0,068181818;0,856975976;0,575021089;
ZOOSYST EVOL;ZOOLOGY;ZOOLOGY;3;JIF;0;0,903;1,209;0,862;0,931;0,8;0;0;0;0;0;3,802;3,802;
ZOOSYST EVOL;ZOOLOGY;ZOOLOGY;3;JIF (without self-
citations);0;0,774;1,047;0,759;0,931;0,743;0;0;0;0;0;3,48;3,48;
ACAROLOGIA;ENTOMOLOGY;ENTOMOLOGY;3;ALL
citations;1054;13;61;56;34;30;26;46;27;17;37;707;334;207;
ACAROLOGIA;ENTOMOLOGY;ENTOMOLOGY;3;ALL citations (excluding self and
most);714;9;46;44;16;16;13;40;22;15;29;464;241;135;
ACAROLOGIA;ENTOMOLOGY;ENTOMOLOGY;3;ZOOTAXA;274;4;10;8;15;12;8;4;5;2;8;198;72;53;
ACAROLOGIA;ENTOMOLOGY;ENTOMOLOGY;3;Self-citation;66;0;5;4;3;2;5;2;0;0;0;45;21;19;
ACAROLOGIA;ENTOMOLOGY;ENTOMOLOGY;3;Ratio of self-
citations;0,062618596;0,0,081967213;0,071428571;0,088235294;0,066666667;0,192307692;0,043478261;0;0;0,06364
9222;0,544083699;0,500605438;
ACAROLOGIA;ENTOMOLOGY;ENTOMOLOGY;3;JIF;0;0,842;1,047;0,866;0,667;1,103;1;0;0;0;0;4,683;4,683;
ACAROLOGIA;ENTOMOLOGY;ENTOMOLOGY;3;JIF (without self-
citations);0;0,777;0,879;0,768;0,623;0,941;0,806;0;0;0;0;4,017;4,017;
ACTA ENT MUS NAT PRA;ENTOMOLOGY;ENTOMOLOGY;4;ALL
citations;460;15;31;40;24;30;49;23;73;28;13;134;311;174;
ACTA ENT MUS NAT PRA;ENTOMOLOGY;ENTOMOLOGY;4;ALL citations (excluding self and
most);243;7;20;15;16;15;24;11;21;19;8;87;149;90;

ACTA ENT MUS NAT

PRA;ENTOMOLOGY;ENTOMOLOGY;4;ZOOTAXA;187;5;9;23;5;13;22;11;49;7;5;38;144;72;

ACTA ENT MUS NAT PRA;ENTOMOLOGY;ENTOMOLOGY;4;Self-citation;30;3;2;2;3;2;3;1;3;2;0;9;18;12;

ACTA ENT MUS NAT PRA;ENTOMOLOGY;ENTOMOLOGY;4;Ratio of self-

citations;0,065217391;0,2;0,064516129;0,05;0,125;0,066666667;0,06122449;0,043478261;0,04109589;0,071428571;0;0,067164179;0,523410008;0,367407285;

ACTA ENT MUS NAT

PRA;ENTOMOLOGY;ENTOMOLOGY;4;JIF;0;0,623;0,487;0,561;0,632;0,617;0,659;0,574;0,963;0,721;0;0;5,214;2,956;

ACTA ENT MUS NAT PRA;ENTOMOLOGY;ENTOMOLOGY;4;JIF (without self-

citations);0;0,588;0,389;0,347;0,436;0,458;0,467;0,377;0,646;0,395;0;0;3,515;2,097;

AFR ENTOMOL;ENTOMOLOGY;ENTOMOLOGY;4;ALL

citations;688;4;44;36;35;64;97;26;32;100;23;227;457;276;

AFR ENTOMOL;ENTOMOLOGY;ENTOMOLOGY;4;ALL citations (excluding self and most);634;2;43;34;34;57;94;23;31;95;22;199;433;262;

AFR ENTOMOL;ENTOMOLOGY;ENTOMOLOGY;4;ZOOTAXA;25;0;0;1;0;0;1;0;0;2;0;21;4;2;

AFR ENTOMOL;ENTOMOLOGY;ENTOMOLOGY;4;Self-citation;29;2;1;1;1;7;2;3;1;3;1;7;20;12;

AFR ENTOMOL;ENTOMOLOGY;ENTOMOLOGY;4;Ratio of self-

citations;0,042151163;0,5;0,022727273;0,027777778;0,028571429;0,109375;0,020618557;0,115384615;0,03125;0,03;0,043478261;0,030837004;0,429182912;0,209070036;

AFR

ENTOMOL;ENTOMOLOGY;ENTOMOLOGY;4;JIF;0;0,65;0,536;0,508;0,813;0,521;0,365;0,772;0,969;0,47;0,338;0;5,292;2,743;

AFR ENTOMOL;ENTOMOLOGY;ENTOMOLOGY;4;JIF (without self-

citations);0;0,634;0,48;0,434;0,613;0,432;0,24;0,667;0,796;0,364;0,265;0;4,291;2,199;

ANN ENTOMOL SOC AM;ENTOMOLOGY;ENTOMOLOGY;2;ALL

citations;6051;65;42;112;141;228;174;106;119;165;164;4735;1251;697;

ANN ENTOMOL SOC AM;ENTOMOLOGY;ENTOMOLOGY;2;ALL citations (excluding self and most);5552;30;35;110;131;218;164;99;114;155;156;4340;1182;658;

ANN ENTOMOL SOC AM;ENTOMOLOGY;ENTOMOLOGY;2;ZOOTAXA;397;0;4;0;2;9;8;4;9;6;351;46;23;

ANN ENTOMOL SOC AM;ENTOMOLOGY;ENTOMOLOGY;2;Self-citation;102;35;3;2;8;1;2;3;1;1;2;44;23;16;

ANN ENTOMOL SOC AM;ENTOMOLOGY;ENTOMOLOGY;2;Ratio of self-

citations;0,016856718;0,538461538;0,071428571;0,017857143;0,056737589;0,004385965;0,011494253;0,028301887;0,008403361;0,006060606;0,012195122;0,009292503;0,216864497;0,161903521;

ANN ENTOMOL SOC

AM;ENTOMOLOGY;ENTOMOLOGY;2;JIF;0;1,51;1,665;1,558;1,222;1,14;1,19;1,174;1,196;1,317;1,031;0;11,493;6,775;

ANN ENTOMOL SOC AM;ENTOMOLOGY;ENTOMOLOGY;2;JIF (without self-

citations);0;1,461;1,632;1,482;1,128;1,07;1,025;1,093;1,063;1,189;0,943;0;10,625;6,337;

ANN SOC ENTOMOL FR;ENTOMOLOGY;ENTOMOLOGY;4;ALL

citations;1607;3;19;27;13;28;30;21;20;35;46;1365;239;117;

ANN SOC ENTOMOL FR;ENTOMOLOGY;ENTOMOLOGY;4;ALL citations (excluding self and most);1034;2;11;12;10;25;24;17;17;26;38;852;180;82;

ANN SOC ENTOMOL FR;ENTOMOLOGY;ENTOMOLOGY;4;ZOOTAXA;524;1;4;9;1;2;4;3;1;8;7;484;39;20;

ANN SOC ENTOMOL FR;ENTOMOLOGY;ENTOMOLOGY;4;Self-citation;49;0;4;6;2;1;2;1;2;1;1;29;20;15;

ANN SOC ENTOMOL FR;ENTOMOLOGY;ENTOMOLOGY;4;Ratio of self-

citations;0,030491599;0,0,210526316;0,222222222;0,153846154;0,035714286;0,066666667;0,047619048;0,1;0,028571429;0,02173913;0,021245421;0,886905251;0,688975644;

ANN SOC ENTOMOL

FR;ENTOMOLOGY;ENTOMOLOGY;4;JIF;0;0,657;0,864;0,515;0,513;0,575;0,513;0,539;0,529;0,537;0,698;0;5,283;2,98;

ANN SOC ENTOMOL FR;ENTOMOLOGY;ENTOMOLOGY;4;JIF (without self-

citations);0;0,514;0,636;0,485;0,461;0,538;0,461;0,539;0,51;0,516;0,407;0;4,553;2,581;

AQUAT INSECT;ENTOMOLOGY;ENTOMOLOGY;4;ALL citations;387;2;8;8;14;0;19;11;19;15;11;280;105;49;

AQUAT INSECT;ENTOMOLOGY;ENTOMOLOGY;4;ALL citations (excluding self and most);264;1;5;8;10;0;14;10;16;11;8;181;82;37;

AQUAT INSECT;ENTOMOLOGY;ENTOMOLOGY;4;ZOOTAXA;108;1;2;0;2;0;4;1;3;2;3;90;17;8;
AQUAT INSECT;ENTOMOLOGY;ENTOMOLOGY;4;Self-citation;15;0;1;0;2;0;1;0;0;2;0;9;6;4;
AQUAT INSECT;ENTOMOLOGY;ENTOMOLOGY;4;Ratio of self-
citations;0,03875969;0;0,125;0;0,142857143;#DIV/0!;0,052631579;0;0,133333333;0;0,032142857;#DIV/0!;#DIV/0!;
AQUAT
INSECT;ENTOMOLOGY;ENTOMOLOGY;4;JIF;0;0,444;0,537;0,583;0,524;0,129;0,395;0,433;0,358;0,496;0,325;0,3,
78;2,168;
AQUAT INSECT;ENTOMOLOGY;ENTOMOLOGY;4;JIF (without self-
citations);0;0,417;0,512;0,458;0,476;0,129;0,395;0,433;0,321;0,46;0,299;0,3,483;1,97;
ARTHROPODA SEL;ENTOMOLOGY;ENTOMOLOGY;4;ALL
citations;338;6;20;30;30;25;18;33;12;13;8;143;189;123;
ARTHROPODA SEL;ENTOMOLOGY;ENTOMOLOGY;4;ALL citations (excluding self and
most);126;0;5;10;8;20;11;8;6;4;5;49;77;54;
ARTHROPODA SEL;ENTOMOLOGY;ENTOMOLOGY;4;Self-citation;156;4;12;18;16;3;6;18;3;9;2;65;87;55;
ARTHROPODA SEL;ENTOMOLOGY;ENTOMOLOGY;4;ZOOTAXA;56;2;3;2;6;2;1;7;3;0;1;29;25;14;
ARTHROPODA SEL;ENTOMOLOGY;ENTOMOLOGY;4;Ratio of self-
citations;0,165680473;0,333333333;0,15;0,066666667;0,2;0,08;0,055555556;0,212121212;0,25;0;0,125;0,202797203;1,
139343434;0,552222222;
ARTHROPODA SEL;ENTOMOLOGY;ENTOMOLOGY;4;JIF;0;0,588;0,951;0,633;0;0;0;0;0;0;1,584;1,584;
ARTHROPODA SEL;ENTOMOLOGY;ENTOMOLOGY;4;JIF (without self-
citations);0;0,235;0,463;0,595;0;0;0;0;0;0;1,058;1,058;
ARTHROPOD SYST PHYLO;ENTOMOLOGY;ENTOMOLOGY;2;ALL
citations;342;6;34;43;29;43;35;11;11;12;22;96;240;184;
ARTHROPOD SYST PHYLO;ENTOMOLOGY;ENTOMOLOGY;2;ALL citations (excluding self and
most);292;5;27;33;26;37;31;9;11;8;17;88;199;154;
ARTHROPOD SYST PHYLO;ENTOMOLOGY;ENTOMOLOGY;2;ZOOTAXA;43;1;5;9;0;6;4;2;0;3;5;8;34;24;
ARTHROPOD SYST PHYLO;ENTOMOLOGY;ENTOMOLOGY;2;Self-citation;7;0;2;1;3;0;0;0;0;1;0;0;7;6;
ARTHROPOD SYST PHYLO;ENTOMOLOGY;ENTOMOLOGY;2;Ratio of self-
citations;0,020467836;0,058823529;0,023255814;0,103448276;0;0;0;0,083333333;0;0,268860953;0,185527619;
ARTHROPOD SYST
PHYLO;ENTOMOLOGY;ENTOMOLOGY;2;JIF;0;1,51;1,175;1,703;2,357;1,655;1,368;1,062;2,318;0;0;0;11,638;8,25
8;
ARTHROPOD SYST PHYLO;ENTOMOLOGY;ENTOMOLOGY;2;JIF (without self-
citations);0;1,451;1,15;1,622;2,262;1,483;1,158;1,2,136;0;0;0;10,811;7,675;
ASIAN MYRMECOL;ENTOMOLOGY;ENTOMOLOGY;4;ALL citations;78;0;7;11;9;7;8;15;0;5;7;9;69;42;
ASIAN MYRMECOL;ENTOMOLOGY;ENTOMOLOGY;4;ALL citations (excluding self and
most);73;0;7;10;8;6;7;15;0;4;7;9;64;38;
ASIAN MYRMECOL;ENTOMOLOGY;ENTOMOLOGY;4;ZOOTAXA;5;0;0;1;1;1;0;0;1;0;0;5;4;
ASIAN MYRMECOL;ENTOMOLOGY;ENTOMOLOGY;4;Self-citation;0;0;0;0;0;0;0;0;0;0;0;0;0;
ASIAN MYRMECOL;ENTOMOLOGY;ENTOMOLOGY;4;Ratio of self-
citations;0;#DIV/0!;0;0;0;0;0;#DIV/0!;0;0;0;#DIV/0!;0;
ASIAN
MYRMECOL;ENTOMOLOGY;ENTOMOLOGY;4;JIF;0;0,621;0,429;0,724;0,8;1,1;0,889;0,625;0,25;0,625;0,867;0;6,
309;3,942;
ASIAN MYRMECOL;ENTOMOLOGY;ENTOMOLOGY;4;JIF (without self-
citations);0;0,621;0,429;0,483;0,6;1;0,667;0,125;0,25;0,5;0,733;0,4,787;3,179;
AUSTRAL ENTOMOL;ENTOMOLOGY;ENTOMOLOGY;2;ALL
citations;377;45;79;56;68;65;64;0;0;0;0;332;332;
AUSTRAL ENTOMOL;ENTOMOLOGY;ENTOMOLOGY;2;ALL citations (excluding self and
most);298;27;58;44;58;54;57;0;0;0;0;271;271;
AUSTRAL ENTOMOL;ENTOMOLOGY;ENTOMOLOGY;2;ZOOTAXA;30;10;4;3;4;6;3;0;0;0;0;20;20;
AUSTRAL ENTOMOL;ENTOMOLOGY;ENTOMOLOGY;2;Self-citation;49;8;17;9;6;5;4;0;0;0;0;41;41;

AUSTRAL ENTOMOL;ENTOMOLOGY;ENTOMOLOGY;2;Ratio of self-citations;0,129973475;0,177777778;0,215189873;0,160714286;0,088235294;0,076923077;0,0625;#DIV/0!;#DIV/0!;#DIV/0!;#DIV/0!;#DIV/0!;#DIV/0!;#DIV/0!;#DIV/0!;0,60356253;

AUSTRAL ENTOMOL;ENTOMOLOGY;ENTOMOLOGY;2;JIF;0;1,552;1,769;1,312;1,341;1,114;0;0;0;0;0;5,536;5,536;

AUSTRAL ENTOMOL;ENTOMOLOGY;ENTOMOLOGY;2;JIF (without self-citations);0;1,253;1,264;1,161;1,121;1,023;0;0;0;0;0;4,569;4,569;

B INSECTOL;ENTOMOLOGY;ENTOMOLOGY;3;ALL citations;892;5;31;57;41;83;70;63;118;106;32;286;601;282;

B INSECTOL;ENTOMOLOGY;ENTOMOLOGY;3;ALL citations (excluding self and most);852;4;23;52;40;79;64;63;115;105;32;275;573;258;

B INSECTOL;ENTOMOLOGY;ENTOMOLOGY;3;ZOOTAXA;9;1;1;0;0;1;0;3;0;0;2;6;3;

B INSECTOL;ENTOMOLOGY;ENTOMOLOGY;3;Self-citation;31;0;7;4;1;4;5;0;0;1;0;9;22;21;

B INSECTOL;ENTOMOLOGY;ENTOMOLOGY;3;Ratio of self-citations;0,034753363;0,0,225806452;0,070175439;0,024390244;0,048192771;0,071428571;0;0;0,009433962;0,0,031468531;0,449427439;0,439993477;

B INSECTOL;ENTOMOLOGY;ENTOMOLOGY;3;JIF;0;1,1;1,062;1,088;1,051;1,075;1,494;0,722;0,375;0,592;0,371;0;7,83;5,77;

B INSECTOL;ENTOMOLOGY;ENTOMOLOGY;3;JIF (without self-citations);0;0,963;0,963;0,938;0,899;0,938;1,325;0,61;0,31;0,539;0,311;0;6,833;5,063;

CAN ENTOMOL;ENTOMOLOGY;ENTOMOLOGY;2;ALL citations;3545;26;56;90;105;90;65;58;63;48;65;2879;640;406;

CAN ENTOMOL;ENTOMOLOGY;ENTOMOLOGY;2;ALL citations (excluding self and most);3042;17;44;76;91;84;55;54;54;44;61;2462;563;350;

CAN ENTOMOL;ENTOMOLOGY;ENTOMOLOGY;2;ZOOTAXA;368;3;6;6;6;4;7;1;5;3;0;327;38;29;

CAN ENTOMOL;ENTOMOLOGY;ENTOMOLOGY;2;Self-citation;135;6;6;8;8;2;3;3;4;1;4;90;39;27;

CAN ENTOMOL;ENTOMOLOGY;ENTOMOLOGY;2;Ratio of self-citations;0,038081805;0,230769231;0,107142857;0,088888889;0,076190476;0,022222222;0,046153846;0,051724138;0,063492063;0,020833333;0,061538462;0,031260854;0,538186287;0,340598291;

CAN ENTOMOL;ENTOMOLOGY;ENTOMOLOGY;2;JIF;0;1,177;1,212;0,948;1,264;1;0,837;0,667;0,901;0,848;0,694;0;8,371;5,261;

CAN ENTOMOL;ENTOMOLOGY;ENTOMOLOGY;2;JIF (without self-citations);0;1,065;1,073;0,83;1,14;0,851;0,76;0,61;0,829;0,768;0,643;0;7,504;4,654;

COLEOPTS BULL;ENTOMOLOGY;ENTOMOLOGY;4;ALL citations;981;19;53;40;45;70;27;38;44;27;21;597;365;235;

COLEOPTS BULL;ENTOMOLOGY;ENTOMOLOGY;4;ALL citations (excluding self and most);617;7;21;23;28;46;18;30;33;20;17;374;236;136;

COLEOPTS BULL;ENTOMOLOGY;ENTOMOLOGY;4;ZOOTAXA;173;2;13;4;6;10;4;3;7;2;2;120;51;37;

COLEOPTS BULL;ENTOMOLOGY;ENTOMOLOGY;4;Self-citation;191;10;19;13;11;14;5;5;4;5;2;103;78;62;

COLEOPTS BULL;ENTOMOLOGY;ENTOMOLOGY;4;Ratio of self-citations;0,194699286;0,526315789;0,358490566;0,325;0,244444444;0,2;0,185185185;0,131578947;0,090909091;0,185185185;0,095238095;0,172529313;1,816031514;1,313120196;

COLEOPTS BULL;ENTOMOLOGY;ENTOMOLOGY;4;JIF;0;0,669;0,697;0,632;0,496;0,575;0,495;0,726;0,398;0,404;0,528;0;4,951;2,895;

COLEOPTS BULL;ENTOMOLOGY;ENTOMOLOGY;4;JIF (without self-citations);0;0,439;0,492;0,458;0,41;0,283;0,379;0,56;0,398;0,303;0,416;0;3,699;2,022;

DEUT ENTOMOL Z;ENTOMOLOGY;ENTOMOLOGY;3;ALL citations;660;1;6;8;8;15;7;8;21;7;9;570;89;44;

DEUT ENTOMOL Z;ENTOMOLOGY;ENTOMOLOGY;3;ALL citations (excluding self and most);389;1;3;6;5;11;4;6;16;4;7;326;62;29;

DEUT ENTOMOL Z;ENTOMOLOGY;ENTOMOLOGY;3;ZOOTAXA;259;0;0;1;3;3;2;4;3;2;238;21;10;

DEUT ENTOMOL Z;ENTOMOLOGY;ENTOMOLOGY;3;Self-citation;12;0;3;1;0;1;0;0;1;0;0;6;6;5;

DEUT ENTOMOL Z;ENTOMOLOGY;ENTOMOLOGY;3;Ratio of self-citations;0,018181818;0,0,5;0,125;0,0,066666667;0;0,047619048;0;0,010526316;0,739285714;0,691666667;

DEUT ENTOMOL

Z;ENTOMOLOGY;ENTOMOLOGY;3;JIF;0;0,778;0,48;0,879;0,697;0,479;0,491;0,732;0,73;0,522;0,529;0,5,539;3,026

; DEUT ENTOMOL Z;ENTOMOLOGY;ENTOMOLOGY;3;JIF (without self-citations);0;0,556;0,4;0,848;0,667;0,396;0,491;0,585;0,622;0,478;0,431;0,4,918;2,802;

ENTOMOL AM NY;ENTOMOLOGY;ENTOMOLOGY;4;ALL citations;49;0;0;7;8;4;5;3;10;4;5;3;46;24;

ENTOMOL AM NY;ENTOMOLOGY;ENTOMOLOGY;4;ALL citations (excluding self and most);35;0;0;3;5;2;4;3;7;4;4;3;32;14;

ENTOMOL AM NY;ENTOMOLOGY;ENTOMOLOGY;4;ZOOTAXA;13;0;0;4;3;2;0;0;3;0;1;0;13;9;

ENTOMOL AM NY;ENTOMOLOGY;ENTOMOLOGY;4;Self-citation;1;0;0;0;0;1;0;0;0;0;1;1;

ENTOMOL AM NY;ENTOMOLOGY;ENTOMOLOGY;4;Ratio of self-citations;0,020408163;#DIV/0!;#DIV/0!;0;0;0;0,2;0;0;0;0;#DIV/0!;#DIV/0!;

ENTOMOL AM

NY;ENTOMOLOGY;ENTOMOLOGY;4;JIF;0;0,259;0,5;1,143;0,75;0,231;0,167;0,459;0,857;0,519;0,133;0,4,759;2,79

1; ENTOMOL AM NY;ENTOMOLOGY;ENTOMOLOGY;4;JIF (without self-citations);0;0,259;0,5;0,371;0,75;0,231;0,167;0,459;0,619;0,481;0,067;0,3,645;2,019;

ENTOMOL FENNICA;ENTOMOLOGY;ENTOMOLOGY;4;ALL

citations;291;1;6;10;18;10;11;12;8;14;9;192;98;55;

ENTOMOL FENNICA;ENTOMOLOGY;ENTOMOLOGY;4;ALL citations (excluding self and most);226;0;5;4;16;7;8;10;8;12;7;149;77;40;

ENTOMOL FENNICA;ENTOMOLOGY;ENTOMOLOGY;4;ZOOTAXA;54;1;0;4;1;1;2;2;0;2;1;40;13;8;

ENTOMOL FENNICA;ENTOMOLOGY;ENTOMOLOGY;4;Self-citation;11;0;1;2;1;2;1;0;0;0;1;3;8;7;

ENTOMOL FENNICA;ENTOMOLOGY;ENTOMOLOGY;4;Ratio of self-citations;0,037800687;0,0,166666667;0,2;0,055555556;0,2;0,090909091;0;0;0;0,111111111;0,015625;0,824242424;0,71

3131313;

ENTOMOL

FENNICA;ENTOMOLOGY;ENTOMOLOGY;4;JIF;0;0,372;0,658;0,256;0,3;0,353;0,377;0,441;0,41;0,333;0,321;0;3,4

49;1,944; ENTOMOL FENNICA;ENTOMOLOGY;ENTOMOLOGY;4;JIF (without self-citations);0;0,302;0,632;0,256;0,275;0,333;0,328;0,382;0,361;0,298;0,286;0,3,151;1,824;

ENTOMOL NEWS;ENTOMOLOGY;ENTOMOLOGY;4;ALL citations;743;10;19;19;16;28;7;8;15;1;27;593;140;89;

ENTOMOL NEWS;ENTOMOLOGY;ENTOMOLOGY;4;ALL citations (excluding self and most);537;7;12;15;14;22;6;8;12;1;25;415;115;69;

ENTOMOL NEWS;ENTOMOLOGY;ENTOMOLOGY;4;ZOOTAXA;193;1;6;4;2;4;1;0;3;0;2;170;22;17;

ENTOMOL NEWS;ENTOMOLOGY;ENTOMOLOGY;4;Self-citation;13;2;1;0;2;0;0;0;0;8;3;3;

ENTOMOL NEWS;ENTOMOLOGY;ENTOMOLOGY;4;Ratio of self-citations;0,017496635;0,2;0,052631579;0;0;0,071428571;0;0;0;0,013490725;0,12406015;0,12406015;

ENTOMOL

NEWS;ENTOMOLOGY;ENTOMOLOGY;4;JIF;0;0,437;0,321;0,456;0,226;0,324;0,447;0,442;0,143;0,309;0;0;2,668;1,

774; ENTOMOL NEWS;ENTOMOLOGY;ENTOMOLOGY;4;JIF (without self-citations);0;0,425;0,321;0,378;0,172;0,284;0,447;0,423;0,143;0,295;0;0,2,463;1,602;

ENTOMOL RES;ENTOMOLOGY;ENTOMOLOGY;3;ALL

citations;467;18;45;47;26;21;29;26;37;15;27;176;273;168;

ENTOMOL RES;ENTOMOLOGY;ENTOMOLOGY;3;ALL citations (excluding self and most);418;13;36;42;19;19;26;24;35;13;24;167;238;142;

ENTOMOL RES;ENTOMOLOGY;ENTOMOLOGY;3;ZOOTAXA;13;0;0;0;3;0;0;1;0;1;2;6;7;3;

ENTOMOL RES;ENTOMOLOGY;ENTOMOLOGY;3;Self-citation;36;5;9;5;4;2;3;1;2;1;1;3;28;23;

ENTOMOL RES;ENTOMOLOGY;ENTOMOLOGY;3;Ratio of self-citations;0,077087794;0,277777778;0,2;0,106382979;0,153846154;0,095238095;0,103448276;0,038461538;0,05405405

4;0,066666667;0,037037037;0,017045455;0,8551348;0,658915504;

ENTOMOL

RES;ENTOMOLOGY;ENTOMOLOGY;3;JIF;0;0,807;0,564;0,462;0,573;0,646;0,398;0,33;0;0;0;2,973;2,643;

ENTOMOL RES;ENTOMOLOGY;ENTOMOLOGY;3;JIF (without self-
 citations);0;0,684;0,479;0,396;0,415;0,38;0,341;0,284;0;0;0;2,295;2,011;
 ENTOMOL SCI;ENTOMOLOGY;ENTOMOLOGY;3;ALL
 citations;753;23;46;55;52;70;67;38;46;63;40;253;477;290;
 ENTOMOL SCI;ENTOMOLOGY;ENTOMOLOGY;3;ALL citations (excluding self and
 most);689;21;44;52;52;61;65;36;42;57;35;224;444;274;
 ENTOMOL SCI;ENTOMOLOGY;ENTOMOLOGY;3;ZOOTAXA;41;1;0;3;0;5;2;1;1;2;1;25;15;10;
 ENTOMOL SCI;ENTOMOLOGY;ENTOMOLOGY;3;Self-citation;23;1;2;0;0;4;0;1;3;4;4;4;18;6;
 ENTOMOL SCI;ENTOMOLOGY;ENTOMOLOGY;3;Ratio of self-
 citations;0,030544489;0,043478261;0,043478261;0;0;0,057142857;0;0,026315789;0,065217391;0,063492063;0,1;0,015
 810277;0,355646362;0,100621118;
 ENTOMOL
 SCI;ENTOMOLOGY;ENTOMOLOGY;3;JIF;0;1,074;1,073;1,069;1,262;1,144;1,065;1,116;0,981;0,673;0,686;0;9,069;5
 ,613;
 ENTOMOL SCI;ENTOMOLOGY;ENTOMOLOGY;3;JIF (without self-
 citations);0;1,053;1,037;1,052;1,206;1,067;0,991;1,018;0,861;0,615;0,608;0;8,455;5,353;
 EUR J ENTOMOL;ENTOMOLOGY;ENTOMOLOGY;3;ALL
 citations;1974;5;54;91;82;131;130;81;102;74;77;1147;822;488;
 EUR J ENTOMOL;ENTOMOLOGY;ENTOMOLOGY;3;ALL citations (excluding self and
 most);1837;5;45;87;80;124;130;77;99;67;68;1055;777;466;
 EUR J ENTOMOL;ENTOMOLOGY;ENTOMOLOGY;3;ZOOTAXA;81;0;5;3;0;4;0;2;2;4;5;56;25;12;
 EUR J ENTOMOL;ENTOMOLOGY;ENTOMOLOGY;3;Self-citation;56;0;4;1;2;3;0;2;1;3;4;36;20;10;
 EUR J ENTOMOL;ENTOMOLOGY;ENTOMOLOGY;3;Ratio of self-
 citations;0,028368794;0,0,074074074;0,010989011;0,024390244;0,022900763;0;0,024691358;0,009803922;0,04054054
 1;0,051948052;0,031386225;0,259337964;0,132354092;
 EUR J
 ENTOMOL;ENTOMOLOGY;ENTOMOLOGY;3;JIF;0;1,051;0,965;1,017;1,167;0,954;0,975;1,076;0,918;1,061;0,945;
 0;9,078;5,078;
 EUR J ENTOMOL;ENTOMOLOGY;ENTOMOLOGY;3;JIF (without self-
 citations);0;1,014;0,901;0,948;1,118;0,867;0,93;0,955;0,81;0,946;0,869;0;8,344;4,764;
 FLA ENTOMOL;ENTOMOLOGY;ENTOMOLOGY;3;ALL
 citations;3285;16;63;110;220;244;256;241;147;150;72;1766;1503;893;
 FLA ENTOMOL;ENTOMOLOGY;ENTOMOLOGY;3;ALL citations (excluding self and
 most);2894;12;54;94;194;221;218;219;123;134;62;1563;1319;781;
 FLA ENTOMOL;ENTOMOLOGY;ENTOMOLOGY;3;ZOOTAXA;192;1;0;1;3;7;16;6;9;4;8;137;54;27;
 FLA ENTOMOL;ENTOMOLOGY;ENTOMOLOGY;3;Self-citation;199;3;9;15;23;16;22;16;15;12;2;66;130;85;
 FLA ENTOMOL;ENTOMOLOGY;ENTOMOLOGY;3;Ratio of self-
 citations;0,060578387;0,1875;0,142857143;0,136363636;0,104545455;0,06557377;0,0859375;0,066390041;0,10204081
 6;0,08;0,027777778;0,037372593;0,81148614;0,535277504;
 FLA
 ENTOMOL;ENTOMOLOGY;ENTOMOLOGY;3;JIF;0;0,972;0,973;1,052;0,964;0,975;0,997;1,056;1,163;1,363;1,052;
 0;9,595;4,961;
 FLA ENTOMOL;ENTOMOLOGY;ENTOMOLOGY;3;JIF (without self-
 citations);0;0,837;0,908;0,948;0,844;0,809;0,794;0,77;0,895;1,103;0,903;0;7,974;4,303;
 INSECTS;ENTOMOLOGY;ENTOMOLOGY;1;ALL
 citations;1925;192;347;370;244;225;132;156;190;59;2;8;1725;1318;
 INSECTS;ENTOMOLOGY;ENTOMOLOGY;1;ALL citations (excluding self and
 most);1646;100;268;324;230;208;123;151;178;54;2;8;1538;1153;
 INSECTS;ENTOMOLOGY;ENTOMOLOGY;1;ZOOTAXA;17;6;5;1;1;1;0;2;0;0;11;9;
 INSECTS;ENTOMOLOGY;ENTOMOLOGY;1;Self-citation;262;86;74;45;13;16;8;5;12;3;0;0;176;156;
 INSECTS;ENTOMOLOGY;ENTOMOLOGY;1;Ratio of self-
 citations;0,136103896;0,447916667;0,213256484;0,121621622;0,053278689;0,071111111;0,060606061;0,032051282;0,
 063157895;0,050847458;0;0;0,6659306;0,519873966;
 INSECTS;ENTOMOLOGY;ENTOMOLOGY;1;JIF;0;2,22;2,139;1,848;0;0;0;0;0;0;3,987;3,987;
 INSECTS;ENTOMOLOGY;ENTOMOLOGY;1;JIF (without self-
 citations);0;1,014;0,901;0,948;1,118;0,867;0,93;0,955;0,81;0,946;0,869;0;8,344;4,764;

INSECT SYST EVOL;ENTOMOLOGY;ENTOMOLOGY;2;ALL
citations;401;21;32;13;27;13;24;28;7;15;6;215;165;109;
INSECT SYST EVOL;ENTOMOLOGY;ENTOMOLOGY;2;ALL citations (excluding self and
most);302;15;23;9;18;12;15;24;4;11;6;165;122;77;
INSECT SYST EVOL;ENTOMOLOGY;ENTOMOLOGY;2;ZOOTAXA;96;6;9;3;8;1;9;4;3;4;0;49;41;30;
INSECT SYST EVOL;ENTOMOLOGY;ENTOMOLOGY;2;Self-citation;3;0;0;1;1;0;0;0;0;0;1;2;
INSECT SYST EVOL;ENTOMOLOGY;ENTOMOLOGY;2;Ratio of self-
citations;0,007481297;0;0;0,076923077;0,037037037;0;0;0;0;0,004651163;0,113960114;0,113960114;
INSECT SYST
EVOL;ENTOMOLOGY;ENTOMOLOGY;2;JIF;0;1,667;1,194;0,763;1,3;1,933;0,806;1,114;0,684;0,686;1;0;9,48;5,996;
INSECT SYST EVOL;ENTOMOLOGY;ENTOMOLOGY;2;JIF (without self-
citations);0;1,63;1,161;0,737;1,3;1,9;0,806;1,057;0,658;0,6;0,951;0,9,17;5,904;
INT J ACAROL;ENTOMOLOGY;ENTOMOLOGY;3;ALL
citations;950;29;48;70;46;58;47;49;50;44;23;486;435;269;
INT J ACAROL;ENTOMOLOGY;ENTOMOLOGY;3;ALL citations (excluding self and
most);690;23;38;52;32;41;31;36;35;33;18;351;316;194;
INT J ACAROL;ENTOMOLOGY;ENTOMOLOGY;3;ZOOTAXA;159;4;6;10;6;11;12;9;8;10;2;81;74;45;
INT J ACAROL;ENTOMOLOGY;ENTOMOLOGY;3;Self-citation;101;2;4;8;8;6;4;4;7;1;3;54;45;30;
INT J ACAROL;ENTOMOLOGY;ENTOMOLOGY;3;Ratio of self-
citations;0,106315789;0,068965517;0,083333333;0,114285714;0,173913043;0,103448276;0,085106383;0,081632653;0,
14;0,022727273;0,130434783;0,111111111;0,934881458;0,56008675;
INT J
ACAROL;ENTOMOLOGY;ENTOMOLOGY;3;JIF;0;0,894;1,236;1,008;0,919;0,774;0,949;0,691;0,554;0,568;0,489;0;
7,188;4,886;
INT J ACAROL;ENTOMOLOGY;ENTOMOLOGY;3;JIF (without self-
citations);0;0,803;1,094;0,797;0,77;0,584;0,758;0,455;0,454;0,411;0,422;0;5,745;4,003;
INT J ODONATOL;ENTOMOLOGY;ENTOMOLOGY;3;ALL
citations;277;2;17;19;13;22;14;19;11;14;9;137;138;85;
INT J ODONATOL;ENTOMOLOGY;ENTOMOLOGY;3;ALL citations (excluding self and
most);204;1;8;13;12;16;10;17;6;10;5;106;97;59;
INT J ODONATOL;ENTOMOLOGY;ENTOMOLOGY;3;ZOOTAXA;55;0;7;5;1;5;1;1;5;4;4;22;33;19;
INT J ODONATOL;ENTOMOLOGY;ENTOMOLOGY;3;Self-citation;18;1;2;1;0;1;3;1;0;0;9;8;7;
INT J ODONATOL;ENTOMOLOGY;ENTOMOLOGY;3;Ratio of self-
citations;0,064981949;0,5;0,117647059;0,052631579;0,045454545;0,214285714;0,052631579;0;0;0,065693431;0,48
2650476;0,430018898;
INT J
ODONATOL;ENTOMOLOGY;ENTOMOLOGY;3;JIF;0;1,029;0,846;0,6;0,647;0,596;0,686;0,5;0,426;0,614;0,791;0;5
39 ,706;3,375;
INT J ODONATOL;ENTOMOLOGY;ENTOMOLOGY;3;JIF (without self-
citations);0;0,943;0,641;0,46;0,471;0,365;0,412;0,321;0,333;0,364;0,605;0;3,972;2,349;
J ARACHNOL;ENTOMOLOGY;ENTOMOLOGY;3;ALL
citations;1435;11;52;35;62;53;37;42;49;49;51;994;430;239;
J ARACHNOL;ENTOMOLOGY;ENTOMOLOGY;3;ALL citations (excluding self and
most);1176;6;42;26;49;42;27;32;34;39;43;836;334;186;
J ARACHNOL;ENTOMOLOGY;ENTOMOLOGY;3;ZOOTAXA;127;4;5;5;4;6;4;4;6;5;4;80;43;24;
J ARACHNOL;ENTOMOLOGY;ENTOMOLOGY;3;Self-citation;132;1;5;4;9;5;6;6;9;5;4;78;53;29;
J ARACHNOL;ENTOMOLOGY;ENTOMOLOGY;3;Ratio of self-
citations;0,091986063;0,090909091;0,096153846;0,114285714;0,14516129;0,094339623;0,162162162;0,142857143;0,1
83673469;0,102040816;0,078431373;0,078470825;1,119105437;0,612102636;
J
ARACHNOL;ENTOMOLOGY;ENTOMOLOGY;3;JIF;0;0,946;1,188;1,236;0,988;0,691;0,624;0,975;0,729;0,626;0,90
1;0;7,958;4,727;
J ARACHNOL;ENTOMOLOGY;ENTOMOLOGY;3;JIF (without self-
citations);0;0,848;1,024;1,045;0,835;0,617;0,535;0,885;0,639;0,475;0,768;0;6,823;4,056;

J ASIA PAC ENTOMOL;ENTOMOLOGY;ENTOMOLOGY;3;ALL citations;1517;46;167;278;191;120;157;68;101;95;64;230;1241;913;

J ASIA PAC ENTOMOL;ENTOMOLOGY;ENTOMOLOGY;3;ALL citations (excluding self and most);1343;36;139;248;161;105;146;62;91;80;61;214;1093;799;

J ASIA PAC ENTOMOL;ENTOMOLOGY;ENTOMOLOGY;3;ZOOTAXA;37;2;11;5;3;3;2;1;2;4;0;4;31;24;

J ASIA PAC ENTOMOL;ENTOMOLOGY;ENTOMOLOGY;3;Self-citation;137;8;17;25;27;12;9;5;8;11;3;12;117;90;

J ASIA PAC ENTOMOL;ENTOMOLOGY;ENTOMOLOGY;3;Ratio of self-citations;0,090309822;0,173913043;0,101796407;0,089928058;0,141361257;0,1;0,057324841;0,073529412;0,079207921;0,115789474;0,046875;0,052173913;0,805812368;0,490410562;

J ASIA PAC ENTOMOL;ENTOMOLOGY;ENTOMOLOGY;3;JIF;0;1,101;0,967;0,875;1,046;0,824;0,946;0,875;0,797;0;0;0,6,33;4,658;

J ASIA PAC ENTOMOL;ENTOMOLOGY;ENTOMOLOGY;3;JIF (without self-citations);0;0,998;0,914;0,693;0,87;0,712;0,837;0,771;0,665;0;0;0,5,462;4,026;

J ENTOMOL RES SOC;ENTOMOLOGY;ENTOMOLOGY;4;ALL citations;120;3;9;12;15;6;2;10;16;26;8;13;104;44;

J ENTOMOL RES SOC;ENTOMOLOGY;ENTOMOLOGY;4;ALL citations (excluding self and most);99;3;9;11;9;4;2;10;13;25;5;8;88;35;

J ENTOMOL RES SOC;ENTOMOLOGY;ENTOMOLOGY;4;ZOOTAXA;14;0;0;1;5;0;0;0;0;1;2;5;9;6;

J ENTOMOL RES SOC;ENTOMOLOGY;ENTOMOLOGY;4;Self-citation;7;0;0;0;1;2;0;0;3;0;1;0;7;3;

J ENTOMOL RES SOC;ENTOMOLOGY;ENTOMOLOGY;4;Ratio of self-citations;0,058333333;0;0;0,066666667;0,333333333;0;0;0,1875;0;0,125;0;0,7125;0,4;

J ENTOMOL RES SOC;ENTOMOLOGY;ENTOMOLOGY;4;JIF;0;0,328;0,182;0,293;0,266;0,181;0,4;0,347;0,275;0,365;0,2;0,2,509;1,322;

J ENTOMOL RES SOC;ENTOMOLOGY;ENTOMOLOGY;4;JIF (without self-citations);0;0,328;0,167;0,293;0,234;0,181;0,314;0,267;0,174;0,25;0,175;0;2,055;1,189;

J HYMENOPT RES;ENTOMOLOGY;ENTOMOLOGY;2;ALL citations;574;17;46;69;53;61;29;36;33;10;17;203;354;258;

J HYMENOPT RES;ENTOMOLOGY;ENTOMOLOGY;2;ALL citations (excluding self and most);418;10;31;44;33;47;20;29;27;8;15;154;254;175;

J HYMENOPT RES;ENTOMOLOGY;ENTOMOLOGY;2;ZOOTAXA;77;1;6;13;9;5;5;4;1;2;1;30;46;38;

J HYMENOPT RES;ENTOMOLOGY;ENTOMOLOGY;2;Self-citation;79;6;9;12;11;9;4;3;5;0;1;19;54;45;

J HYMENOPT RES;ENTOMOLOGY;ENTOMOLOGY;2;Ratio of self-citations;0,137630662;0,352941176;0,195652174;0,173913043;0,20754717;0,147540984;0,137931034;0,083333333;0,151515152;0,0,058823529;0,093596059;1,15625642;0,862584405;

J HYMENOPT RES;ENTOMOLOGY;ENTOMOLOGY;2;JIF;0;1,322;0,939;0,902;0,793;0,783;0,903;0,966;0,524;0,531;0,5;0,6,841;4,32;

J HYMENOPT RES;ENTOMOLOGY;ENTOMOLOGY;2;JIF (without self-citations);0;1,08;0,827;0,745;0,69;0,71;0,833;0,475;0,333;0,51;0,341;0,5,464;3,805;

J KANSAS ENTOMOL SOC;ENTOMOLOGY;ENTOMOLOGY;4;ALL citations;1376;1;2;17;19;32;14;23;29;13;20;1206;169;84;

J KANSAS ENTOMOL SOC;ENTOMOLOGY;ENTOMOLOGY;4;ALL citations (excluding self and most);1241;1;2;17;18;23;12;22;27;12;18;1089;151;72;

J KANSAS ENTOMOL SOC;ENTOMOLOGY;ENTOMOLOGY;4;ZOOTAXA;135;0;0;0;1;9;2;1;2;1;2;117;18;12;

J KANSAS ENTOMOL SOC;ENTOMOLOGY;ENTOMOLOGY;4;Self-citation;0;0;0;0;0;0;0;0;0;0;0;0;0;0;

J KANSAS ENTOMOL SOC;ENTOMOLOGY;ENTOMOLOGY;4;Ratio of self-citations;0;0;0;0;0;0;0;0;0;0;0;0;0;0;

J KANSAS ENTOMOL SOC;ENTOMOLOGY;ENTOMOLOGY;4;JIF;0;0,292;0,216;0,299;0,505;0,277;0,539;0,397;0,551;0,493;0,653;0,3,93;1,836;

J KANSAS ENTOMOL SOC;ENTOMOLOGY;ENTOMOLOGY;4;JIF (without self-citations);0;0,292;0,189;0,299;0,418;0,231;0,474;0,346;0,526;0,384;0,611;0,3,478;1,611;

J LEPID SOC;ENTOMOLOGY;ENTOMOLOGY;4;ALL citations;500;1;22;29;20;21;21;14;12;11;4;345;154;113;

J LEPID SOC;ENTOMOLOGY;ENTOMOLOGY;4;ALL citations (excluding self and most);404;1;15;24;18;15;14;12;11;10;3;281;122;86;

J LEPID SOC;ENTOMOLOGY;ENTOMOLOGY;4;ZOOTAXA;32;0;2;2;0;1;3;0;0;1;0;23;9;8;

J LEPID SOC;ENTOMOLOGY;ENTOMOLOGY;4;Self-citation;64;0;5;3;2;5;4;2;1;0;1;41;23;19;
J LEPID SOC;ENTOMOLOGY;ENTOMOLOGY;4;Ratio of self-
citations;0,128;0,0,227272727;0,103448276;0,1;0,238095238;0,19047619;0,142857143;0,083333333;0,0,25;0,11884058
;1,335482908;0,859292432;
J LEPID
SOC;ENTOMOLOGY;ENTOMOLOGY;4;JIF;0;0,646;0,518;0,463;0,474;0,38;0,515;0,333;0,219;0,267;0,559;0,3,728;2
8 ,35;
J LEPID SOC;ENTOMOLOGY;ENTOMOLOGY;4;JIF (without self-
citations);0,0,544;0,386;0,4,0,382;0,354;0,333;0,27;0,125;0,233;0,525;0,3,008;1,855;
MYRMECOL NEWS;ENTOMOLOGY;ENTOMOLOGY;1;ALL
citations;615;3;35;75;44;12;69;47;81;53;37;159;453;235;
MYRMECOL NEWS;ENTOMOLOGY;ENTOMOLOGY;1;ALL citations (excluding self and
most);573;2;32;70;39;10;64;46;77;50;35;148;423;215;
MYRMECOL NEWS;ENTOMOLOGY;ENTOMOLOGY;1;ZOOTAXA;17;0;1;3;3;0;2;0;2;3;1;2;15;9;
MYRMECOL NEWS;ENTOMOLOGY;ENTOMOLOGY;1;Self-citation;25;1;2;2;2;3;1;2;0;1;9;15;11;
MYRMECOL NEWS;ENTOMOLOGY;ENTOMOLOGY;1;Ratio of self-
citations;0,040650407;0,333333333;0,057142857;0,026666667;0,045454545;0,166666667;0,043478261;0,021276596;0,
024691358;0,0,027027027;0,056603774;0,412403978;0,339408997;
MYRMECOL
NEWS;ENTOMOLOGY;ENTOMOLOGY;1;JIF;0;2,558;2,619;1,838;1,805;2,386;2,898;1,582;2,157;2,644;0;0,17,929;
11,546;
MYRMECOL NEWS;ENTOMOLOGY;ENTOMOLOGY;1;JIF (without self-
citations);0,2,465;2,167;1,568;1,463;2,159;2,265;1,373;1,549;2;0;0;14,544;9,622;
NEOTROP ENTOMOL;ENTOMOLOGY;ENTOMOLOGY;2;ALL
citations;2079;51;121;117;106;60;79;90;76;167;178;1034;994;483;
NEOTROP ENTOMOL;ENTOMOLOGY;ENTOMOLOGY;2;ALL citations (excluding self and
most);1903;29;106;100;99;55;74;81;68;160;166;965;909;434;
NEOTROP ENTOMOL;ENTOMOLOGY;ENTOMOLOGY;2;ZOOTAXA;70;7;5;3;3;4;2;3;3;3;34;29;17;
NEOTROP ENTOMOL;ENTOMOLOGY;ENTOMOLOGY;2;Self-citation;106;15;10;14;4;1;3;6;5;4;9;35;56;32;
NEOTROP ENTOMOL;ENTOMOLOGY;ENTOMOLOGY;2;Ratio of self-
citations;0,050986051;0,294117647;0,082644628;0,11965812;0,037735849;0,016666667;0,037974684;0,066666667;0,0
65789474;0,023952096;0,050561798;0,03384913;0,501649981;0,294679947;
NEOTROP
ENTOMOL;ENTOMOLOGY;ENTOMOLOGY;2;JIF;0;1,33;1,09;0,886;0,756;0,834;0,772;0,85;0,675;0,603;0,646;0;7,
112;4,338;
NEOTROP ENTOMOL;ENTOMOLOGY;ENTOMOLOGY;2;JIF (without self-
citations);0;1,196;1,039;0,863;0,705;0,804;0,707;0,775;0,634;0,538;0,558;0,6,623;4,118;
NOTA LEPIDOPTEROLOGI;ENTOMOLOGY;ENTOMOLOGY;3;ALL
citations;180;0;15;9;7;8;8;6;12;6;2;107;73;47;
NOTA LEPIDOPTEROLOGI;ENTOMOLOGY;ENTOMOLOGY;3;ALL citations (excluding self and
most);112;0;8;7;3;6;5;4;2;2;71;41;29;
NOTA LEPIDOPTEROLOGI;ENTOMOLOGY;ENTOMOLOGY;3;ZOOTAXA;55;0;5;2;2;1;1;1;8;4;0;31;24;11;
NOTA LEPIDOPTEROLOGI;ENTOMOLOGY;ENTOMOLOGY;3;Self-citation;13;0;2;0;2;1;2;1;0;0;5;8;7;
NOTA LEPIDOPTEROLOGI;ENTOMOLOGY;ENTOMOLOGY;3;Ratio of self-
citations;0,072222222;#DIV/0!;0,133333333;0,0,285714286;0,125;0,25;0,166666667;0;0;0,046728972;0,960714286;0,
794047619;
NOTA LEPIDOPTEROLOGI;ENTOMOLOGY;ENTOMOLOGY;3;JIF;0;0,75;0,25;0;0;0;0;0;0;0,25;0,25;
NOTA LEPIDOPTEROLOGI;ENTOMOLOGY;ENTOMOLOGY;3;JIF (without self-
citations);0,0,688;0,214;0;0;0;0;0;0,214;0,214;
NZ ENTOMOL;ENTOMOLOGY;ENTOMOLOGY;4;ALL citations;196;1;8;2;14;5;5;8;9;5;10;129;66;34;
NZ ENTOMOL;ENTOMOLOGY;ENTOMOLOGY;4;ALL citations (excluding self and
most);170;1;7;1;10;4;5;6;8;4;8;116;53;27;
NZ ENTOMOL;ENTOMOLOGY;ENTOMOLOGY;4;ZOOTAXA;19;0;0;0;3;1;0;1;1;1;11;8;4;
NZ ENTOMOL;ENTOMOLOGY;ENTOMOLOGY;4;Self-citation;7;0;1;1;1;0;0;1;0;0;1;2;5;3;

NZ ENTOMOL;ENTOMOLOGY;ENTOMOLOGY;4;Ratio of self-citations;0,035714286;0,0,125;0,5;0,071428571;0,0;0,125;0;0,1;0,015503876;0,921428571;0,696428571;

NZ ENTOMOL;ENTOMOLOGY;ENTOMOLOGY;4;JIF;0;0,588;0,429;0,615;0,517;0,92;0,867;0,688;0,793;0;0;0,4,829;3,348;

NZ ENTOMOL;ENTOMOLOGY;ENTOMOLOGY;4;JIF (without self-citations);0;0,471;0,381;0,538;0,483;0,8;0,567;0,469;0,517;0;0;0,3,755;2,769;

ODONATOLOGICA;ENTOMOLOGY;ENTOMOLOGY;4;ALL citations;455;1;11;7;10;15;12;13;12;8;10;356;98;55;

ODONATOLOGICA;ENTOMOLOGY;ENTOMOLOGY;4;ALL citations (excluding self and most);262;1;4;3;6;6;5;9;6;5;6;211;50;24;

ODONATOLOGICA;ENTOMOLOGY;ENTOMOLOGY;4;ZOOTAXA;137;0;2;2;5;5;3;5;3;2;108;29;16;

ODONATOLOGICA;ENTOMOLOGY;ENTOMOLOGY;4;Self-citation;56;0;5;2;2;4;2;1;1;0;2;37;19;15;

ODONATOLOGICA;ENTOMOLOGY;ENTOMOLOGY;4;Ratio of self-citations;0,123076923;0;0,454545455;0,285714286;0,2;0,266666667;0,166666667;0,076923077;0,083333333;0;0,2;0,103932584;1,733849484;1,373593074;

ODONATOLOGICA;ENTOMOLOGY;ENTOMOLOGY;4;JIF;0;0,439;0,5;0,769;0,718;0,521;0,276;0,305;0,483;0,355;0,355;0;4,282;2,784;

ODONATOLOGICA;ENTOMOLOGY;ENTOMOLOGY;4;JIF (without self-citations);0;0,268;0,405;0,667;0,538;0,354;0,155;0,254;0,367;0,242;0,274;0;3,256;2,119;

ORIENT INSECTS;ENTOMOLOGY;ENTOMOLOGY;4;ALL citations;323;9;14;9;8;13;7;7;6;7;1;242;72;51;

ORIENT INSECTS;ENTOMOLOGY;ENTOMOLOGY;4;ALL citations (excluding self and most);213;5;13;9;7;11;4;7;4;7;1;145;63;44;

ORIENT INSECTS;ENTOMOLOGY;ENTOMOLOGY;4;ZOOTAXA;94;2;1;0;0;1;1;0;0;0;89;3;3;

ORIENT INSECTS;ENTOMOLOGY;ENTOMOLOGY;4;Self-citation;16;2;0;0;1;1;2;0;2;0;0;8;6;4;

ORIENT INSECTS;ENTOMOLOGY;ENTOMOLOGY;4;Ratio of self-citations;0,049535604;0,222222222;0;0;0,125;0,076923077;0,285714286;0;0,333333333;0;0,033057851;0,820970696;0,487637363;

ORIENT INSECTS;ENTOMOLOGY;ENTOMOLOGY;4;JIF;0;0,333;0,278;0,195;0,238;0,36;0,268;0,176;0,173;0,263;0,164;0;2,115;1,339;

ORIENT INSECTS;ENTOMOLOGY;ENTOMOLOGY;4;JIF (without self-citations);0;0,333;0,259;0,122;0,214;0,32;0,232;0,157;0,173;0,263;0,082;0;1,822;1,147;

PAN PAC ENTOMOL;ENTOMOLOGY;ENTOMOLOGY;3;ALL citations;458;0;12;12;14;9;6;6;16;4;9;370;88;53;

PAN PAC ENTOMOL;ENTOMOLOGY;ENTOMOLOGY;3;ALL citations (excluding self and most);343;0;11;9;12;9;6;5;13;4;9;265;78;47;

PAN PAC ENTOMOL;ENTOMOLOGY;ENTOMOLOGY;3;ZOOTAXA;109;0;1;3;2;0;0;1;3;0;0;99;10;6;

PAN PAC ENTOMOL;ENTOMOLOGY;ENTOMOLOGY;3;Self-citation;6;0;0;0;0;0;0;0;0;0;6;0;0;

PAN PAC ENTOMOL;ENTOMOLOGY;ENTOMOLOGY;3;Ratio of self-citations;0,013100437;#DIV/0!;0;0;0;0;0;0;0,016216216;0;0;

PAN PAC ENTOMOL;ENTOMOLOGY;ENTOMOLOGY;3;JIF;0;0,727;0,389;0,526;0,421;0,444;0,617;0,464;0,387;0,304;0,472;0;4,024;2,397;

PAN PAC ENTOMOL;ENTOMOLOGY;ENTOMOLOGY;3;JIF (without self-citations);0;0,727;0,361;0,5;0,342;0,4;0,583;0,446;0,258;0,304;0,306;0,3,5;2,186;

P ENTOMOL SOC WASH;ENTOMOLOGY;ENTOMOLOGY;4;ALL citations;1270;5;24;48;44;16;13;10;13;26;21;1050;215;145;

P ENTOMOL SOC WASH;ENTOMOLOGY;ENTOMOLOGY;4;ALL citations (excluding self and most);888;2;15;33;38;15;7;7;10;19;14;728;158;108;

P ENTOMOL SOC WASH;ENTOMOLOGY;ENTOMOLOGY;4;ZOOTAXA;308;2;6;10;2;1;4;1;0;5;6;271;35;23;

P ENTOMOL SOC WASH;ENTOMOLOGY;ENTOMOLOGY;4;Self-citation;74;1;3;5;4;0;2;2;3;2;1;51;22;14;

P ENTOMOL SOC WASH;ENTOMOLOGY;ENTOMOLOGY;4;Ratio of self-citations;0,058267717;0,2;0,125;0,104166667;0,090909091;0,0,153846154;0,2;0,230769231;0,076923077;0,047619048;0,048571429;1,029233267;0,473921911;

P ENTOMOL SOC WASH;ENTOMOLOGY;ENTOMOLOGY;4;JIF;0;0,655;0,723;0,619;0,42;0,593;0,532;0,479;0,385;0,402;0,447;0;4,6;2,887;

P ENTOMOL SOC WASH;ENTOMOLOGY;ENTOMOLOGY;4;JIF (without self-
citations);0,582;0,545;0,452;0,232;0,525;0,484;0,423;0,33;0,333;0,406;0,3,73;2,238;
REV BRAS ENTOMOL;ENTOMOLOGY;ENTOMOLOGY;3;ALL
citations;1239;7;37;48;68;66;44;65;73;65;99;667;565;263;
REV BRAS ENTOMOL;ENTOMOLOGY;ENTOMOLOGY;3;ALL citations (excluding self and
most);990;5;31;35;57;47;34;54;64;54;94;515;470;204;
REV BRAS ENTOMOL;ENTOMOLOGY;ENTOMOLOGY;3;ZOOTAXA;183;1;3;6;7;13;4;9;8;5;4;123;59;33;
REV BRAS ENTOMOL;ENTOMOLOGY;ENTOMOLOGY;3;Self-citation;66;1;3;7;4;6;6;2;1;6;1;29;36;26;
REV BRAS ENTOMOL;ENTOMOLOGY;ENTOMOLOGY;3;Ratio of self-
citations;0,053268765;0,142857143;0,081081081;0,145833333;0,058823529;0,090909091;0,136363636;0,030769231;0,
01369863;0,092307692;0,01010101;0,043478261;0,659887234;0,513010671;
REV BRAS
ENTOMOL;ENTOMOLOGY;ENTOMOLOGY;3;JIF;0;0,825;1,101;0,88;0,711;0,659;0,597;0,67;0,577;0,536;0,514;0;
6,245;3,948;
REV BRAS ENTOMOL;ENTOMOLOGY;ENTOMOLOGY;3;JIF (without self-
citations);0,728;1,055;0,829;0,632;0,602;0,517;0,619;0,513;0,444;0,421;0,5,632;3,635;
REV COLOMB ENTOMOL;ENTOMOLOGY;ENTOMOLOGY;4;ALL
citations;389;0;10;24;18;23;28;23;32;44;27;160;229;103;
REV COLOMB ENTOMOL;ENTOMOLOGY;ENTOMOLOGY;4;ALL citations (excluding self and
most);358;0;9;23;17;21;27;21;30;40;22;148;210;97;
REV COLOMB ENTOMOL;ENTOMOLOGY;ENTOMOLOGY;4;ZOOTAXA;17;0;1;1;1;0;1;2;3;5;12;4;
REV COLOMB ENTOMOL;ENTOMOLOGY;ENTOMOLOGY;4;Self-citation;14;0;0;0;1;1;1;0;2;7;7;2;
REV COLOMB ENTOMOL;ENTOMOLOGY;ENTOMOLOGY;4;Ratio of self-
citations;0,035989717;#DIV/0!;0;0;0,043478261;0,035714286;0,043478261;0;0,045454545;0,074074074;0,04375;0,24
2199427;0,079192547;
REV COLOMB
ENTOMOL;ENTOMOLOGY;ENTOMOLOGY;4;JIF;0;0,42;0,197;0,203;0,253;0,219;0,36;0,331;0,197;0,248;0,265;0;
2,273;1,232;
REV COLOMB ENTOMOL;ENTOMOLOGY;ENTOMOLOGY;4;JIF (without self-
citations);0,42;0,145;0,177;0,253;0,219;0,36;0,291;0,18;0,181;0,181;0;1,987;1,154;
REV SOC ENTOMOL ARGE;ENTOMOLOGY;ENTOMOLOGY;4;ALL
citations;355;2;12;8;12;15;7;11;13;11;27;237;116;54;
REV SOC ENTOMOL ARGE;ENTOMOLOGY;ENTOMOLOGY;4;ALL citations (excluding self and
most);270;2;8;7;12;13;7;10;9;10;26;166;102;47;
REV SOC ENTOMOL ARGE;ENTOMOLOGY;ENTOMOLOGY;4;ZOOTAXA;66;0;2;0;0;2;0;1;3;1;0;57;9;4;
REV SOC ENTOMOL ARGE;ENTOMOLOGY;ENTOMOLOGY;4;Self-citation;19;0;2;1;0;0;0;1;0;1;14;5;3;
REV SOC ENTOMOL ARGE;ENTOMOLOGY;ENTOMOLOGY;4;Ratio of self-
citations;0,053521127;0,0,166666667;0,125;0;0;0;0,076923077;0,0,037037037;0,05907173;0,405626781;0,291666667
;
REV SOC ENTOMOL ARGE;ENTOMOLOGY;ENTOMOLOGY;4;JIF;0;0,455;0,25;0,38;0;0;0;0;0;0;0,63;0,63;
REV SOC ENTOMOL ARGE;ENTOMOLOGY;ENTOMOLOGY;4;JIF (without self-
citations);0,386;0,232;0,32;0;0;0;0;0;0,552;0,552;
SHILAP REV LEPIDOPT;ENTOMOLOGY;ENTOMOLOGY;4;ALL
citations;325;10;29;28;36;17;36;12;16;7;9;125;190;146;
SHILAP REV LEPIDOPT;ENTOMOLOGY;ENTOMOLOGY;4;ALL citations (excluding self and
most);109;0;8;15;18;7;8;6;3;2;4;38;71;56;
SHILAP REV LEPIDOPT;ENTOMOLOGY;ENTOMOLOGY;4;Self-citation;153;9;18;9;14;6;23;5;10;4;3;52;92;70;
SHILAP REV LEPIDOPT;ENTOMOLOGY;ENTOMOLOGY;4;ZOOTAXA;63;1;3;4;4;4;5;1;3;1;2;35;27;20;
SHILAP REV LEPIDOPT;ENTOMOLOGY;ENTOMOLOGY;4;Ratio of self-
citations;0,193846154;0,1,0,103448276;0,142857143;0,111111111;0,235294118;0,138888889;0,083333333;0,1875;0,14
2857143;0,222222222;0,28;1,367512235;0,731599536;
SHILAP REV
LEPIDOPT;ENTOMOLOGY;ENTOMOLOGY;4;JIF;0;0,491;0,35;0,223;0,264;0,408;0,435;0,304;0,306;0,312;0,133;0;
2,735;1,68;

SHILAP REV LEPIDOPT;ENTOMOLOGY;ENTOMOLOGY;4;JIF (without self-citations);0;0,259;0,145;0,134;0,127;0,117;0,118;0,072;0,139;0,15;0,033;0;1,035;0,641;

SOUTHWEST ENTOMOL;ENTOMOLOGY;ENTOMOLOGY;4;ALL citations;754;5;53;76;52;58;54;34;42;22;43;315;434;293;

SOUTHWEST ENTOMOL;ENTOMOLOGY;ENTOMOLOGY;4;ALL citations (excluding self and most);609;2;31;61;32;37;40;25;36;20;37;288;319;201;

SOUTHWEST ENTOMOL;ENTOMOLOGY;ENTOMOLOGY;4;Self-citation;115;3;16;12;14;18;11;7;5;1;5;23;89;71;

SOUTHWEST ENTOMOL;ENTOMOLOGY;ENTOMOLOGY;4;ZOOTAXA;30;0;6;3;6;3;3;2;1;1;1;4;26;21;

SOUTHWEST ENTOMOL;ENTOMOLOGY;ENTOMOLOGY;4;Ratio of self-citations;0,039787798;0;0,113207547;0,039473684;0,115384615;0,051724138;0,055555556;0,058823529;0,023809524;0,045454545;0,023255814;0,012698413;0,526688953;0,37534554;

SOUTHWEST ENTOMOL;ENTOMOLOGY;ENTOMOLOGY;4;JIF;0;0,561;0,565;0,462;0,482;0,478;0,462;0,407;0,504;0,422;0,329;0;4,111;2,449;

SOUTHWEST ENTOMOL;ENTOMOLOGY;ENTOMOLOGY;4;JIF (without self-citations);0;0,439;0,381;0,256;0,295;0,319;0,336;0,361;0,391;0,336;0,28;0;2,955;1,587;

SYST APPL ACAROL UK;ENTOMOLOGY;Top Ten;2;ALL citations;1164;71;286;237;140;99;29;83;26;25;21;147;946;791;

SYST APPL ACAROL UK;ENTOMOLOGY;Top Ten;2;ALL citations (excluding self and most);543;20;107;105;69;53;13;49;11;13;11;92;431;347;

SYST APPL ACAROL UK;ENTOMOLOGY;Top Ten;2;ZOOTAXA;220;12;73;42;20;14;7;9;7;7;6;23;185;156;

SYST APPL ACAROL UK;ENTOMOLOGY;Top Ten;2;Self-citation;401;39;106;90;51;32;9;25;8;5;4;32;330;288;

SYST APPL ACAROL UK;ENTOMOLOGY;Top Ten;2;Ratio of self-citations;0,344501718;0,549295775;0,370629371;0,379746835;0,364285714;0,323232323;0,310344828;0,301204819;0,307692308;0,2;0,19047619;0,217687075;2,747612389;1,748239071;

SYST APPL ACAROL UK;ENTOMOLOGY;Top Ten;2;JIF;0;1,614;1,732;1,696;1,467;1,378;1,253;1,115;0;0;0;8,641;7,526;

SYST APPL ACAROL UK;ENTOMOLOGY;Top Ten;2;JIF (without self-citations);0;1,009;0,923;1,152;1,148;0,844;0,791;1;0;0;0;5,858;4,858;

SYST ENTOMOL;ENTOMOLOGY;ENTOMOLOGY;1;ALL citations;2604;78;198;189;160;257;185;154;122;95;137;1029;1497;989;

SYST ENTOMOL;ENTOMOLOGY;ENTOMOLOGY;1;ALL citations (excluding self and most);2009;53;153;142;115;212;142;123;88;75;109;797;1159;764;

SYST ENTOMOL;ENTOMOLOGY;ENTOMOLOGY;1;ZOOTAXA;413;21;22;24;34;25;31;24;24;17;16;175;217;136;

SYST ENTOMOL;ENTOMOLOGY;ENTOMOLOGY;1;Self-citation;182;4;23;23;11;20;12;7;10;3;12;57;121;89;

SYST ENTOMOL;ENTOMOLOGY;ENTOMOLOGY;1;Ratio of self-citations;0,069892473;0,051282051;0,116161616;0,121693122;0,06875;0,077821012;0,064864865;0,045454545;0,081967213;0,031578947;0,087591241;0,055393586;0,695882561;0,449290614;

SYST ENTOMOL;ENTOMOLOGY;ENTOMOLOGY;1;JIF;0;3,909;3,727;4,237;4,474;3,343;2,784;2,553;2,876;2,943;2,568;0;29,505;18,565;

SYST ENTOMOL;ENTOMOLOGY;ENTOMOLOGY;1;JIF (without self-citations);0;3,444;3,424;4;3,969;3,067;2,557;2,351;2,557;2,552;2,351;0;26,828;17,017;

T AM ENTOMOL SOC;ENTOMOLOGY;ENTOMOLOGY;4;ALL citations;702;0;9;18;7;5;6;4;2;7;5;639;63;45;

T AM ENTOMOL SOC;ENTOMOLOGY;ENTOMOLOGY;4;ALL citations (excluding self and most);411;0;4;8;2;2;2;1;5;4;381;30;18;

T AM ENTOMOL SOC;ENTOMOLOGY;ENTOMOLOGY;4;ZOOTAXA;277;0;3;6;4;1;4;2;1;2;1;253;24;18;

T AM ENTOMOL SOC;ENTOMOLOGY;ENTOMOLOGY;4;Self-citation;14;0;2;4;1;2;0;0;0;0;5;9;9;

T AM ENTOMOL SOC;ENTOMOLOGY;ENTOMOLOGY;4;Ratio of self-citations;0,01994302;#DIV/0!;0,222222222;0,222222222;0,142857143;0,4;0;0;0;0;0,007824726;0,987301587;0,987301587;

T AM ENTOMOL SOC;ENTOMOLOGY;ENTOMOLOGY;4;JIF;0;0,365;0,291;0,457;0,27;0,647;0,2;0,206;0,216;0,311;0,282;0,2,88;1,865;

T AM ENTOMOL SOC;ENTOMOLOGY;ENTOMOLOGY;4;JIF (without self-
citations);0;0,284;0,2;0,2;0,27;0,588;0,2;0,206;0,216;0,222;0,205;0,2,307;1,458;
PALEONTOL J;-;Top Ten;4;ALL citations;1627;27;44;77;108;95;61;78;61;72;67;937;663;385;
PALEONTOL J;-;Top Ten;4;ALL citations (excluding self and
most);1314;8;33;57;71;70;52;62;46;54;55;806;500;283;
PALEONTOL J;-;Top Ten;4;ZOOTAXA;44;1;0;2;3;2;2;0;6;2;2;24;19;9;
PALEONTOL J;-;Top Ten;4;Self-citation;269;18;11;18;34;23;7;16;9;16;10;107;144;93;
PALEONTOL J;-;Top Ten;4;Ratio of self-
citations;0,165334972;0,666666667;0,25;0,233766234;0,314814815;0,242105263;0,114754098;0,205128205;0,1475409
84;0,222222222;0,149253731;0,114194237;1,879585552;1,15544041;
PALEONTOL J;-;Top Ten;4;JIF;0;0,5;0,716;0,608;0,48;0,57;0,514;0,579;0,472;0,454;0,591;0,4,984;2,888;
PALEONTOL J;-;Top Ten;4;JIF (without self-
citations);0;0,38;0,46;0,416;0,341;0,372;0,327;0,295;0,335;0,294;0,345;0,3,185;1,916;
CRETACEOUS RES;-;Top Ten;2;ALL citations;4914;173;521;395;490;380;186;271;437;138;79;1844;2897;1972;
CRETACEOUS RES;-;Top Ten;2;ALL citations (excluding self and
most);3719;121;317;265;382;262;148;212;300;112;63;1537;2061;1374;
CRETACEOUS RES;-;Top Ten;2;ZOOTAXA;91;8;23;15;10;6;1;2;17;0;0,9;74;55;
CRETACEOUS RES;-;Top Ten;2;Self-citation;1104;44;181;115;98;112;37;57;120;26;16;298;762;543;
CRETACEOUS RES;-;Top Ten;2;Ratio of self-
citations;0,224664225;0,25433526;0,347408829;0,291139241;0,2;0,294736842;0,198924731;0,210332103;0,274599542
;0,188405797;0,202531646;0,161605206;2,208078731;1,332209643;
CRETACEOUS RES;-;Top Ten;2;JIF;0;1,854;2,12;1,928;2,015;2,196;1,904;2,39;1,63;1,537;1,706;0;17,426;10,163;
CRETACEOUS RES;-;Top Ten;2;JIF (without self-
citations);0;1,255;1,377;1,433;1,341;1,596;1,575;1,898;1,241;1,354;1,552;0;13,367;7,322;
J SYST PALAEONTOL;-;Top Ten;2;ALL citations;1448;107;182;124;93;107;88;112;92;85;105;353;988;594;
J SYST PALAEONTOL;-;Top Ten;2;ALL citations (excluding self and
most);1356;103;164;116;83;102;81;104;89;80;97;337;916;546;
J SYST PALAEONTOL;-;Top Ten;2;ZOOTAXA;31;2;5;2;3;1;4;2;0;2;5;5;24;15;
J SYST PALAEONTOL;-;Top Ten;2;Self-citation;61;2;13;6;7;4;3;6;3;3;3;11;48;33;
J SYST PALAEONTOL;-;Top Ten;2;Ratio of self-
citations;0,042127072;0,018691589;0,071428571;0,048387097;0,075268817;0,037383178;0,034090909;0,053571429;0,
032608696;0,035294118;0,028571429;0,031161473;0,416604243;0,266558572;
J SYST PALAEONTOL;-;Top Ten;2;JIF;0;2,833;2,315;2,326;2,963;3,143;3,727;2,852;2,25;3;3,844;0;26,42;14,474;
J SYST PALAEONTOL;-;Top Ten;2;JIF (without self-
citations);0;2,657;2,228;2,244;2,877;3,026;3,652;2,689;2,173;2,912;3,625;0;25,426;14,027;
;;;;;;0,527494908;0,414285714;0,600896861;0,481060606;0,286764706;0,271264368;0,236842105;0,454248366;0,48076
9231;0,248120301;,,,,
;;;;;;0,386028029;,,,,;,,,,,

Appendix C

Rebuttal Letter

Dear Dr. Jeff Thompson,
Associate Editor, RSOS

Thank you for sending the revised version of our manuscript.

We were pleased with all corrections and suggestions, especially the useful comments by the reviewer 3, Camilla Souto. We virtually accepted all corrections and suggestions, including minor changes did the reviewer 3 in the PDF file. Certainly, these suggestions improved the general quality of the text. We concluded the review, and a new version was uploaded in the system. All modifications were did using track changes tool, thus both marked and clean versions were uploaded such requested by the journal. We rewrote sentences or added others aiming to make the text clearer and, this way, adjust to the required suggestions. We consider all reviews when adjusting the text. The major criticism was about the somewhat vague aspects about the methodology adopted to select the analyzed journals and the length of text. The first shortcoming we hope to be solved with inclusion of a Venn diagram and added more details in M&M section. However, the manuscript itself is not much long, it has just twelve pages of text including references. The primary data used in the analysis (supplementary file 1) occupy more than half of file length of early version. We are confident that merging the Results with Discussion into a single section is not the best format for this study. Reviewer 1 suggested to turn "details" of the analysis available as supplementary files. We believe it refers to the results, which we agree, so 2 figures were moved to the supplementary files. For this version we exclude deleted all figures embedded in the text file. Even though we do not disagree of any suggestion and rewrote sentences to fulfill all raised questions a few points are answered in detail below.

Thank you once again and please, get in contact if have any question.

Sincerely,
Ângelo

Reviewer: 1

Comment: Donat Agosti and others have published compelling arguments for open content of taxonomic information that might be considered in the conclusions (at authors' discretion). A paper by Valdecasas, Castroviejo and Marcus (Nature 403: 698) should be considered for citation.

The conclusions and observations in this manuscript are important and timely. I believe the paper can have a greater impact if it is tightened up and significantly shortened with details of analysis available as an online source.

One apparent disconnect exists in the concluding arguments. Clearly it is good advice to pay greater attention to employment for those doing taxonomy, but so long as administrators rely on citation impact in their hiring decisions it is not clear how to bridge this gap in thinking by institutional leaders. The answer is not immediately clear, for reasons explained well in the text about the growing reliance on indices.

Author's answer: Indeed, Donat Agosti has many studies on open data and taxonomy, however these are not strongly linked to our study. We knew the paper by Valdecasas et al. (2000), but there are many more on this subject and those cited in our manuscript play the role of illustrating similar views.

We agree that some passages, even essential, would be considered a bit long, but most likely is no allowed by the journal guidelines make available details of analysis as supplementary files, if the reviewer is referring to the figures we moved two as supplementary files. Finally, we worked on the discussion to clarify some sentences.

Reviewer: 2

Nothing to mention.

Reviewer: 3

Comment: "Materials and Methods: Methods lack clarity and it's challenging to understand how the journals were chosen without first looking at the figures. Text should be improved to guarantee research replicability, especially the following: ..."

Author's answer: Almost all the points highlighted by the reviewer is concerning to the actual number of journals selected because some of them are included in multiple categories in the database of Web of Science. Although we believe have clarified all doubts already in the text after working on the sentences with putative misconceptions, we have also include a new figure to eliminate any doubts about the number of journals selected.

Comment: Results/Discussion “The way the manuscript is organized, the results seem secondary and the discussion largely an opinion piece. To improve the robustness of the discussion, I suggest the authors merge results & Discussion and present the results when they are pertinent to the discussion.”

Author’s answer: In despite that merging the sections Results and Discussion would be a feasible way of presenting our study, we are confident it is not be best format such it was written. First, we followed the RSOS template in which is required these sections independently and verified recent issues of the journal to elaborate our manuscript. Merging these sections will require an entire rearrangement and potentially will obscuring the main messages, this way we opted to maintain theses section separate.